# Transcriptional linkage analysis with in vivo AAV-Perturb-seq

Antonio J. Santinha[1], Esther Klingler[2,7], Maria Kuhn[1,8], Rick Farouni[1], Sandra Lagler[1], Georgios Kalamakis[1,3], Ulrike Lischetti[1,9], Denis Jabaudon[2] & Randall J. Platt[1,4,5,6 ✉]

The ever-growing compendium of genetic variants associated with human pathologies demands new methods to study genotype–phenotype relationships in complex tissues in a high-throughput manner[1,2]. Here we introduce adeno-associated virus (AAV)-mediated direct in vivo single-cell CRISPR screening, termed AAV-Perturb-seq, a tuneable and broadly applicable method for transcriptional linkage analysis as well as high-throughput and high-resolution phenotyping of genetic perturbations in vivo. We applied AAV-Perturb-seq using gene editing and transcriptional inhibition to systematically dissect the phenotypic landscape underlying 22q11.2 deletion syndrome[3,4] genes in the adult mouse brain prefrontal cortex. We identified three 22q11.2-linked genes involved in known and previously undescribed pathways orchestrating neuronal functions in vivo that explain approximately 40% of the transcriptional changes observed in a 22q11.2-deletion mouse model. Our findings suggest that the 22q11.2-deletion syndrome transcriptional phenotype found in mature neurons may in part be due to the broad dysregulation of a class of genes associated with disease susceptibility that are important for dysfunctional RNA processing and synaptic function. Our study establishes a flexible and scalable direct in vivo method to facilitate causal understanding of biological and disease mechanisms with potential applications to identify genetic interventions and therapeutic targets for treating disease.

Advances in single-cell CRISPR screening methods are making it possible to study complex genotype–phenotype landscapes in a high-throughput manner[5,6]. The combination of pooled CRISPR libraries, lentiviral delivery and single-cell omics were applied in vitro to study protein misfolding[7], gene regulation[6] and immunity[8] as well as in vivo to study mouse neurodevelopment[9]. Although these efforts have fundamentally changed our ability to investigate the genetic networks underlying complex cellular processes, current methods are restricted to in vitro applications or a very narrow range of developmental timepoints, tissues and cell types conducive to lentiviral infection in vivo. A general framework for broadly applicable direct in vivo single-cell screens is urgently needed to enable the systematic interrogation of the growing catalogue of disease-associated risk alleles in disease-relevant cells and tissues[10], understand their causality, function and pathology, as well as develop new diagnostics and therapeutics[1].

To address this challenge, we developed AAV-Perturb-seq, an AAV-based single-cell or single-nucleus CRISPR screening method that is simple to implement, tuneable and broadly applicable for in vivo functional genomics studies. We achieved this by creating a recombinant AAV vector for efficient guide RNA (gRNA) expression and detection within single-cell libraries as well as optimizing delivery and transgene expression for obtaining large numbers of single nuclei infected by single viruses from complex tissues. The use of AAVs for in vivo delivery offers many advantages over previous lentivirus-based screening approaches that are commonly used in vitro[11], including the possibility of systemic delivery through intravenous injections leading to the targeting of a wide range of tissues and cell types in animals of any age in a tuneable manner. We applied AAV-Perturb-seq, using either gene editing in LSL-Cas9 mice[12] or transcriptional inhibition in dCas9-KRAB mice, to systematically investigate the genotype–phenotype landscape of individual genes linked to 22q11.2 deletion syndrome (hereafter, 22q11.2DS)—a complex genetic disorder that affects numerous organs, including the brain, in which dysfunction is typically clinically expressed as schizophrenia or autism spectrum disorder (ASD)[3,4]. Using our data-analysis pipeline, we extracted high-quality transcriptomes spanning perturbations and brain cell types, enabling us to highlight previously underappreciated genetic contributions and identify new cellular phenotypes that may contribute to 22q11.2DS pathology. Our results establish AAV-Perturb-seq as a robust methodology for transcriptional linkage analysis and systematic transcriptional profiling of genotype–phenotype landscapes in vivo.

[1]Department of Biosystems Science and Engineering, ETH Zurich, Basel, Switzerland. [2]Department of Basic Neurosciences, University of Geneva, Geneva, Switzerland. [3]Novartis Institutes for BioMedical Research, Basel, Switzerland. [4]Botnar Research Center for Child Health, Basel, Switzerland. [5]Department of Chemistry, University of Basel, Basel, Switzerland. [6]NCCR Molecular Systems Engineering, Basel, Switzerland. [7]Present address: VIB-KU Leuven Center for Brain & Disease Research, KU Leuven Department of Neurosciences, Leuven Brain Institute, Leuven, Belgium. [8]Present address: Pharma Research and Early Development (pRED), Roche, Basel, Switzerland. [9]Present address: Department of Biomedicine, University Hospital Basel and University of Basel, Basel, Switzerland. ✉e-mail: rplatt@ethz.ch

## In vivo single-nucleus CRISPR screening

Towards creating a robust and broadly applicable direct in vivo single-cell CRISPR screening platform, we reasoned that it must have the following features: (1) simple to apply in mouse models; (2) relevant to a broad range of tissues and cell types yet also tuneable to subsets of interest; (3) capable of inducing efficient genetic perturbations and recovering this information with a transcriptomic readout; and (4) delivery enabling low multiplicity of infection such that single cells receive single perturbations (Fig. 1a). We hypothesized that systemic AAV-mediated delivery may offer each of these features and we therefore set out to establish and characterize this approach in vivo in the mouse brain.

To test whether AAVs permit infection of many cells at a low multiplicity of infection, we performed an in vivo titration experiment. We prepared three AAV transfer plasmids to independently express mTag-BFP, Venus or mCherry under the control of a ubiquitous CBh promotor (Extended Data Fig. 1a). Each fluorescent protein was additionally fused to a KASH domain, which physically attaches proteins to the nuclear membrane, therefore enabling nucleus sorting. In each case, we used the AAV.PHP.B[13] capsid to achieve brain-wide infection after systemic delivery in LSL-Cas9 mice. We injected an equal mixture of the three viruses through the tail vein at a low ($2.5 \times 10^9$), medium ($5.0 \times 10^9$) or high ($2.5 \times 10^{10}$) dose of total virus particles (Extended Data Fig. 1b). Flow cytometry analysis of nuclei isolated from brain tissue revealed a direct correlation between the viral dose, the number of infected cells and the multiplicity of infection (Extended Data Fig. 1c,d). For subsequent experiments, we selected the dose of $5.0 \times 10^9$ AAV particles per mouse as it maximized the total number of cells infected with a single AAV (Fig. 1b and Extended Data Fig. 1d–f).

We next focused on establishing a method to capture both mRNA and CRISPR gRNA molecules from the same AAV-infected nucleus. The use of nuclei rather than cells permits the study of complex, mature tissues from which good-quality single-cell suspensions are challenging to obtain[14]. We designed two different strategies in which the gRNA expression cassette was either embedded within a mRNA (pAS006) or expressed independently (pAS088), enabling either 3′ (CROP-seq[5]) or 5′ (ECCITE-seq[15]) capture sequencing methods, respectively (Extended Data Fig. 1g,h). We injected AAV.PHP.B containing small pools of ten distinct gRNAs for each construct. Four weeks after injection, we isolated single nuclei and prepared single-nucleus RNA-sequencing (snRNA-seq) libraries using either the 5′ or 3′ capture method for cells infected with pAS088 or pAS006, respectively. The percentage of total nuclei with a gRNA detected was 65% (around 25 unique molecular identifiers (UMIs) per gRNA) and 20% (around 3 UMIs per gRNA) for the 5′- and 3′-based approaches, respectively, and most infected nuclei contained a unique gRNA (Extended Data Fig. 1i–k). Taken together, we established that the 5′-based approach, combining independent gRNA expression (pAS088) with 5′ capture sequencing, best captures mRNA and gRNA information from AAV-infected nuclei and we therefore proceeded with this method for the subsequent experiments (Fig. 1a).

## AAV-Perturb-seq of genes at the 22q11.2 locus

We used AAV-Perturb-seq to examine 22q11.2-locus genes in mature somatic cells in the prefrontal cortex of adult mice. Heterozygous deletion of the 22q11.2 locus is one of the most common chromosomal deletions in humans and results in a complex spectrum of phenotypes, including altered neuronal development and function[3], but the function(s) of individual 22q11.2 genes in the adult brain are poorly understood. To identify candidate genes that are important for brain function in adult mice, we analysed DropViz[16] data to measure the expression of the mouse homologues of 22q11.2 genes. This analysis revealed that 29 of the 37 genes in the locus are expressed in the adult mouse prefrontal cortex, a region that is thought to underlie many of

the 22q11.2DS neuropsychiatric manifestations[3] (Fig. 1c, Extended Data Fig. 2a and Supplementary Table 1). We designed a CRISPR gRNA library to target each of the 29 adult expressed genes with two independent gRNAs and included five control gRNAs targeting mouse safe-harbour[17] (SH) loci (Supplementary Table 2). We then used this library, packaged within AAV.PHP.B, to perform an AAV-Perturb-seq screen in vivo (Extended Data Fig. 1h).

We first focused our analysis on cell type identification and perturbation assignment (Extended Data Fig. 2b). Clustering analysis using Seurat[18] identified expected neuronal and non-neuronal brain cells, highlighting our ability to infect and recover transcriptional information from a broad range of cell types (Fig. 1d and Extended Data Fig. 2c–e). Expression of gRNA molecules was detected in all cell types, most of which contained a single gRNA (Extended Data Fig. 3a–c). Furthermore, we detected all gRNAs in the library, with average numbers of nuclei per gRNA ranging from around 10 in microglia to 400 in interneurons (Fig. 1e and Extended Data Fig. 3d). We did not observe a change in the composition of cell types (Extended Data Fig. 3e). Taken together, our direct in vivo screen experiment resulted in a single-nucleus transcriptomic dataset containing about 60,000 nuclei spanning 6 brain cell types and perturbation of all 22q11.2 genes expressed in the adult prefrontal cortex.

## Gene- and cell-specific phenotypes

We developed a data analysis pipeline to associate gRNAs, and therefore genetic perturbations, with cell-type-specific transcriptional phenotypes (Fig. 2a and Extended Data Fig. 2b). For each cell type separately, we created pseudobulk profiles by aggregating nuclei with the same perturbation and used edgeR[19] to calculate the pairwise differential expression between the control and each perturbation. We used pseudobulk rather than single-cell-specific methods given that it is less biased towards highly expressed genes and less prone to false-positives[20] (Extended Data Fig. 3f). Using our pseudobulk approach, we found substantial transcriptional phenotypes in four perturbations across all neuron types (*Dgcr8*, *Dgcr14*, *Gnb1l* and *Ufd1l*) (Fig. 2b). Transcriptional phenotype scoring analysis using the Hoteling's $T^2$ statistic[21] confirmed the identity of the genes with strong transcriptional phenotypes when perturbed in neurons (Extended Data Fig. 3g). We also observed that all four genes are present within the 1.5 Mb minimal region that is believed to be critical in 22q11.2-related disorders[3] (Fig. 1c).

Next, we characterized the transcriptional phenotypes resulting from perturbation of *Dgcr8*, *Dgcr14*, *Gnb1l* and *Ufd1l* across neuron types. The overarching result was that perturbation of each gene led to a largely distinct transcriptional phenotype that was mostly shared across neuron types. Support for this came from: (1) clustering the top 20 (Fig. 2c) or all (Extended Data Fig. 3h) upregulated genes for each perturbation; (2) Augur score analysis, which scores cells on the basis of their dissimilarity to the control condition (Extended Data Fig. 3i); (3) correlation analysis using all differentially expressed genes (DEGs) (Extended Data Fig. 3j); and (4) two-dimensional uniform manifold approximation and projection (UMAP) embeddings, which directly segregated nuclei with different perturbations from each other and from SH control cells (Fig. 2d). Taken together, these observations demonstrate that AAV-Perturb-seq retrieves both mutation and cell-type-specific signatures and indicate that perturbation of *Dgcr8*, *Dgcr14*, *Gnb1l* and *Ufd1l* affect specific subsets of unique genes across neuron types.

## In vivo gene editing efficiency

To assess whether underperforming gRNAs were confounding our ability to robustly identify perturbed cells, we prepared eight individual AAV.PHP.B viruses expressing gRNAs targeting the four genes

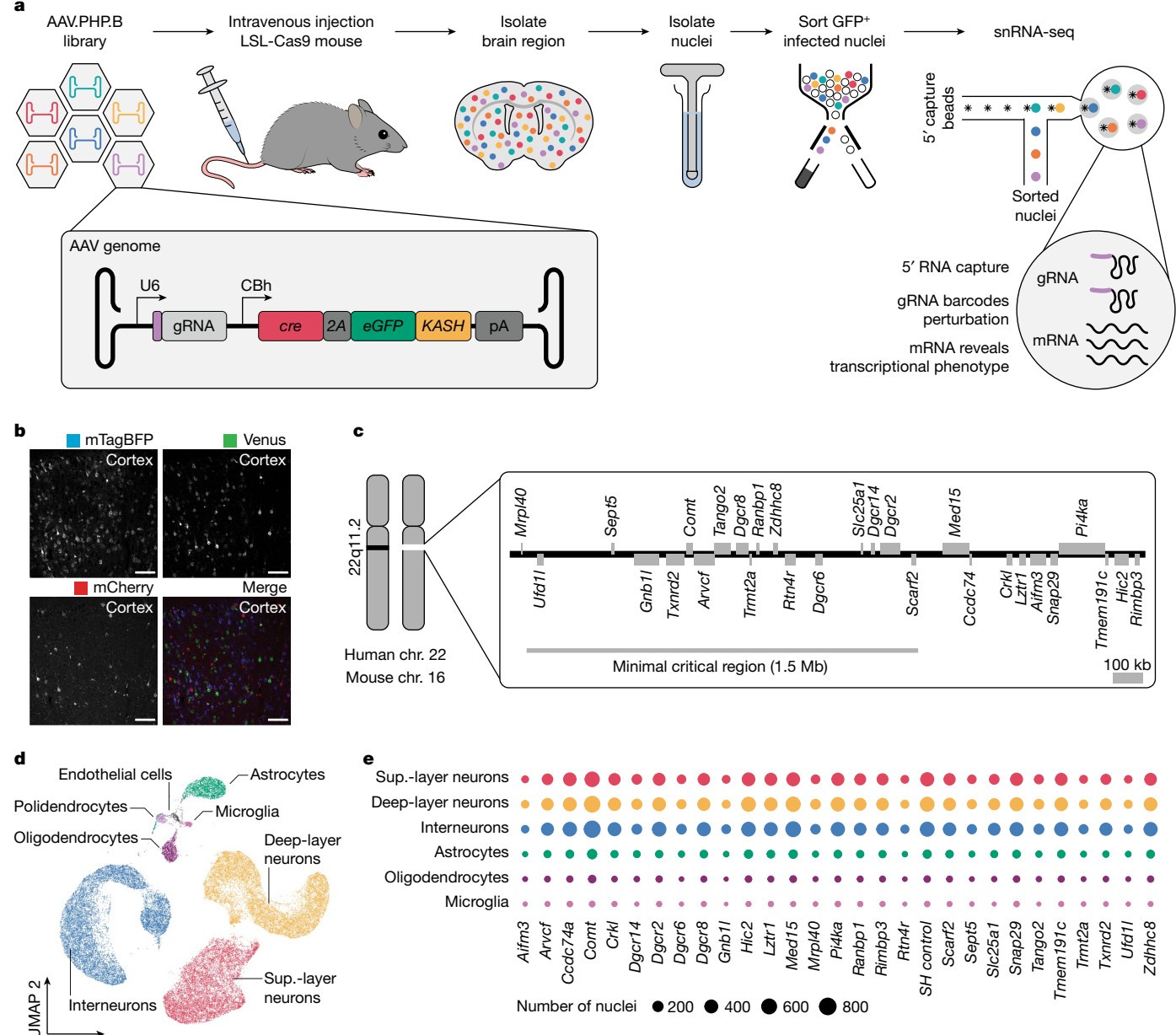

**Fig. 1 | In vivo single-nucleus pooled CRISPR screening in the adult brain enabled by systemic administration of AAV.PHP.B and 5′ gRNA capture.**
**a**, The AAV-Perturb-seq experimental pipeline. **b**, Expression of mTagBFP, Venus and mCherry in the prefrontal cortex after systemic injection of an equal mixture of $5.0 \times 10^9$ total AAV particles. Scale bars, 100 μm. The experiments were repeated in $n = 3$ mice. **c**, Representation of the 22q11.2 locus showing the genes expressed in the adult mouse prefrontal cortex. The human 22q11.2 locus is conserved in mouse chromosome (chr.) 16. **d**, UMAP embedding of around 150,000 AAV.PHP.B-infected nuclei isolated from the mouse prefrontal cortex. **e**, The number of nuclei with a unique gRNA for each perturbation across cell types.

with a strong transcriptional change (*Dgcr8*, *Dgcr14*, *Gnb1l* and *Ufd1l*) and four randomly chosen genes with no apparent transcriptional phenotype (*Comt*, *Med15*, *Ranbp1* and *Pi4ka*), and then individually injected these viruses into distinct mice. Analysis of Cas9-mediated mutations (indels) revealed that the percentage of mutated cells was similar across all tested gRNAs, with the majority of edited cells containing frame-shifting loss-of-function mutations in the targeted gene (Fig. 2e and Extended Data Fig. 4a), indicating that gene editing efficiency is not confounding our analysis.

While our analysis revealed efficient gene editing and DEGs across perturbations and neuron types, we set out to examine the possibility of another confounding factor—gene editing mosaicism. As not all nuclei expressing a gRNA necessarily carry a loss-of-function mutation, and

merging perturbed and non-perturbed transcriptomes into a pseudobulk profile could dampen and/or confound transcriptional phenotypes, we focused on identifying and filtering non-perturbed nuclei from the analysis (Extended Data Fig. 2b). Using the previously detected DEGs for each perturbation as variables (Fig. 2b), we used linear discriminant analysis (LDA) to identify gRNA-containing nuclei with a transcriptional phenotype that is significantly distinct from SH control nuclei. This analysis revealed that, on average, around 50% of nuclei containing a particular gRNA were perturbed (Extended Data Fig. 4b), consistent with our observed gene editing efficiency (Fig. 2e) and expected non-loss-of-function genotypes (Extended Data Fig. 4a). After discarding non-perturbed nuclei and repeating the pseudobulk differential expression analysis, we observed that nuclei filtering increased our

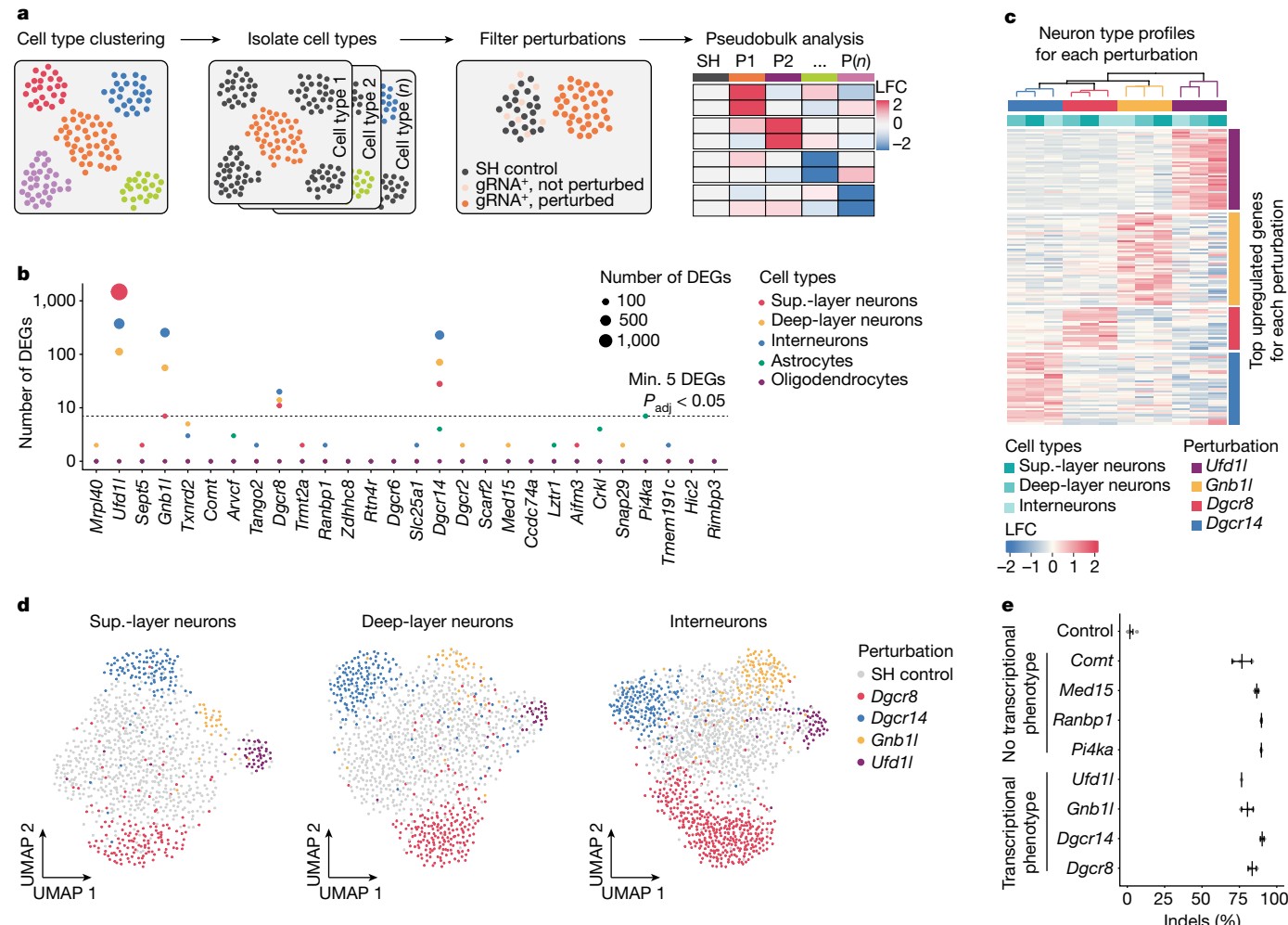

**Fig. 2 | Perturbation of the 22q11.2-linked genes *Dgcr8*, *Dgcr14*, *Gnb1l* and *Ufd1l* results in strong transcriptional changes in adult brain cell types.**
**a**, Schematic of the analysis pipeline. The SH control was nuclei with control gRNAs targeting the SH locus. P, perturbation. *n* is the total number of perturbations. **b**, The number of DEGs for all perturbations in individual cell types. The dashed line indicates five DEGs with an adjusted *P* value ($P_{adj}$) of less than 0.05. *P* values were calculated using edgeR-LRT with FDR correction for multiple comparisons. **c**, Heat map and hierarchical clustering of the 20 top upregulated genes (rows) after *Dgcr8*, *Dgcr14*, *Gnb1l* and *Ufd1l* perturbation in

the indicated neuron types (columns). **d**, UMAP embedding of control nuclei and nuclei passing the filter after perturbation in *Dgcr8*, *Dgcr14*, *Gnb1l* and *Ufd1l* for each neuron type using DEGs as variables (LFC > 0.5 and FDR < 0.01). **e**, Deep-sequencing-based gene editing (indel) analysis of four gRNAs targeting 22q11.2 genes with strong transcriptional phenotypes (*Dgcr8*, *Dgcr14*, *Gnb1l* and *Ufd1l*) and four gRNAs targeting genes without apparent transcriptional phenotypes (*Comt*, *Med15*, *Ranbp1* and *Pi4ka*). *n* = 3 biologically independent animals per target gene. Data are mean ± s.d.

sensitivity to detect DEGs without biasing the transcriptional phenotype (Extended Data Fig. 4c).

These results reveal the robustness of Cas9-mediated gene editing in vivo but also show that not all mutated genes lead to transcriptional phenotypes, which could be explained by subtle transcriptional changes in low-expressed genes not detected by snRNA-seq[22], genetic compensation mechanisms or a lack of transcriptional consequences after these perturbations in the cell type or state examined. Our approach therefore focuses on genes that result in substantial transcriptional changes when perturbed under homeostatic conditions.

## Arrayed perturbations support screening

To confirm the fidelity of our pooled screening method, we performed validation experiments by perturbing selected genes individually in vivo followed by analysis using snRNA-seq (Fig. 3a). In individual LSL-Cas9 mice, we injected AAVs expressing one gRNA targeting *Dgcr8*, *Dgcr14* or *Gnb1l* or an SH control. We excluded *Ufd1l* from this and subsequent

analyses after confirming that its transcriptional response was enriched for terms associated with apoptosis, consistent with its predicted role as an essential gene[3] (Extended Data Fig. 4d,e). Moreover, we focused our attention on neurons by exchanging the ubiquitous CBh promotor for the neuron-specific promoter hSyn (Extended Data Fig. 5a). After sequencing around 6,000 nuclei per condition (3 mice each), dataset integration and clustering revealed the presence of mostly neurons and only a residual level of non-neuronal cells (Extended Data Fig. 5b), with individual perturbations detected in all cell types (Extended Data Fig. 5c). Similar to our findings in the pooled screen, pseudobulk analysis results from the arrayed experiments highlighted strong transcriptional changes induced by all gRNAs, which were distinct from one another and led to condition-specific phenotypes (Extended Data Fig. 5d–f). A direct comparison of pooled and arrayed experiments revealed a high correlation between transcriptional phenotypes for all perturbations and neuron types, indicating that AAV-Perturb-seq has the ability to faithfully capture single-cell transcriptomes from pooled perturbation experiments (Fig. 3b).

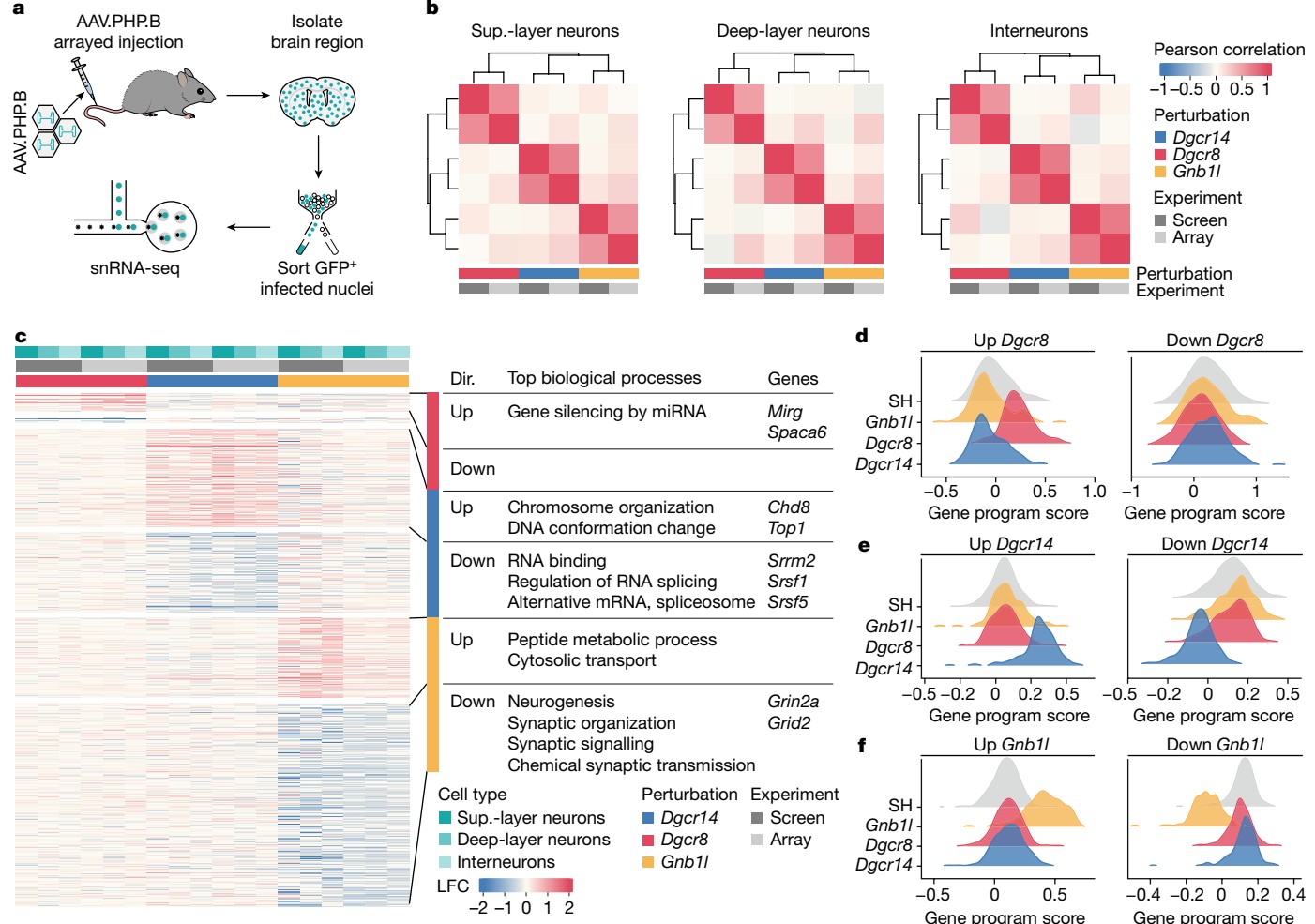

**Fig. 3 | Perturbation of 22q11.2 genes results in the disruption of distinct sets of biological processes. a**, Schematic of the arrayed validation experiments. **b**, Pearson correlation and hierarchical clustering of transcriptional signatures (LFC values) mediated by *Dgcr8*, *Dgcr14* or *Gnb1l* perturbation in pooled screen and arrayed confirmation experiments for each neuron type. **c**, Heat map showing the six transcriptional programs (grouped rows) altered in *Dgcr8*-, *Dgcr14*- and *Gnb1l*-perturbed cells (columns) across cell types and experiments (screen or arrayed). Left, LFC values for each altered gene across neuron types and experiments. Right, disrupted biological process for each genetic program

(top biological processes), direction of expression change (dir.) and representative genes. **d**–**f**, Gene program scores for upregulated (up) and downregulated (down) genes in interneurons with *Dgcr8* (**d**), *Dgcr14* (**e**) and *Gnb1l* (**f**) perturbation from the screen dataset. Extended Data Figure 8b–d contains ridge plots with gene expression scores for all neuron types and experiments. Statistical analysis was performed using one-way analysis of variance with post hoc Tukey's test; adjusted *P* < 0.01 (SH control versus gene program perturbation); comparisons of SH control versus other perturbations were not significant.

## Analysis of zygosity states

The control of zygosity is a general challenge in CRISPR screens and it is unclear whether heterozygous and homozygous mutations lead to the same transcriptional outcomes. This is especially important for modelling haploinsufficiency, as is the case for 22q11.2DS, motivating us to develop computational and experimental approaches to stratify and investigate these states. Analysis of SH control and perturbed nuclei using diffusion maps revealed an apparent bimodal distribution, probably capturing the transition from heterozygous to homozygous cell states (Extended Data Fig. 6a–i). The two groups are indistinguishable in terms of the dysregulated genes that define the transcriptional phenotypes but are distinguishable in terms of the expression levels of those genes, suggesting the lack of a haploinsufficiency-specific cell state (Extended Data Fig. 6j–o).

To further support these findings, we hypothesized that CRISPR inhibition (CRISPRi)-mediated knockdown may reduce target gene expression to levels observed in a heterozygous condition and may therefore be used to simulate the phenotypes generated by haploinsufficiency.

We prepared AAVs carrying SH- or *Dgcr8*-targeting gRNAs and injected them into a dCas9-KRAB mouse model (Extended Data Fig. 7a,b). Across neuron types, CRISPRi-mediated *Dgcr8* mRNA reduction was comparable to the values observed for 22q11.2DS[23] (Extended Data Fig. 7c). We also confirmed that CRISPRi- and Cas9-mediated *Dgcr8* perturbation led largely to analogous transcriptional phenotypes and changes in the known functions of *Dgcr8*[24] (Extended Data Fig. 7d–f). These results strongly indicate that heterozygous and homozygous mutations in *Dgcr8*, *Dgcr14* and *Gnb1l* result in a continuous phenotype and the assessment of both genotypes captures the effect of the perturbation and are relevant to haploinsufficiency.

## Perturbation-specific expression profiles

Next, we focused on characterizing the transcriptional phenotypes and disrupted biological processes resulting from perturbation of individual 22q11.2DS genes. For each perturbation, we divided dysregulated genes into two genetic programs to represent the upregulated and downregulated groups (Fig. 3c, Extended Data Fig. 7g and

Supplementary Tables 3–9). Gene Ontology (GO) analysis revealed dozens of disrupted biological processes (Fig. 3c, Extended Data Fig. 8a and Supplementary Table 10). To confirm that genes identified by differential expression analysis had altered expression, we calculated their gene program score—the average normalized expression of all genes in a program—for each nucleus in both screen and array datasets (Fig. 3d–f and Extended Data Fig. 8b–d). Across all of the neuron types, this analysis confirmed that programs are perturbation specific and their expression changes coincide with log-transformed fold change (LFC) values calculated by pseudobulk differential expression analysis, encouraging us to go deeper into functionally interpreting the data.

*Dgcr8* encodes a component of the microprocessor complex involved in processing primary microRNA (miRNA) transcripts (pri-miRNAs) into precursor miRNAs (pre-miRNAs), which are ultimately further processed by Dicer into mature miRNAs[24] (Extended Data Fig. 9a), and has been extensively studied in the context of 22q11.2DS[24,25]. In the upregulated genetic program, we identified a disruption in genes related to miRNA-mediated RNA silencing (Fig. 3c), which included several long non-coding RNAs (lncRNAs) such as *Mirg* and *Spaca6* (Extended Data Fig. 9b). These lncRNAs encode pri-miRNAs and their upregulation was previously reported in mouse models of *Dgcr8* haploinsufficiency and 22q11.2DS[23]. Moreover, we identified upregulation of the pri-miRNAs *Mir181a-1hg*, *Mir9-3hg* and *Mir124a-1hg* (Extended Data Fig. 9c), the miRNA products of which have been associated with cortical development and neuron physiology[25–28]. The accumulation of these pri-miRNAs implies that there is less mature miRNA being produced and that there is potentially an increase in the expression of genes targeted by the disrupted miRNAs (Extended Data Fig. 9a). Although miRNA-target enrichment analysis[29] revealed no significant enrichment for targets of *miR-9* and *miR-124a*, we observed a strong accumulation of *miR-181a* targets among upregulated genes in *Dgcr8*-perturbed cells (Extended Data Fig. 9d). This enrichment was not observed in *Dgcr14*- or *Gnb1l*-perturbed neurons, indicating that the miRNA-associated phenotype is unique to *Dgcr8*.

*Dgcr14* encodes the nuclear protein DGCR14[30], a component of C complex spliceosomes[31]. In the downregulated genetic program, we found a specific enrichment for genes associated with regulation of RNA splicing and the spliceosome, supporting the involvement of *Dgcr14* in RNA-maturation processes (Fig. 3c). Among splicing-related genes, we found constituents of the serine and arginine protein family (*Srrm2*, *Srsf1*, *Srsf2*, *Srsf5*, *Srsf6* and *Srsf11*) that are essential for spliceosome assembly[32] as well as constituents of the heterogeneous nuclear ribonucleoproteins family (*Hnrnph3*, *Hnrnpm* and *Hnrnpu*) that are involved in pre-mRNA processing, mRNA transport and metabolism[33]. An analysis of the upregulated genetic program showed disruption of chromatin binding and organization (Fig. 3c), which included dysregulation of the genes from the chromodomain-helicase-DNA-binding family (*Chd1*, *Chd3*, *Chd6* and *Chd8*), topoisomerases (*Top1* and *Top2b*) and *Setd5*, a gene that regulates chromatin structures to control RNA elongation and splicing[34]. Many of these genes and their respective pathways are associated with neurodevelopmental disorders[34].

*Gnb1l* encodes a protein of unknown function[35] that contains six WD40 repeats that facilitate protein–protein interactions and the formation of multiprotein complexes[36]. The downregulated genetic program was enriched for genes involved in neuronal development, synaptic organization and function, and chemical transmission (Fig. 3c), including genes that encode glutamatergic receptor subunits (*Gria1*, *Gria4*, *Grik3*, *Grin2a* and *Grin2b*), regulation of action potential (*Ank3*, *Cnr1*, *Fgf13*, *Foxp1* and *Trpc4*) and regulation of a prepulse inhibition phenotype (*Ctnna2* and *Nrxn1*) that is consistently observed in 22q11.2DS mouse models[3]. Overall, this suggests that perturbation of *Gnb1l* results in impaired neuronal communication that is distinct from those shown for other 22q11.2-linked genes[4].

These results show that AAV-Perturb-seq both confirms previously published work and provides new insights indicating that these 22q11.2 genes have an active role in mature neurons in the mouse brain, which may also contribute to 22q11.2DS symptomatology.

## 22q11.2DS mouse brain cell atlas

Our AAV-Perturb-seq screen revealed several mechanistic connections between 22q11.2-locus genes and biological processes in the adult mouse brain. We next set out to characterize the relationship between those findings versus transcriptional phenotypes observed in prefrontal cortex cells from a 22q11.2DS mouse model. We chose the LgDel model, which presents a deletion of 25 genes, analogous to the 1.5 Mb minimal critical deletion observed in patients with 22q11.2DS, and exhibits behavioural phenotypes reminiscent of ASD and schizophrenia[37,38].

We generated a prefrontal cortex cell atlas from *LgDel*[+/−] (LgDel) and *LgDel*[+/+] (wild type (WT)) adult male mice by recovering around 30,000 high-quality single nuclei using three animals per genotype (Fig. 4a and Extended Data Fig. 10a). Unbiased clustering and UMAP embedding of nucleus profiles from WT and LgDel samples revealed the presence of superficial and deep-layer excitatory neurons (layers 2/3, 5 and 6), inhibitory neurons (interneurons CGE and MGE) and non-neuronal brain cells, mainly oligodendrocytes and astrocytes (Fig. 4b). Clustering was unaffected by individual samples or the experimental conditions (Extended Data Fig. 10b). We did not detect significant differences in the proportion of cell populations between LgDel and WT mice, indicating that the deletion does not alter the gross cellular landscape of the adult brain (Extended Data Fig. 10c). Hierarchical clustering of the bulk transcriptional profiles revealed a primary clustering driven by cell type followed by second-level clustering by genotype (Extended Data Fig. 10d).

We applied our pseudobulk differential expression analysis pipeline to identify genes that are dysregulated in LgDel mice. First, focusing on genes that were previously targeted in our pooled screen, we found that only those within the 1.5 Mb deleted locus exhibited a significant negative LFC, whereas adjacent genes had minimal expression changes (Fig. 4c). This observation was consistent across cell types and confirms that locus heterozygosity leads to approximately a 50% reduction in the expression of affected genes. We next investigated 22q11.2-deletion-mediated transcriptional changes and found DEGs in all cell types (Extended Data Fig. 10e). Excitatory neurons presented the highest number of dysregulated genes, with superficial and deep-layer neurons showing 168 and 138 DEGs, respectively, whereas interneurons showed a substantially lower number of DEGs (23 genes). Correlation analysis using calculated LFC values for each cell type highlighted a strong similarity between neuron types, indicating that the deletion leads to similar signatures in all of the neuron types, but with a smaller amplitude in interneurons (Extended Data Fig. 10f).

To further support these findings and avoid a potential bias introduced by arbitrary DEG thresholds, we applied gene set enrichment analysis (GSEA) to identify biological processes that are dysregulated by the deletion. This analysis confirmed the similarity between the phenotypes in neuron types, with a strong overlap in the identified GO terms (Fig. 4d, Extended Data Fig. 10g and Supplementary Table 11). Although we identified an average of 68 DEGs in non-neuronal cells (Extended Data Fig. 10e), we found substantially less GO terms significantly affected by the deletion in these cells (Extended Data Fig. 10g). Among the downregulated biological processes altered by the deletion in neurons, we found terms related to neuronal communication (ion transmembrane transport and regulation of synaptic plasticity) and neuronal development (neurogenesis, cell projection organization and axonogenesis). These findings from adult mouse neurons also recapitulate recent findings from human cerebral spheroids derived from patients with 22q11.2DS[39] (Extended Data Fig. 10h). These results also reiterate many of the biological functions identified after perturbation of individual genes (Fig. 3c), suggesting that the 22q11.2DS phenotype in neurons may arise due to both dysfunctional development

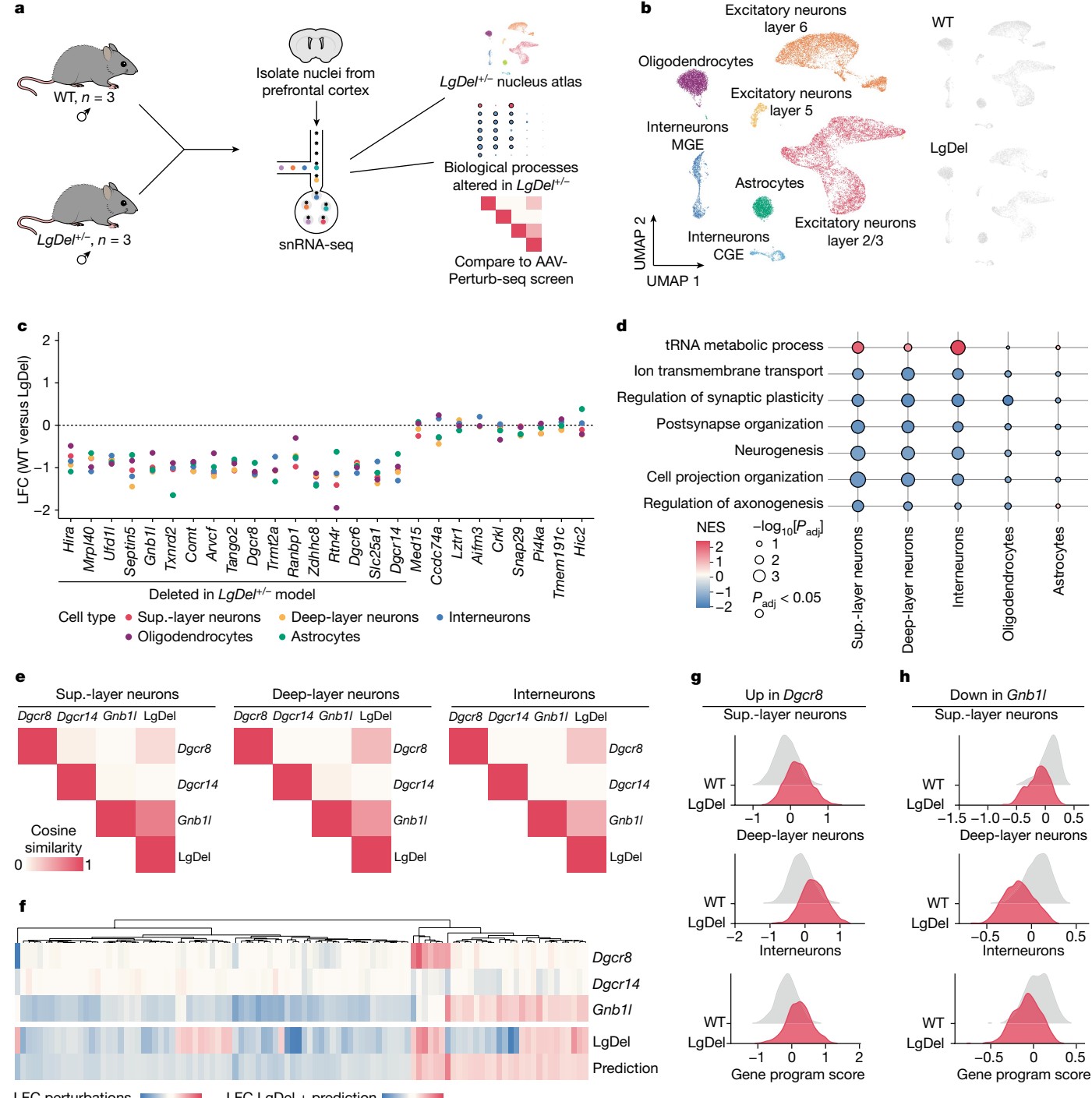

**Fig. 4 | Transcriptional changes found in LgDel model neurons are partially explained by perturbation of *Dgcr8*, *Dgcr14* and *Gnb1l*. a**, Schematic of the LgDel single-nucleus cortex atlas experimental design. snRNA-seq analysis of 10-week-old *LgDel*⁺/⁻ (LgDel) and WT control (*LgDel*⁺/⁺) mouse brain prefrontal cortex. *n* = 3 male mice for each condition. **b**, UMAP embedding depicting the cell types identified in WT and LgDel samples (left). Right, individual UMAP representations of WT and LgDel nuclei. **c**, Pseudobulk differential expression analysis of genes targeted in the pooled screen across cell types, comparing LgDel against the WT control. *Dgcr2* and *Rimbp3* were omitted due to their low expression levels and therefore inaccuracy in calculating the LFC. **d**, Biological processes enriched in LgDel transcriptional profiles from each cell type. NES, normalized enrichment score; $P_{adj}$, *P* value adjusted using Bonferroni's multiple-comparison test. **e**, Cosine similarity of LFC profiles between

individual perturbations and LgDel for each cell type. **f**, Heat map showing the LFC values for the top 100 predicted genes in individual perturbations, LgDel and the model (LgDel = 0.21 Dgcr8 + 0.18 Gnb1l + (−0.11) Dgcr14, dcor = 0.40) prediction based on individual perturbation profiles. Extended Data Fig. 11 shows similar heat maps for the other neuron types. **g**, The gene program score in WT control and LgDel nuclei for the upregulated program in *Dgcr8*-perturbed nuclei. Statistical analysis was performed using two-sided Student's *t*-tests, FDR-adjusted *P* values: <0.01 (superficial-layer neurons), <0.01 (deep-layer neurons) and <0.01 (interneurons). **h**, The gene program score in WT control and LgDel nuclei for the downregulated program in *Gnb1l*-perturbed nuclei. Statistical analysis was performed using two-sided Student's *t*-tests; FDR-adjusted *P* values: <0.01 (superficial-layer neurons), <0.01 (deep-layer neurons) and <0.01 (interneurons).

and the additive effects of reduced expression from a specific subset of single genes.

## Perturbations partially explain 22q11.2DS

We set out to quantify the extent to which individual perturbations explain the transcriptional signature observed in LgDel neurons. We started by considering the top DEGs for each perturbation and examined how their LFC values correlated across models. For all neuron types, we found that transcriptional changes mediated by *Dgcr8* and *Gnb1l* perturbation show a positive cosine similarity to the LgDel model (0.2 (*Dgcr8*) and 0.26 (*Gnb1l*)), while the same was not observed for *Dgcr14* (−0.11) (Fig. 4e). We next used a linear regression model[40] to assess the extent to which the LFC observations in the LgDel model are explained by individual perturbations. For all neurons, we observed that *Dgcr8* and *Gnb1l* perturbations have larger coefficients ($c_{Dgcr8} = 0.21$ and $c_{Gnb1l} = 0.18$), whereas *Dgcr14* showed the smallest contribution ($c_{Dgcr14} = −0.11$) (Fig. 4f and Extended Data Fig. 11a,b). The linear model was capable of predicting around 40% (dcor = 0.40) of the variance observed in the LgDel dataset. Of the transcriptional changes that were correctly predicted by individual perturbations (Fig. 4g,h), the *Dgcr8* contribution was focused on upregulated genes that are mostly associated with the accumulation of miRNA primary genes (*Mirg, Spaca6, Mir9-3hg* and *Mir181a-1hg*). The smaller *Dgcr14* contribution included downregulated spliceosomal genes *Srsf1, Srsf2* and *Srsf6*, whereas the *Gnb1l* contribution was primarily related to downregulation of genes involved with synapse signalling (Fig. 5 and Extended Data Fig. 11c,d). Together, these results show that perturbation of just three genes of the 22q11.2 locus directly in the adult brain prefrontal cortex explains 40% of the transcriptional changes observed in the LgDel model. This indicates that the 22q11.2DS phenotype observed in adult neurons may be partially explained by active disruption of cellular processes, and not exclusively a result of developmental defects.

## Disruption of disease-linked risk genes

Patients diagnosed with 22q11.2DS typically present brain functional and behavioural deficits that are associated with ASD, schizophrenia and other neurodevelopmental and psychiatric disorders[41]. Given these connections, we next examined whether the transcriptional signatures observed for individual perturbations and detected in LgDel neurons may increase the risk for those conditions through dysregulation of diseases susceptibility genes (Fig. 5). To answer this question, we analysed the intersection between our curated list of genes that are commonly dysregulated in the LgDel model and in individual perturbations with genes previously associated with ASD, schizophrenia, attention deficit hyperactivity disorder and bipolar disorder from the DisGeNET dataset[42]. We found the strongest overlap between genes that are downregulated by *Gnb1l* perturbation and the schizophrenia list (false-discovery rate (FDR)-adjusted $P < 0.001$, hypergeometric test), with GO analysis of those overlapping genes highlighting a strong presence of synaptic signalling genes (22 out of 44 genes) (Extended Data Fig. 11e,f). These results are consistent with recent studies indicating that ASD- and schizophrenia-associated proteins are strongly concentrated in pre- and post-synaptic compartments and are involved in functions related to synaptic organization, differentiation and transmission[43,44]. Overall, our results indicate that *Dgcr8, Dgcr14* and *Gnb1l* may contribute to 22q11.2DS by broadly altering the expression of genes associated with disease susceptibility in vivo, emerging after development and through a mechanism that involves RNA regulation in mature neurons.

## Discussion

Here we describe AAV-Perturb-seq—a direct in vivo single-cell CRISPR screening method that is tuneable, scalable and broadly applicable

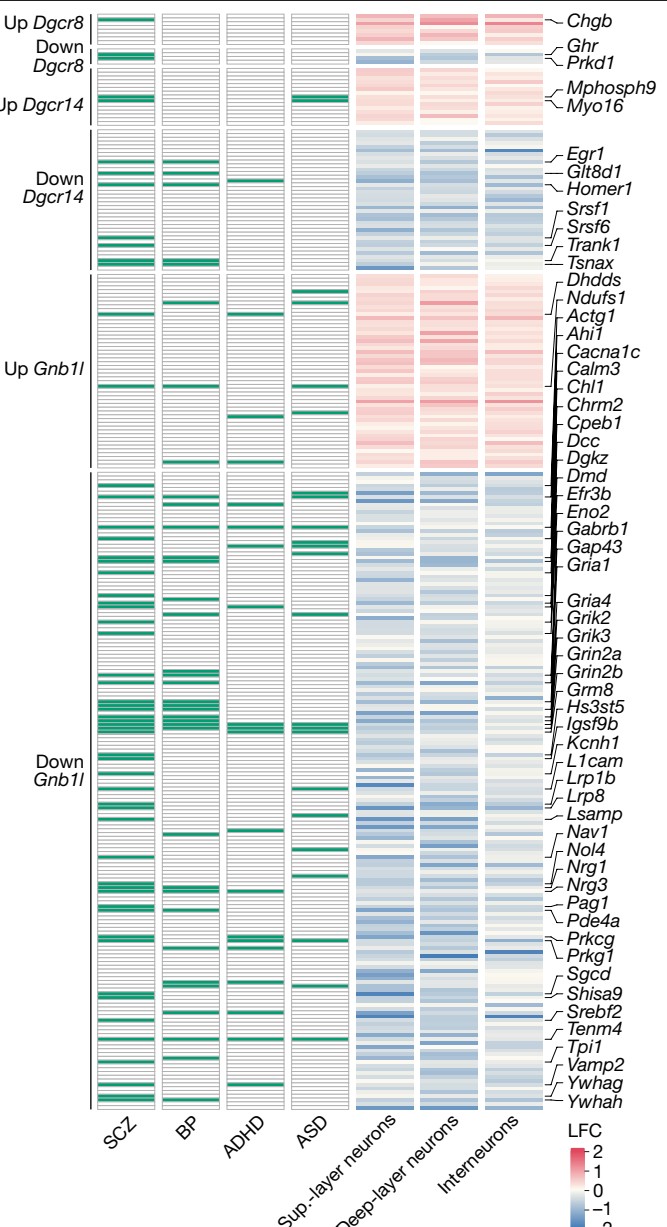

**Fig. 5 | LgDel and individual 22q11.2 gene perturbations alter the expression of disease-associated risk genes.** Heat map highlighting genes that are commonly dysregulated in individual perturbations and LgDel transcriptional profiles (right) and their association with neurodevelopmental disorders (left). ADHD, attention deficit hyperactivity disorder; BP, bipolar disorder; SCZ, schizophrenia.

for transcriptional linkage analysis and systematically investigating genetic elements in vivo in a high-throughput manner. Using a single systemic injection of AAVs containing a library of gRNAs and gene editing in LSL-Cas9 mice or transcriptional inhibition in dCas9-KRAB mice along with either constitutive or cell-type-specific promoters, AAV-Perturb-seq enables flexible perturbation of numerous disease-associated genes in all or specific brain cell types. AAV-Perturb-seq offers the opportunity to directly examine multiple genes in several cell types at the single-cell level in the same animal without restriction to tissue or developmental timepoints, opening further possibilities for studying processes of health and disease in vivo.

We used AAV-Perturb-seq to examine the genotype–phenotypic landscape underpinning 22q11.2DS. In contrast to other deletion syndromes in which the observed phenotype can be explained by single

genes, none of the genes within the 22q11.2 locus can largely explain the predisposition that 22q11.2DS confers for neurodevelopmental and psychiatric disorders[44]. Furthermore, the function of each gene in mature brain cells is poorly understood and has never been systematically investigated. In addition to providing in vivo evidence to support previous findings, such as *Ufd1l* cellular essentiality and *Dgcr8* mediating pri-miRNA processing, we also identified connections between *Dgcr14* and *Gnb1l* to adult neuron physiology relevant to 22q11.2DS pathology.

Approximately 40% of the transcriptional changes observed in the LgDel mouse model neurons could be recapitulated by the perturbation of three genes (individually) in adult animals. We suspect that the remaining transcriptional phenotype may be due to disruptions during development and/or genetic interactions among 22q11.2 genes or their downstream networks, as well as distinct non-cell autonomous interactions between the mosaic setting of the AAV-Perturb-seq experiments and the germline setting of the LgDel model, all of which represent promising areas for further study. Overall, our findings suggest that the 22q11.2DS transcriptional phenotype found in mature neurons may to some extent be due to the continuous reduction in gene expression of a specific subset of 22q11.2 genes after development. A promising area for further study is determining whether 22q11.2DS-associated neuronal and cognitive phenotypes can be rescued exclusively through restoring *Dgcr8*, *Dgcr14*, *Gnb1l* and/or *Ufd1l* expression during or after development.

We envision that AAV-Perturb-seq will broadly enable the study of genotype–phenotype landscapes directly in vivo in different tissues, cell types, developmental stages and under different health and disease contexts. The ability to study complex in vivo biology at scale could lead to breakthroughs in our causal understanding of biological and disease mechanisms as well as in our ability to identify genetic interventions and targets for treating disease.

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

## Methods

### Experimental procedures

**Plasmid design and cloning.** AAV genome plasmids (Fig. 1a and Extended Data Figs. 1a,g,h and 5a) were based on Addgene plasmid 60231 (ref. 12). To achieve widespread transgene expression, the hSyn promoter was replaced by the ubiquitous CBh promoter (pAS088). For the triple-colour experiments (Extended Data Fig. 1a), the U6 expression cassette and Cre were removed, while eGFP was replaced by mTagBF2 (Addgene plasmid, 55302), Venus (Addgene plasmid, 22663) or mCherry (Addgene plasmid, 27970) (pAS132, pAS133, pAS134). To prepare the 3′ capture AAV genome (Extended Data Fig. 1h), the original U6 expression cassette was first removed by restriction digestion with MluI (Thermo Fisher Scientific) and XbaI (Thermo Fisher Scientific) from upstream of the Pol II promoter and cloned between the WPRE and poly(A) signal sequences (pAS006).

**gRNA library design.** We focused on a set of genes located within the human 22q11.2 locus that is conserved in the mouse genome (Supplementary Table 1). Using BrainSpan data[45], we identified 29 genes with detectable expression in the adult mouse cortex. Individual gRNA sequences targeting those genes were designed using the online tool GUIDES (http://guides.sanjanalab.org/#/) (Supplementary Table 2). The two best-scoring gRNAs for each target were selected. As a control, we used SH-targeting gRNAs established previously[17]. The use of SH-targeting rather than non-targeting gRNAs enabled us to control for transcriptional changes induced by CRISPR–Cas9 DNA double-stranded breaks that are not directly related to the target gene of interest. To facilitate Gibson assembly cloning, we appended a 5′ arm (TGGAAAGGACGAAACACCG) and a 3′ arm (GTTTTAGAGCTAGAAATAGC AAGTTAAAATAAGGC) to the gRNA sequences. Sequences were ordered individually as single-strand oligo DNA nucleotides (ssODNs) and pooled at a final concentration of 100 mM.

**gRNA library cloning.** The plasmid backbone (2.5 μg) was digested with BsmBI (Thermo Fisher Scientific) for 1 h at 37 °C followed by an inactivation step for 5 min at 80 °C. The Gibson assembly reaction was set as follows: 50 ng of digested plasmid backbone, 2 μl (200 fmol of ssDNA oligos (stock at 100 mM), 10 μl NEBuilder HiFi DNA Assembly Master Mix (NEB, E2621L) and H$_2$O up to 20 μl total reaction. The reaction was incubated for 1 h at 50 °C. Isopropanol purification was used to concentrate the cloned gRNA library by mixing the total Gibson Assembly reaction with 20 μl isopropanol, 0.2 μl GlycoBlue Coprecipitant (Thermo Fisher Scientific, AM9515) and 0.4 μl NaCl solution (stock at 5 M). The precipitation reaction was incubated at room temperature for 15 min, followed by centrifugation at >15,000$g$ for 15 min at room temperature. The supernatant was discarded and the DNA pellet was washed with 1 ml ice-cold 80% ethanol and finally resuspended in 10 μl TE buffer.

**Plasmid amplification of pooled gRNA libraries.** Pooled gRNA libraries were amplified as previously described[46]. In brief, the plasmid library was electroporated into Endura ElectroCompetent cells (Lucigen, 60242-2) according to the manufacturer's instructions, followed by a 1 h recovery period at 37 °C. Bacteria were grown on a bioassay plate (Merck, D4803-1CS) for 14 h at 37 °C. The colonies were collected by scraping the plate surface before plasmid isolation using the QIAGEN Plasmid Maxi kit (Qiagen, 12165) according to the manufacturer's protocol. To confirm the distribution of gRNAs, the gRNA expression cassette was PCR amplified using KAPA HiFi ReadyMix with 100 ng of the final library as template and 0.5 μM of both custom Illumina P5 primer (AATGATACGGCGACCACCGAGATCTACA C-NNNNNNNN-ACACTCTTTCCCTACACGACGCTCTTCCGATCTGCTT TATATATCTTGTGGAAAGGACGAAACACC) and P7 primer (CAAGCAGAAG ACGGCATACGAGAT-NNNNNNNN-GTGACTGGAGTTCAGACGTGTGCTC

TTCCGATCCCGACTCGGTGCCACTTTTTCAA). PCR of the reaction mixture was performed as follows: (1) 95 °C for 3 min; (2) 18 cycles of 98 °C for 20 s, 63 °C for 15 s and 72 °C 20 s; and finally (3) 72 °C for 2 min. The PCR reaction was purified with double-size 0.6×–1.0× AMPURE bead selection (A63882, Beckman Coulter). Deep-sequencing libraries were sequenced using the NextSeq 550 75 cycle kit with the following cycle distribution: 75 to read 1, 8 to index 1 and 8 to index 2.

**AAV production and purification.** AAVs were produced in HEK293T cells (Sigma-Aldrich, regularly tested for mycoplasma) and purified by iodixanol gradient centrifugation. In brief, HEK293T cells were expanded in DMEM (Merck) + 10% FBS (Merck) + 1% HEPES (Thermo Fisher Scientific). Then, 24 h before the beginning of AAV production, cells were seeded in 15 cm dishes (HuberLab) at a density of 0.6 M cells per ml and a total of 20 ml medium per dish. Cells were transiently transfected with 21 μg of an equal molar-ratio mix of the AAV genome, AAV serotype plasmid (AAV.PHP.B) and the adeno helper plasmid pAd-DeltaF6 (Puresyn) using polyethylenimine max. At 48 h after transfection, the medium was replaced with fresh medium without FBS. The collected medium was mixed with 5× AAV precipitation buffer (400 g PEG 8000, 146.1 g NaCl in 1 l H$_2$O) and kept at 4 °C. Then, 1 day later, the cells were mechanically dislodged and centrifuged at 800$g$ for 15 min. The cell pellet was resuspended in 12 ml AAV lysis buffer (50 ml of 1 M TRIS-HCl (pH 8.5), 58.44 g NaCl, 5 ml of 2 M MgCl$_2$ in 1 l) and flash-frozen in liquid nitrogen. The supernatant was mixed with 5× AAV precipitation buffer, added to the medium collected previously, incubated for 2 h at 4 °C and centrifuged at 3,000$g$ for 1 h at 4 °C. The resulting pellet was resuspended in 3 ml AAV lysis buffer and added to the first cell pellet. The pellet was subjected to three freeze–thaw cycles and incubated with SAN (Merck) (50 U per 15 cm dish) for 1 h at 37 °C. After two centrifugation steps (saving the supernatant) at 3,000$g$ for 15 min at 4 °C, 14.5 ml of the supernatant containing AAV particles was poured into an ultracentrifuge-ready tube (Beckman Coulter). Iodixanol gradients were prepared by sequential pipetting of the following iodixanol solutions: 9 ml (15%), 6 ml (25%), 5 ml (40%) and 5 ml (54%). Gradients were ultracentrifuged using the Beckman type 70 Ti rotor at 63,000 rpm for 2 h at 4 °C. To recover the AAV particles, the tubes were pierced at the bottom, 4 ml of gradient (mainly 54% phase) were allowed to pass through and discarded, and the next 3.5 ml (containing isolated AAV) were retained. To remove the iodixanol and concentrate the AAV, the solution was diluted with PBS + 10% glycerol and centrifuged through a 15 ml Amicon 100 kDa MWCO filter unit (Amicon) at 1,000$g$ for 10 min. The dilution and centrifugation steps were repeated for three rounds. The resulting AAV solutions were aliquoted and flash-frozen in liquid nitrogen. The AAV particle concentration was determined by droplet digital PCR (ddPCR; BioRad). In brief, 5 μl of isolated AAVs were diluted 10× in water and treated with DNase I (NEB, M0303S) before preparing tenfold serial dilutions with ddPCR dilution buffer (Ultrapure Water with 2 ng μl$^{-1}$ sheared salmon sperm DNA (Thermo Fisher Scientific, AM9680) and 0.05% Pluronic F-68 (Thermo Fisher Scientific, 24040032)). ddPCR reactions with primers targeting the WPRE sequence (WPRE_fwd, CTTTCCCCC TCCCTATTG; WPRE_rev, CAACACCACGGAATTGTC; WPRE_probe, CACGGCGGAACTCATCG) were performed with 5.5 μl of the diluted AAV template, 11 μl ddPCR supermix for probes (BioRad, 1863024), 0.9 μM of both primers and 0.25 μM probe in a total of 22 μl. Droplets were generated using the BioRad ddPCR apparatus according to the manufacturer's indications. The amplification reaction was performed as follows: (1) 95 °C for 10 min; (2) 42 cycles of 95 °C for 30 s and 60 °C for 1 min (42 cycles); (3) 72 °C for 15 s; and finally (4) 98 °C for 10 min. Data were collected and analysed using the BioRad ddPCR apparatus to calculate the number of viral particles per μl.

**Mice.** All animal work was performed under the guidelines of the ETH Animal Welfare Office, the University Basel Veterinary

Office and the Basel-Stadt Cantonal Veterinary Office. Mice were kept under specific-pathogen-free conditions on a standard light cycle. Male Rosa26-LSL-Cas9 mice[12] (aged 6 to 8 weeks) were used unless otherwise indicated below. Male dCas9-KRAB mice (aged 6 to 8 weeks; JAX, 030000) were used. Male *LgDel*[+/+] and *LgDel*[+/−] mice[37] (aged 8 weeks) were used for the 22q11.2DS model snRNA-seq cell atlas. Animals were randomly selected for injection.

**Mouse injections. Triple-colour experiment.** We developed the triple-colour experiment to fine-tune the AAV injection conditions (Extended Data Fig. 1a). The three AAV genomes were individually packaged into the AAV.PHP.B capsid and purified as described in the 'AAV production and purification' section. Different viral particle doses (low, $2.5 \times 10^9$; medium, $5.0 \times 10^9$; and high, $2.5 \times 10^{10}$, total number of particles) were generated by pooling equal portions of the three viruses. The mice were spit into cages according to their experimental groups. After tail-vein injection of 100 µl of the AAV mixtures into LSL-Cas9 mice, the mice were kept for 4 weeks under standard conditions before tissue extraction and processing.

**Pooled screen.** AAV particles carrying gRNAs to target 22q11.2 locus genes were generated as described in the 'AAV production and purification' section. A single dose of $5.0 \times 10^9$ viral particles in a total volume of 100 µl was injected per mouse. The mice were kept for 4 weeks under normal conditions before brain tissue extraction and processing.

**Arrayed confirmation experiments and CRISPRi experiment.** Virus carrying gRNAs to target validation genes were individually prepared as described in the 'AAV production and purification' section. Animals were split into cages according to their experimental groups before tail-vein injection (100 µl) of $5.0 \times 10^9$ viral particles carrying unique gRNAs. The mice were kept for 4 weeks under standard conditions before tissue extraction and processing. Mice injected with *Ufd1l*-targeting gRNAs presented comorbidities 3 weeks after injection and had to be euthanized at that timepoint.

**Brain tissue collection for nucleus preparations.** Mice were intravenously injected with a lethal dose of pentobarbital (100 mg per kg body weight) before transcardial perfusion with 15 ml of ice-cold 1× PBS followed by 15 ml of ice-cold artificial cerebrospinal fluid (87 mM NaCl, 2.5 mM KCl, 1.25 mM NaH$_2$PO$_4$, 26 mM NaHCO$_3$, 75 mM sucrose, 20 mM glucose, 1 mM CaCl$_2$, 7 mM MgSO$_4$). The brain was removed, placed into a mouse brain matrix slicer (Zivic Instruments, BSMAS001-1), 1 mm slices were immediately snap-frozen and the region of interest was manually dissected into a frozen Eppendorf tube. Tissue samples were kept at −80 °C.

**Nucleus isolation.** Nucleus isolation was performed using mechanical and chemical tissue dissociation procedures. A tissue grinder (Sigma-Aldrich, D8938) was filled with 2 ml of ice-cold nucleus isolation buffer (NIB) (Sigma-Aldrich, NUC101-1KT) and frozen pieces of tissue were directly placed inside the grinder. For all of the experiments, nuclei from different animals were isolated in individual grinders, except for the 22q11.2 pooled screen, in which tissue of 15 animals was joined into three grinders to reduce the number of isolations and the waiting time before the subsequent procedures. The tissue was mechanically disrupted with 25 strokes with pestle A, followed by 25 strokes with pestle B. The homogenized solution was transferred to a protein low-binding tube (Eppendorf, 0030122216), mixed with an additional 2 ml of NIB, incubated for 5 min and immediately centrifuged at 500$g$ for 5 min at 4 °C. The supernatant was discarded and the pellet was resuspended in 4 ml NIB, incubated for 5 min and centrifuged at 500$g$ for 5 min at 4 °C. The pellet was resuspended in 4 ml of nucleus wash buffer (NWF; 1% BSA in 1× PBS, 50 U ml$^{-1}$ Superase-in RNA inhibitor (Thermo Fisher Scientific, AM2694) and 50 U ml$^{-1}$ Enzymatics RNA inhibitor (Enzymatics, Y9240L)) and centrifuged at 500$g$ for 5 min at 4 °C. Finally, the nucleus pellet was resuspended in 1 ml NWF and filtered through a 30 µm cell strainer (Sysmex) into a new protein low-binding tube.

**FANS analysis.** Fluorescence-activated nucleus sorting (FANS) was performed to (1) quantify infected nuclei in the triple-colour experiment; (2) purify nuclei from debris to ensure a clean nucleus solution before snRNA-seq library preparation; (3) isolate GFP$^+$ nuclei to prepare snRNA-seq libraries with nuclei from infected cells. In brief, isolated nucleus solutions were spiked with 2 µl per ml of Vybrant DyeCycle Ruby Stain (Thermo Fisher Scientific, V10273) and sorted using the MA900 apparatus (Sony). Singlet nuclei were gated on the basis of the DNA dye signal as illustrated in Extended Data Fig. 1f and GFP$^+$ nuclei were sorted into ice-cold NWF. Nuclei were centrifuged at 500$g$ for 7 min at 4 °C, resuspended in NWF and counted on the Spectrum Cell Counter (Cellometer) apparatus. Quantification of infected nuclei in the triple-colour experiment was performed using FlowJo (v.10.5.0).

**Fluorescence imaging.** For fluorescence imaging analyses, mice were euthanized by intravenous injection with a lethal dose of pentobarbital (100 mg per kg body weight), followed by perfusion with 15 ml of ice-cold 1× PBS and 15 ml of 4% PFA in 1× PBS. Brain tissue was incubated in 4% PFA in 1× PBS overnight and subsequently transferred in 1× PBS with 30% sucrose in which they were left until they sunk. The brains were then embedded in OCT and sections of a thickness of 20 µm were cut on a cryotome. Imaging was performed using the LSM 900 apparatus (Zeiss) with the experimentalist blinded to the sample of origin.

**Single-nucleus library preparation for gene expression and gRNA capture.** Nuclei infected with AAVs carrying the 3′ capture design (pAS006) were sequenced using the Chromium Single Cell 3′ Reagent Kit v3 (10x Genomics). The nuclei suspension was diluted to 1,000 nuclei per µl and processed according to the kit's protocol with 13 cycles of cDNA amplification and 14 cycles of sample indexing PCR. To amplify gRNA sequences from the RNA-polymerase-II-driven transcript, we performed a tri-step hemi-nested PCR reaction using the KAPA HiFi ReadyMix. To avoid overamplification, all PCRs were spiked in with EvaGreen (Biotium), monitored by qPCR and stopped before they reached saturation (exiting of the exponential phase). PCR 1 with primers targeting the U6 promotor sequence (TTTCCC ATGATTCCTTCATATTTGC) and read 1 sequence (ACACTCTTTCCC TACACGACG) was performed with 20 ng of full-length single-cell cDNA library as a DNA template. PCR 2 was performed with a forward primer targeting the U6 sequence immediately before the gRNA and containing the P7 adapter (GTCTCGTGGGCTCGGAGATGTGTATAAGAG ACAGcTTGTGGAAAGGACGAAACAC), a reverse P5 primer (AATGAT ACGGCGACCACCGAGATCTACACTCTTTCCCTACACGACG) and 2 µl of PCR 1 reaction as a template. Finally, a third PCR to index samples for deep sequencing used 2 µl of PCR 2 rection as template and was performed with a forward P7 index primer (CAAGCAGAAGACGGCATAC GAGATNNNNNNNNNGTCTCGTGGGCTCGG) and the P5 primer as reverse (same primer used in PCR 2). All primers were used at a final concentration of 0.3 µM. Amplification reactions were performed as follows: (1) 95 °C for 3 min; (2) 98 °C for 20 s, 65 °C for 15 s (72 °C for PCR 3), 72 °C 20 s (number of cycles up to qPCR saturation); and finally (3) 72 °C for 2 min. The final PCR reaction was cleaned and purified with double-size 0.6×–1.2× AMPURE bead selection (Beckman Coulter). Gene expression and gRNA libraries (5% of flow cell) were sequenced using the NextSeq 550 75 cycle kit with the following cycle distribution: 28 to read 1, 8 to index 1 and 56 to read 2.

Nuclei infected with the 5′ capture design (pAS088, preliminary experiments, pooled screen and confirmation experiments) were sequenced using the Chromium Single Cell 5′ Reagent Kit v1 (10x Genomics). To capture gRNA molecules, we altered the reverse transcription reaction to also include a gRNA-constant-region-targeting reverse transcription primer (0.15 µM, AAGCAGTGGTATCAACGCAG AGTACCAAGTTGATAACGGACTAGCC)[15,47]. After cDNA amplification (16 cycles), the reaction was purified with 0.6× SPRI beads (Beckman Coulter). At this point, longer cDNAs (more than 300 bp) from mRNA

molecules bind to the beads, whereas the shorter cDNAs (approximately 200 bp) from gRNA sequences are free in the supernatant. The preparation of gene expression libraries was performed as indicated by the kit's protocol, with 14 cycles of sample indexing PCR. To recover the gRNA–cDNA sequences, the supernatant from the above step was purified with 1.4× SPRI beads and eluted in 30 µl of ultrapure water (Thermo Fisher Scientific). A 1:10 diluted aliquot was loaded onto the Agilent Bioanalyzer High Sensitivity (Agilent) system to confirm the presence of a gRNA band of around 180 bp. The gRNA–cDNA library (30 ng) was processed for sample indexing PCR using the KAPA HiFi ReadyMix and 1 µM of P5 primer (AATGATACGGCGACCACCGAGATCT ACACTCTTTCCCTACACGACGCTC) and P7 indexing primer binding to the gRNA constant region directly downstream of the spacer sequence (CAAGCAGAAGACGGCATACGAGATNNNNNNNNNGTCTCGTGGGCTCGG AGATGTGTATAAGAGACAGTATTTCTAGCTCTAAAAC). Amplification reactions were performed as follows: (1) 95 °C for 3 min; (2) 15 cycles of 98 °C for 20 s, 54 °C for 30 s and 72 °C 20 s; and finally (3) 72 °C for 5 min. The final PCR reaction was cleaned and purified with double-size 0.6×–1.2× SPRI bead selection. Gene expression and gRNA libraries (5% of flow cell) were sequenced using the NextSeq 550 75 cycle kit or the NovaSeq 100 cycle kit with the following cycle distribution: 26 to read 1, 8 to index 1, and 56 (NextSeq) or 91 (NovaSeq) to read 2.

**Deep sequencing quantification of Cas9-induced indels.** To quantify the efficiency of Cas9 gene editing, 10,000 GFP-positive nuclei were sorted into quick extraction buffer (1 mM CaCl₂, 3 mM MgCl₂, 1 mM EDTA, 10 mM Tris pH 7.5; 1% Triton X-100 and 0.2 mg ml⁻¹ proteinase K freshly added before use) and subjected to DNA extraction at 65 °C for 10 min, 68 °C for 10 min and 98 °C for 10 min. Genomic DNA was used as the template for a first PCR reaction followed by a second PCR to index individual samples and attach P5 and P7 sequences. All of the reactions were performed using the KAPA HiFi Ready Mix. In brief, PCR 1 was performed to specifically amplify ~150 bp around the Cas9 cut site (keeping the cut site central in the amplicon) from genomic DNA (5 µl) with gene-specific primers containing adapters (0.5 µM, fwd, ACACTCTTTCCCTACACGACGCTCTTCCGATCT + forward gene specific sequence; rev, GTGACTGGAGTTCAGACGTGTGCTCTTCC GATC + reverse gene specific sequence). PCR 1 amplification was performed as follows: (1) 95 °C for 3 min; (2) 15 cycles of 98 °C for 20 s, primer set specific annealing temperature for 15 s, 72 °C 20 s; and finally (3) 72 °C for 2 min. A second PCR to index samples with P5 and P7 primers (0.25 µM, P5, AATGATACGGCGACCACCGAGATCTA CAC-NNNNNNNN-ACACTCTTTCCCTACACGACGCTCTTCCGATCT; P7, CAAGCAGAAGACGGCATACGAGAT-NNNNNNNN-GTGACTGGAGTTCAG ACGTGTGCTCTTCCGATC) was performed as follows: (1) 95 °C for 3 min; (2) 15 cycles of 98 °C for 20 s, 70 °C for 15 s, 72 °C 20 s; and finally (3) 72 °C for 2 min. Indexed samples were pooled, purified using the PCR Purification & Concentration Kit (Zymo Research, D4013) and loaded onto a 2% E-Gel (Thermo Fisher Scientific, G402022). The PCR product (~250 bp) was extracted from the agarose gel using the QIAquick Gel Extraction Kit (Qiagen, 28706×4) and sequenced using the NextSeq 550 150 cycle kit with the following cycle distribution: 150 to read 1, 8 to index 1 and 8 to index 2.

## Data processing

**Single-nucleus data processing.** Raw reads of snRNA-seq gene expression libraries were initially analysed with CellRanger v.4.0 (10x Genomics) using a mouse reference genome (Ensembl mouse GRCm38) to generate UMI count matrixes and to join data resulting from different library preparation lanes. Normalization and cell-type clustering were performed using the Seurat v.3.0 package in R[18]. In brief, UMI counts were scaled to 10,000 molecules per nucleus and log-normalized using the NormalizeData function, followed by selection of the top 2,000 variable genes with FindVariableFeatures. The normalized expression of the top genes was scaled and standardized across all nuclei (z-score transformation) with the function ScaleData and used for dimensional reduction with principal component analysis (PCA) implemented in RunPCA. We used the first 10 principal components (PCs) to cluster nuclei on the basis of the expression of the top variable genes (FindNeighbors and FindClusters). Nuclei were projected to two dimensions with UMAP embedding (RunUMAP and DimPlot) and coloured by cluster to evaluate clustering performance. To assign clusters to specific cell types, we first identified gene markers (FindAllMarkers) for each cluster and investigated their expression in the brain cell reference dataset[16] (https://DropViz.org).

**gRNA assignment to individual nucleus.** To assign gRNAs identities to nuclei, we first analysed raw deep sequencing reads (from gRNA-specific enrichment libraries) with CellRanger. Although these libraries do not align to the mouse reference genome, CellRanger outputs a BAM file containing reads tagged with corrected cell barcodes and UMIs. Cell barcode correction is important to increase the alignment between barcodes found in both gene and gRNA expression datasets. Available scripts were used to extract gRNA count tables (from the correct BAM files) containing information about cell barcode, gRNA sequence, UMI counts and read counts (https://github.com/shendure-lab/single-cell-ko-screens.git)[48]. The gRNA sequences were aligned to a reference list using BOWTIE 2 v.2.3.5, permitting an editing distance of maximum 2 bp (ref. 49). For each nucleus, we removed all gRNAs with coverage (READ_counts/UMI_counts) less than 60 (coverage should be calculated for each new deep sequencing run as it depends on the total number of reads attributed to the library), only 1 captured molecule or when the gRNA-UMI counts represented less than 10% of all gRNAs identified in the nucleus. The gRNA counts, as well as the number of gRNAs detected in each nucleus and their identity, were appended to the nuclei metadata in the Seurat object.

**Differential expression analysis.** Recent studies have highlighted that pseudobulk analysis of single-cell gene expression data better recapitulates true differences between conditions. Thus, we applied a pseudobulk profile and bulk RNA-seq statistical method to calculate the LFC and FDR. Pseudobulk profiles in pooled screen datasets were generated as follows: for each cell type and perturbation, we summed raw UMI counts across single-nucleus library lanes (nine lanes in total). This step transforms the data from a matrix where each column represents one nucleus to a matrix where each column is the sum of all nuclei from the same lane that contains a given perturbation. For LgDel samples (*LgDel^{+/−}* and *LgDel^{+/+}*), pseudobulk profiles were generated by aggregating raw UMI counts of nuclei from the same sample (that is, same animal) and cell type. Differential gene expression of pseudobulk profiles was performed with the R package edgeR (v.3.36.0)[19]. For each cell type, we used edgeR with the likelihood ratio test (egdeR-LRT) to calculate the LFC and FDR values for each perturbation against the SH control. The same process was used to compare *LgDel^{+/−}* against *LgDel^{+/+}* samples. Differential expression analysis with a scRNA-seq method was performed using the Seurat function FindMarkers (parameter test.use = "LR") to compare each perturbation against the SH control for each cell type individually. For all analyses based on LFC values, we focused on genes with average expression higher than 0.25 UMI per nucleus in the control group (SH control or *LgDel^{+/+}*), which typically resulted in a list of ~5,000 genes.

**Perturbation and nucleus filtering.** To identify perturbations leading to a strong transcriptional signature, we proceeded as follows (for each cell type C and perturbation P):

1. Considering all nuclei belonging to C and P, we calculated pseudobulk differential expression against SH control nuclei from cell type C. If the number of detected DEGs was less than 5 (FDR < 0.05), we assumed that P did not lead to a strong transcriptional phenotype in cell type C and the perturbation was considered to be non-relevant.

2. As CRISPR–Cas9-induced mutations are typically not observed in all cells carrying a gRNA, we focused on implementing a filtering step. Here the goal is to identify and remove nuclei with a transcriptional signature closer to control than to perturbed nuclei. For all perturbations identified as significant (DEG > 5, FDR < 0.05), we applied LDA using the lda function in the R package MASS (v.7.3-50)[50]. For each $P$, we trained a LDA model with a single-nucleus matrix containing nuclei belonging to SH control and $P$ as observations and $P$-specific DEG as variables, and used the model to predict nucleus labels (SH control or $P$). We removed all nuclei belonging to a perturbation group of which the predicted label did not agree with the true experimental label and kept all SH control nuclei.

**Hotelling's $T^2$ statistic.** To orthogonally identify perturbations leading to strong transcriptional phenotypes and confirm the results from DEG-based filtering, we use Hotelling's $T^2$ statistics (multivariate $t$(Extended Data Fig. 3g)[21,51]. In brief, for each cell type, we performed dimensional reduction with PCA to reduce the multivariate space from ~5,000 genes to 20 principal components and performed a pairwise comparison of each perturbation to SH control nuclei.

**Nuclei UMAP embedding based on perturbation transcriptional signature.** For each cell type individually, we evaluated whether UMAP embedding was able to separate nuclei on the basis of their perturbation transcriptional phenotype (Fig. 2d and Extended Data Fig. 5d). First, DEGs (LFC > 0.5, FDR < 0.01) were selected from all perturbations. This process yielded a matrix with nuclei from all perturbations and the SH control as observations and DEGs from all perturbations as variables. Normalized UMI counts were centred and scaled with the Seurat function ScaleData and used for UMAP embedding with the R package UWOT (v.0.1.8)[52]. The following parameters were used: metric = "cosine"; n_neighbors = 10; min_dist = 5; and spread = 10.

**Augur scoring analysis.** The Augur (v.1.0.0) R package[53] was created to identify cell types that exhibit a high degree of transcriptional changes when comparing control and perturbed cells (Extended Data Fig. 3i). The same rationale can be applied to identify perturbations that lead to a transcriptional phenotype. In brief, for each cell type, we use the function calculate_auc() with the recommended parameters to calculate augur scores for each perturbation. We focused on genes with average expression higher than 0.25 UMI per nucleus in the control group and used the entire group of nuclei for each perturbation and cell type combination.

**Identifying perturbation-specific transcriptional phenotypes.** For all perturbations with a strong transcriptional phenotype, we performed pseudobulk differential expression using all nuclei passing LDA filtering against the SH control group. This step was repeated for each cell type individually. The top 20 upregulated genes (LFC > 0.5 and FDR < 0.01) from each perturbation were used for the heat map in Fig. 2c. To create genetic programs relevant for each perturbation, we selected all genes with an absolute LFC above 0.5 and FDR < 0.01 and spit them into two programs: upregulated genes (LFC > 0.5) and downregulated genes (LFC < −0.5). To reveal biological processes associated with dysregulated genes, the entire list of genes served as input for functional enrichment analysis with the R package g:Profiler (v.0.2.0)[54] functions g:GOSt and g:SCS. A multiple-hypothesis testing correction method applying a significance threshold of 0.05 was used. The top biological processes (GO:BP) for each gene program were selected as representative terms in Fig. 3c.

**Computational dissection of zygosity in perturbed nuclei.** We applied the R package destiny (v.3.17)[55] to align nuclei along a pseudotemporal space with diffusion maps[56]. In brief, for each perturbation, we extracted gene expression data (from the arrayed perturbations

experiment) from SH control and perturbed nuclei, used it as an input to the function DiffusionMaps(), and extracted the first two diffusion components (DC) for plotting (Extended Data Fig. 6a–c). To calculate artificial zygosity labels shown in Extended Data Fig. 6g–i, we performed $k$-means clustering of DC1 with $k$ = 3. Differential expression was performed as indicated in the 'Differential expression analysis' section.

**Gene program scores.** Gene programs (Fig. 3d–f and Extended Data Fig. 8b–d) were identified as indicated in the 'Identifying perturbation-specific transcriptional phenotypes' section. To calculate scores (that is, the average expression of all genes belonging to a program across nuclei), we first normalized and centre-scaled raw UMI counts for all nuclei. Then, for each nucleus, we averaged the expression of genes in the program and divided nuclei by perturbation before visual representation with ridge plots.

**Pearson correlation analysis.** Pearson correlation between individual perturbations and cell types (Extended Data Figs. 3j and 5e) was calculated using the LFC values of all genes differentially expressed in at least one condition (abs(LFC) > 0.5) and FDR < 0.01 as variables. To calculate correlations between screen and array experiments (Fig. 3b), we selected all genes that were differentially expressed (abs(LFC) > 0.5) and FDR < 0.01) in at least one condition and experimental group.

**Indel analysis.** Deep-sequencing libraries for indel analysis were generated as described in the 'Deep sequencing quantification of Cas9-induced indels' section and analysed using CRISPresso2 (v.2.0.20)[57] with the following parameters: -r1 "fastq file name"; -a "amplicon sequence"; -c "amplicon sequence"; -g "gRNA sequence"; --default_min_aln_score 60; --plot_window_size 20; --min_bp_quality_or_N 0; --exclude_bp_from_left 15; --exclude_bp_from_right 15; -w 1; and -wc −3.

**Disease, gene set and mRNA-target enrichment analysis.** Disease enrichment analysis (Extended Data Fig. 11e) was performed with the list of genes commonly dysregulated in individual perturbations and the deletion model using the R package Enrichr (v.2.1)[58] and the DisGeNET database[42]. To investigate biological processes associated with LgDel transcriptional signatures (Fig. 4d and Extended Data Fig. 10g), the list of expressed genes ranked by higher to lower LFC value was used as input to the R package fgsea (v.3.17)[59] to run GSEA using the mouse GO:BP dataset. This step was repeated for each cell type individually. To study miRNA-target enrichment (Extended Data Fig. 9d), the top 1,000 upregulated genes from each perturbation and cell type were uploaded as input to the online tool MIENTURNET[29] using the miRTarBase (http://userver.bio.uniroma1.it/apps/mienturnet/) reference dataset[60].

**Robust regression model.** The use of robust regression to model the LgDel transcription profile using individual perturbations follows the assumption that the LgDel expression profile is a combination of each individual perturbation[40] (LgDel = $c_{Dgcr8}$Dgcr8 + $c_{Dgcr14}$Dgcr14 + $c_{Gnb1l}$Gnb1l). The expression profile of each condition (that is, LgDel, Dgcr8, Dgcr14 and Gnb1l) is the change induced by the deletion or each perturbation (all nuclei from a given perturbation) relative to WT control nuclei (LgDel) or SH control nuclei (screen). Our pseudobulk differential expression analysis approach calculates, for each expressed gene, LFC values that quantify the difference between a given group and the control condition. Thus, differential expression analysis results in a vector of LFC values that can be directly used in the model. To fit the model, we used the R package MASS v.7.3-50 function rlm and focused on genes with average expression higher than 0.25 UMI per nucleus in the control group (SH control or $LgDel^{+/+}$). We used distance correlation (dcor) with the R package energy to evaluate the model fit [$d$ = dcor(LgDel, [$c_{Dgcr8}$Dgcr8 + $c_{Dgcr14}$Dgcr14 + $c_{Gnb1l}$Gnb1l])].

## Reporting summary

Further information on research design is available in the Nature Portfolio Reporting Summary linked to this article.

## Data availability

Raw and processed sequencing data generated for this study are available at the Gene Expression Omnibus (GSE236519).

## Code availability

Calculations were performed at the sciCORE (http://scicore.unibas.ch/) scientific computing centre at the University of Basel. Custom made scripts are available at the Platt laboratory GitHub (https://github.com/plattlab/AAV-Perturb-seq).

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

**Acknowledgements** We thank C. Beisel, E. Burcklen, I. Nissen, K. Eschbach, M. Feldkamp and T. Schär from the Genomics Facility Basel for assistance in Illumina sequencing; V. Jäggin, M. Tacchio and A. Gumienny from the Single Cell Facility Basel for support in flow cytometry; M. Hussherr from the UniBasel Animal Facility for support with in vivo experiments; all of the members of the Platt and Jabaudon laboratories for discussions. The authors are supported by the Swiss National Science Foundation grant 31003A_175830 (to M.K., S.L. and R.J.P.); ETH Zurich grant ETH-27 18-2 (to M.K. and R.J.P.); Botnar Research Centre for Child Health Multi-Investigator Project Grant (to R.F. and R.J.P.); National Centres of Competence—Molecular Systems Engineering 51NF40-182895 (to M.K., S.L., G.K., R.F. and R.J.P.); and Brain & Behavior Research Foundation grant 26606, Personalized Health and Related Technologies grants 2021-542, PHRT-203 and 2017-510 (to U.L.); and Swiss Cancer Research Foundation grant KFS-4863-08-2019, European Research Council grant 851021, Botnar Research Centre for Child Health Fast Track Call grant, EMBO grant 4217 and the Fickel Family Fund (to R.J.P.). The Jabaudon laboratory (E.K. and D.J.) is supported by the Swiss National Science Foundation, the European Research Council, the Carigest foundation and the Société Académique de Genève.

**Author contributions** A.J.S. and R.J.P. conceived and designed the experiments. E.K. curated the 22q11.2-locus gene list and performed LgDel model experiments. A.J.S. and M.K. performed animal injections and tissue collection. A.J.S. and U.L. prepared snRNA-seq libraries. A.J.S. and S.L. performed FANS. G.K. performed imaging experiments. A.J.S. and R.F. wrote code to analyse and visualize data. A.J.S. and R.J.P. wrote the manuscript with contributions from all of the authors.

**Funding** Open access funding provided by Swiss Federal Institute of Technology Zurich.

**Competing interests** A.J.S. and R.J.P. are listed as inventors on a patent application relating to work in this manuscript.

**Additional information**
**Correspondence and requests for materials** should be addressed to Randall J. Platt.

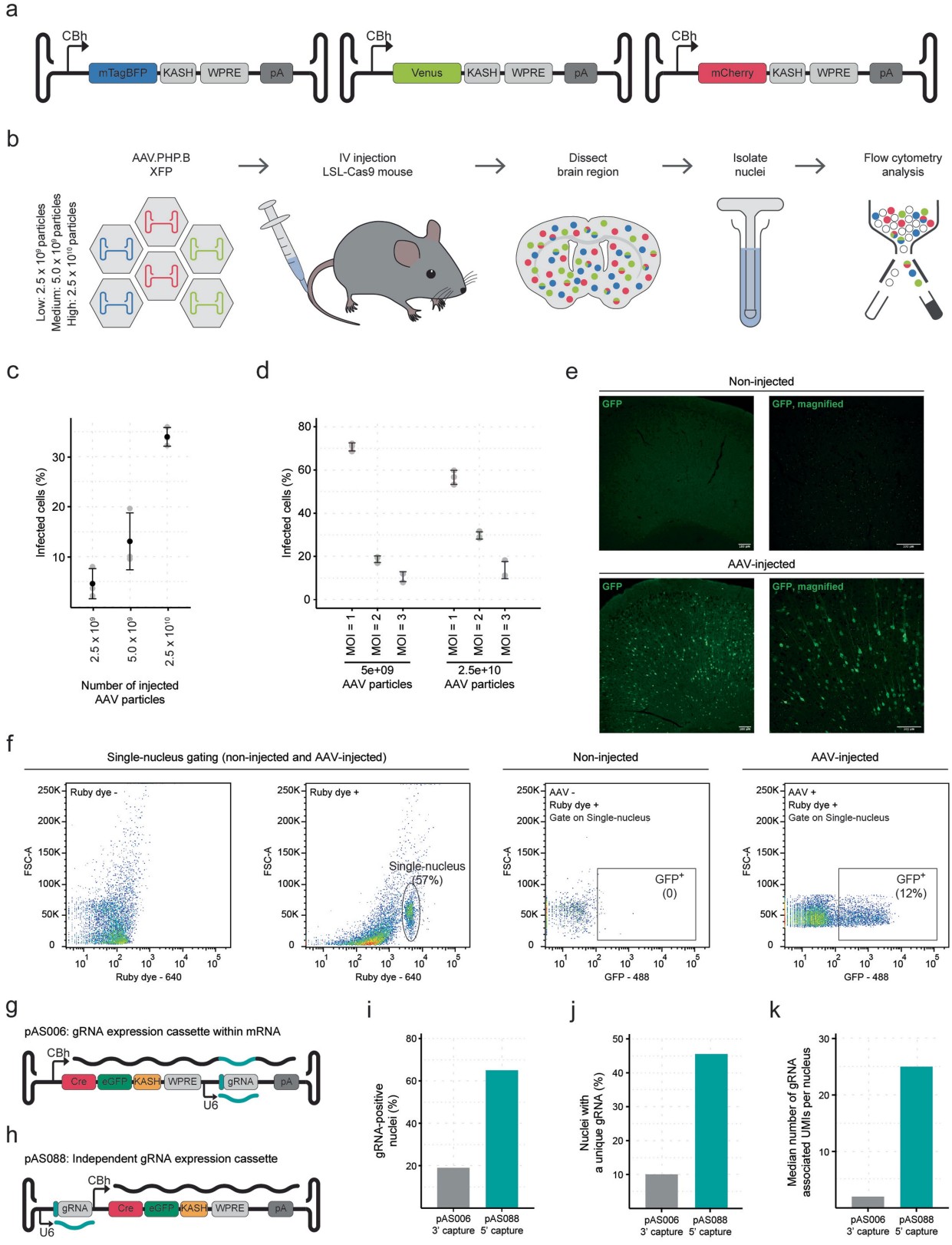

**Extended Data Fig. 1** | See next page for caption.

**Extended Data Fig. 1 | Establishing a method to capture both mRNA and CRISPR gRNA information from the same AAV-infected nucleus.**
**a**. Schematic representation of AAV genomes used to deliver and express mTagBFP, Venus, or mCherry under the control of the CBh promoter. **b**. Schematic representation of the triple colour experiment. An equal-ratio mix of the three AAVs was injected in LSL-Cas9 animals with different doses (Low: $2.5 \times 10^9$; Medium: $5.0 \times 10^9$; High: $2.5 \times 10^{10}$, total AAV particles). **c**. Percentage of infected nuclei (*i.e.*, nuclei expressing at least one fluorescent protein) after systemic injection of different viral doses, n = 3 biologically independent animals. Data are presented as mean values +/− SD. **d**. Percentage of infected nuclei expressing one, two, or the three FPs. Data shown for injections with $5.0 \times 10^9$ and $2.5 \times 10^{10}$ total AAV particles, n = 3 biologically independent animals. Data are presented as mean values +/− SD. **e**. Fluorescence imaging of brain cells expressing GFP four weeks after systemic injection of $5.0 \times 10^9$ AAV particles. Experiments were repeated in n = 3 mice. Scale Bar is 100 μm. **f**. Flow cytometry gating strategy to sort GFP-positive nuclei. **g-h**. Schematic representation of the AAV genome engineered for 3' capture (pAS006) (**g**) and 5' capture (pAS088) (**h**) of gRNA molecules. **i**. Percentage of nuclei with detected gRNAs (*i.e.*, at least one gRNA molecule recovered) for 3' or 5' capture methods. **j**. Percentage of nuclei with a unique gRNA (*i.e.*, all gRNA molecules found have the same sequence) for 3' or 5' capture methods. **k**. Median number of UMIs associated with gRNAs in each nucleus for 3' or 5' capture methods.

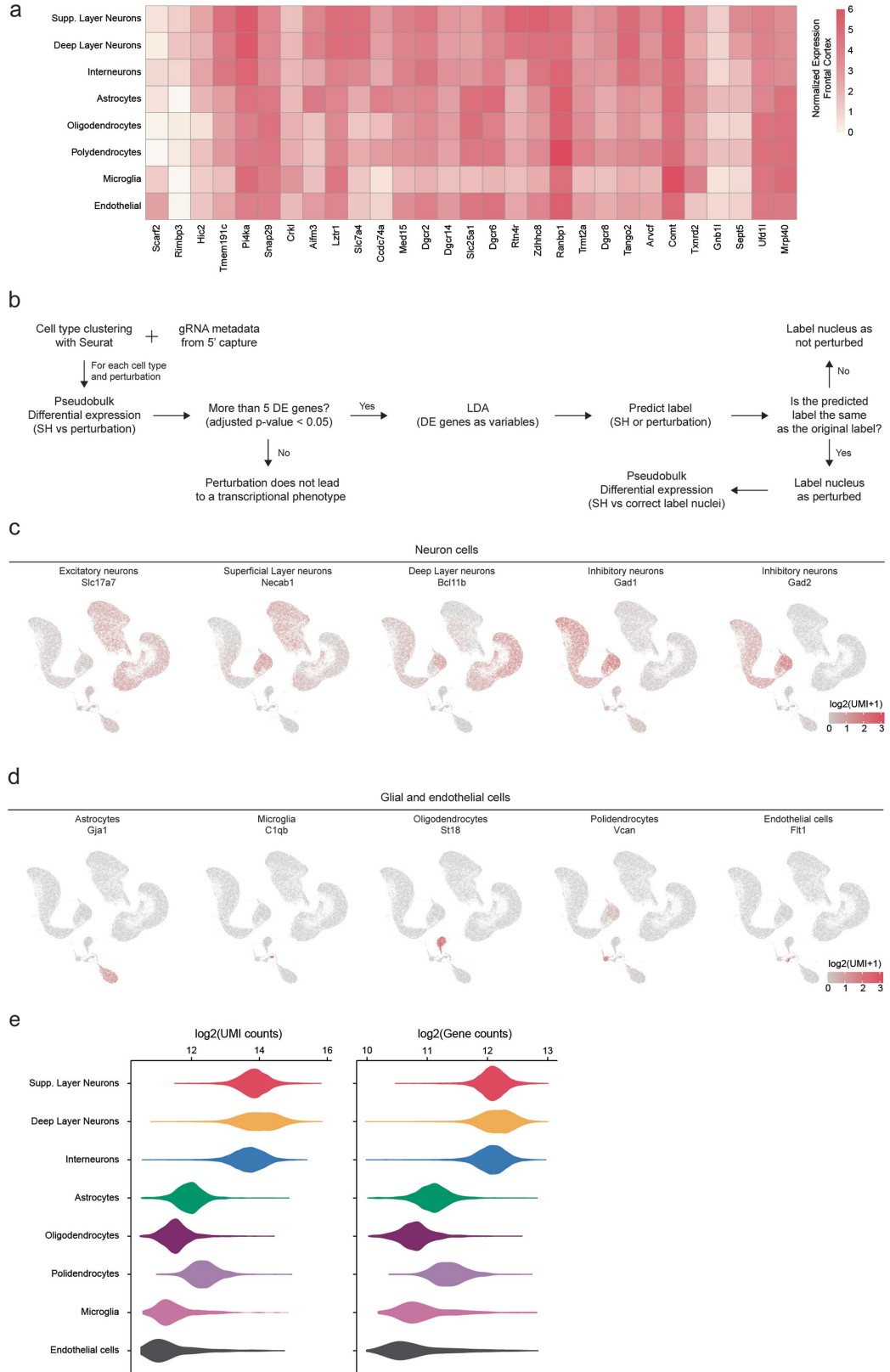

**Extended Data Fig. 2 | AAV-Perturb-seq of 22q11.2DS genes yields a rich single-nucleus dataset spanning genes and brain cell types from adult mice. a.** Normalized expression of 22q11.2 locus target genes (columns) in brain prefrontal cortex cell types (rows). Data from DropViz mouse brain cell atlas project. **b.** Data analysis workflow and filtering strategies to reveal perturbation-associated transcriptional phenotypes. **c**–**d.** UMAP representation with normalized expression of neuron type (**c**) or non-neuronal brain cell (**d**) marker genes. **e.** Average UMI and gene counts for each cell type in the screen dataset.

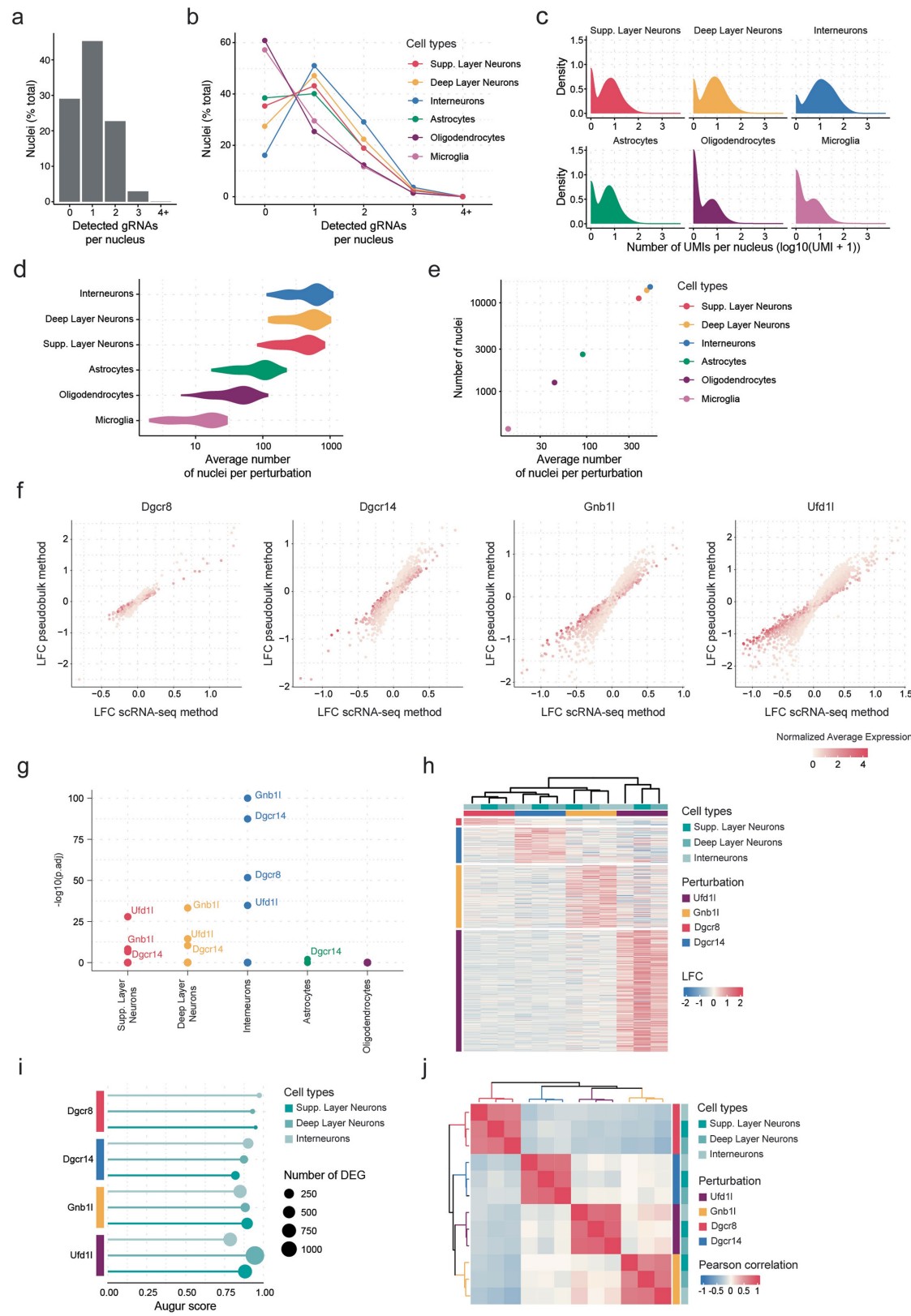

**Extended Data Fig. 3 | Identification of perturbed nuclei and transcriptional phenotypes. a**. Percentage of gRNAs detected per nucleus for all cell types combined. **b**. Percentage of gRNAs detected per nucleus for each cell type individually. **c**. Number of total UMI counts per nucleus across cell types. **d**. Average number of nuclei per perturbation across cell types. **e**. Correlation between the number of nuclei in a given cell type cluster and the average number of nuclei per perturbation. **f**. Correlation between LFC values calculated with scRNA-seq or pseudobulk methods for each target gene. The colour gradient indicates average gene expression. **g**. Hoteling T-squared statistics of transcriptional phenotypes induced by 22q11.2 gene perturbations in different cell types. **h**. Heatmap with all up-regulated differentially expressed (LFC > 0.5 and FDR < 0.01) genes (rows) for each cell type and perturbation (columns). **i**. AUGUR score for each cell type and perturbation. Bubble size indicates number of DEGs (LFC > 0.5; FDR < 0.01). **j**. Pearson correlation of transcriptional profiles induced by different perturbations across neuron types.

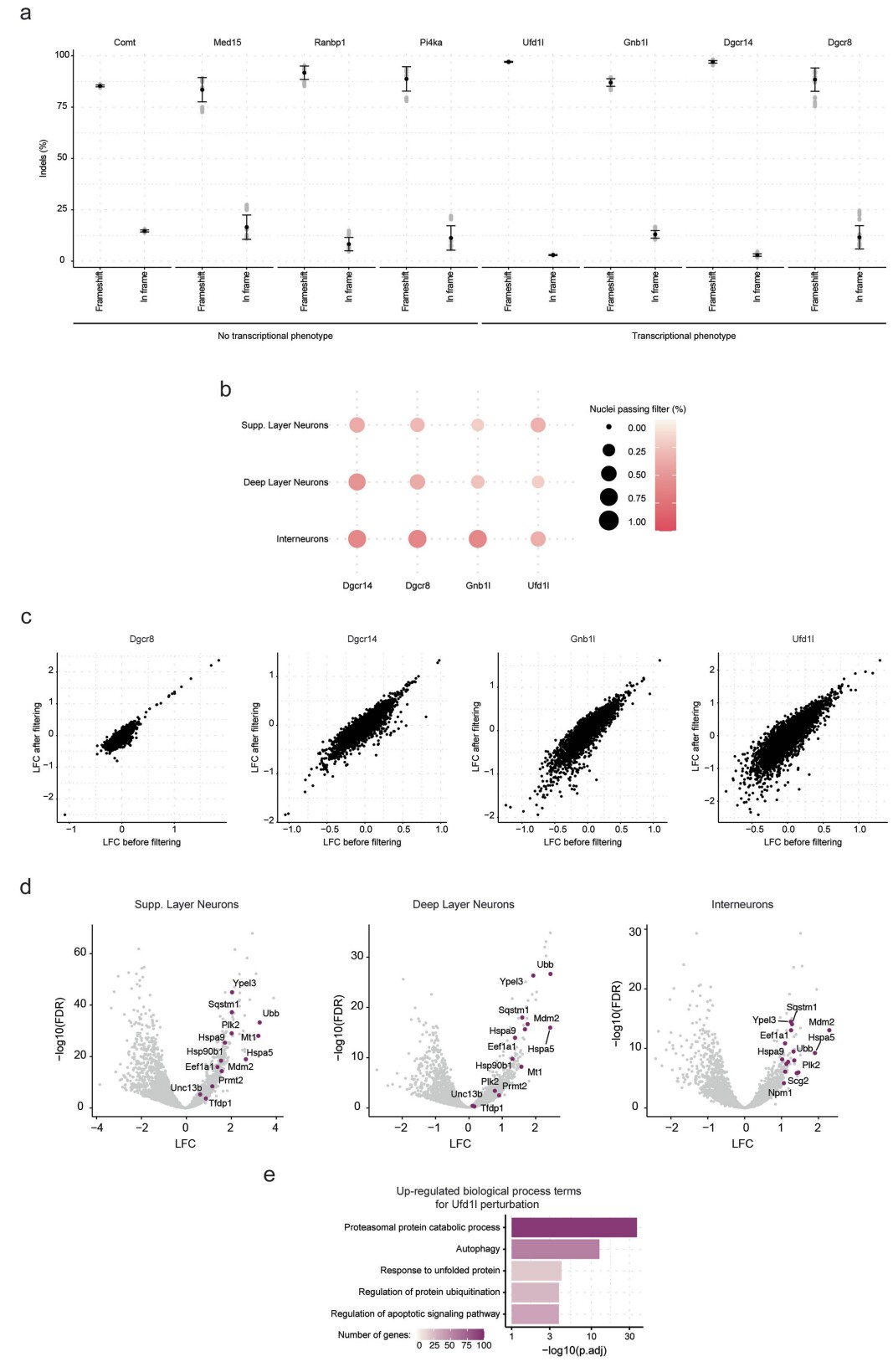

**Extended Data Fig. 4 | Testing the robustness of AAV-Perturb-seq.**
**a**. Percentage of frameshift and in-frame Cas9-induced indels mediated by four gRNAs targeting genes that either induce a strong transcriptional phenotype (*Ufd1l, Gnb1l, Dgcr14,* and *Dgcr8*) or do not induce a strong transcriptional phenotype (*Comt, Med15, Ranbp1,* and *Pi4ka*), n = 3 biologically independent animals. Data are presented as mean values +/– SD. **b**. Percentage of nuclei passing LDA filtering for perturbations across neuron types. **c**. Correlation between LFC values calculated with all gRNA-containing nuclei before filtering and nuclei after passing LDA filtering. **d**. Volcano plots with LFC values for *Ufd1l*-perturbed nuclei across neuron types. Purple colour indicates up-regulated DEGs related with cell death processes. P-values were calculated using edgeR-LRT with FDR multiple comparison test correction **e**. Biological processes (GO:BP) associated with up-regulated genes in *Ufd1l*-perturbed nuclei. P-values were adjusted with Bonferroni's multiple comparison test.

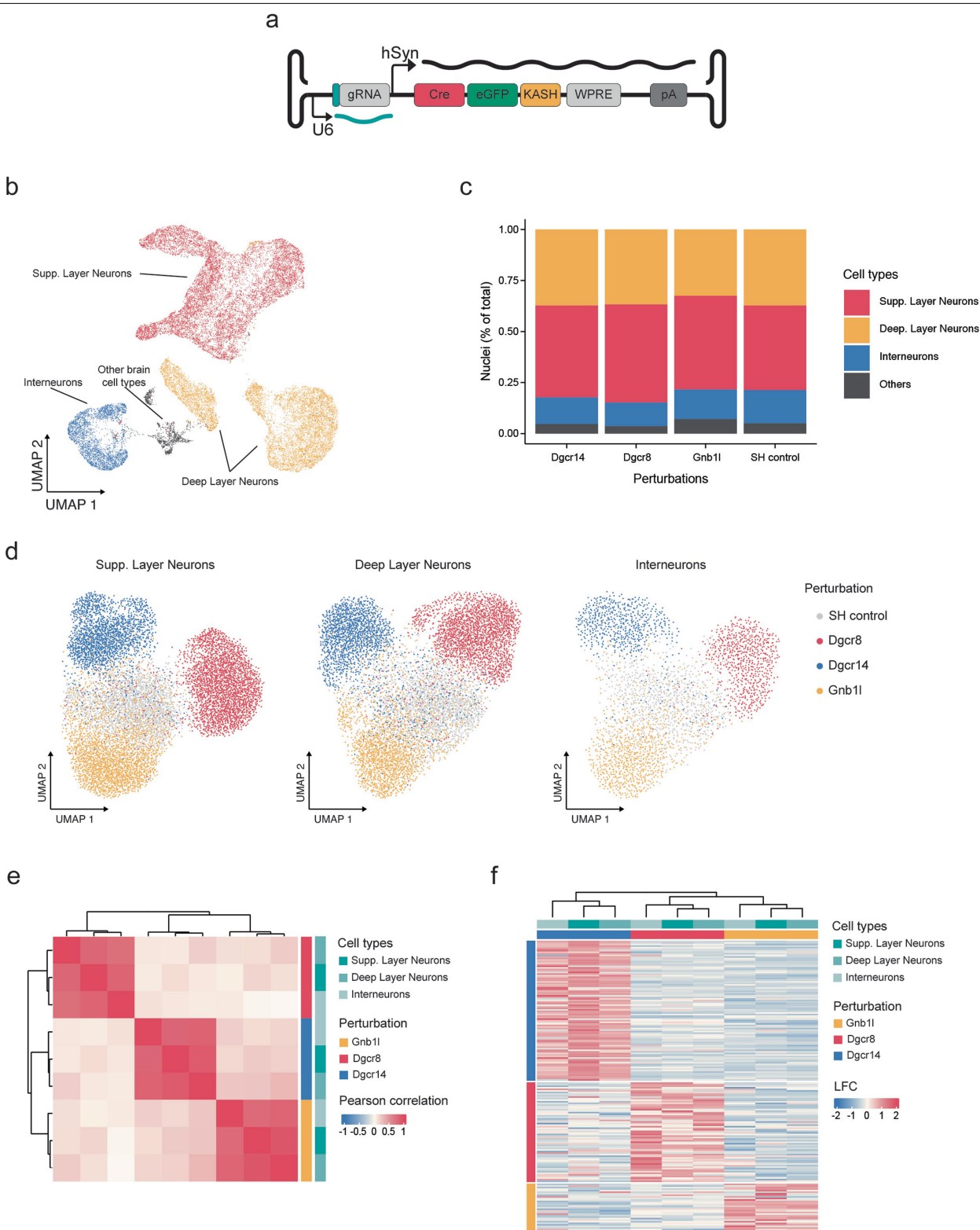

**Extended Data Fig. 5 | Arrayed perturbation experiments confirm the pooled screen results. a.** Schematic of AAV genome used in arrayed experiments. **b.** UMAP embedding of prefrontal cortex nuclei from arrayed experiments using gRNAs targeting SH control, *Dgcr8*, *Dgcr14*, and *Gn1bl*. **c.** Percentage of nuclei from individual perturbations for each cell type. **d.** UMAP embedding of SH control nuclei and nuclei passing filter perturbed in *Dgcr8*, *Dgcr14*, or *Gnb1l*, for each neuron type individually in arrayed experiments using DEGs identified in the pooled dataset as variables. **e.** Pearson correlation of transcriptional profiles induced by different perturbations across neuron types in arrayed experiments. **f.** Heatmap with all up-regulated differentially expressed (LFC > 0.5 and FDR < 0.01) genes for each cell type and perturbation (columns) in arrayed experiments.

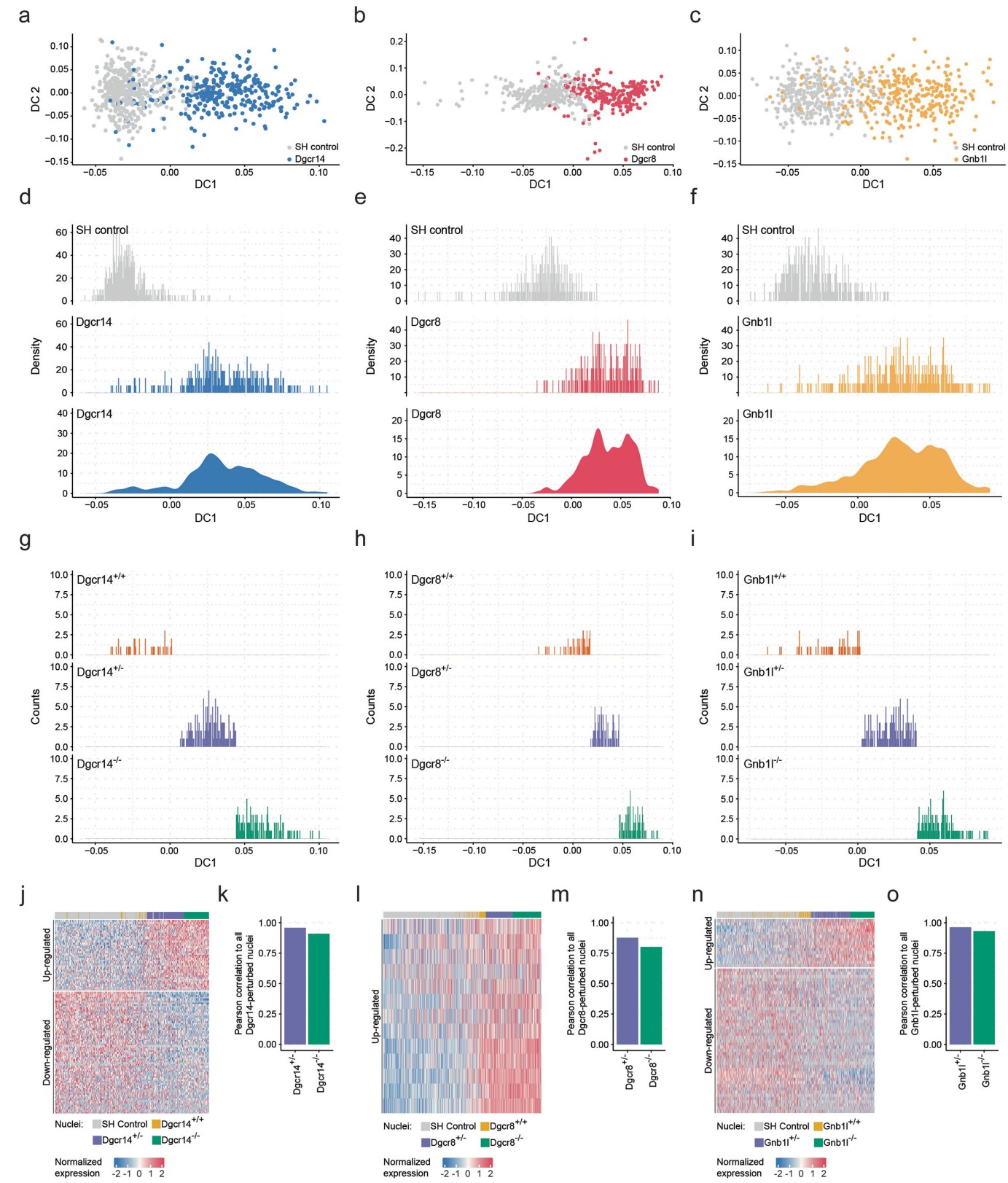

**Extended Data Fig. 6 | Computational stratification of zygosity cell states in perturbed nuclei. a–c**. Diffusion map-based stratification of SH control and perturbed nuclei. **d–f**. Histogram and density plots highlighting the position of SH control and nuclei across the first diffusion component (DC). **g–i**. Stratification of perturbed nuclei into three zygosity states. **j–o**. Expression of DEGs calculated with all perturbed nuclei together (**j, l**, and **n**) and correlation of LFC between zygosity states and all nuclei (**k, m**, and **o**).

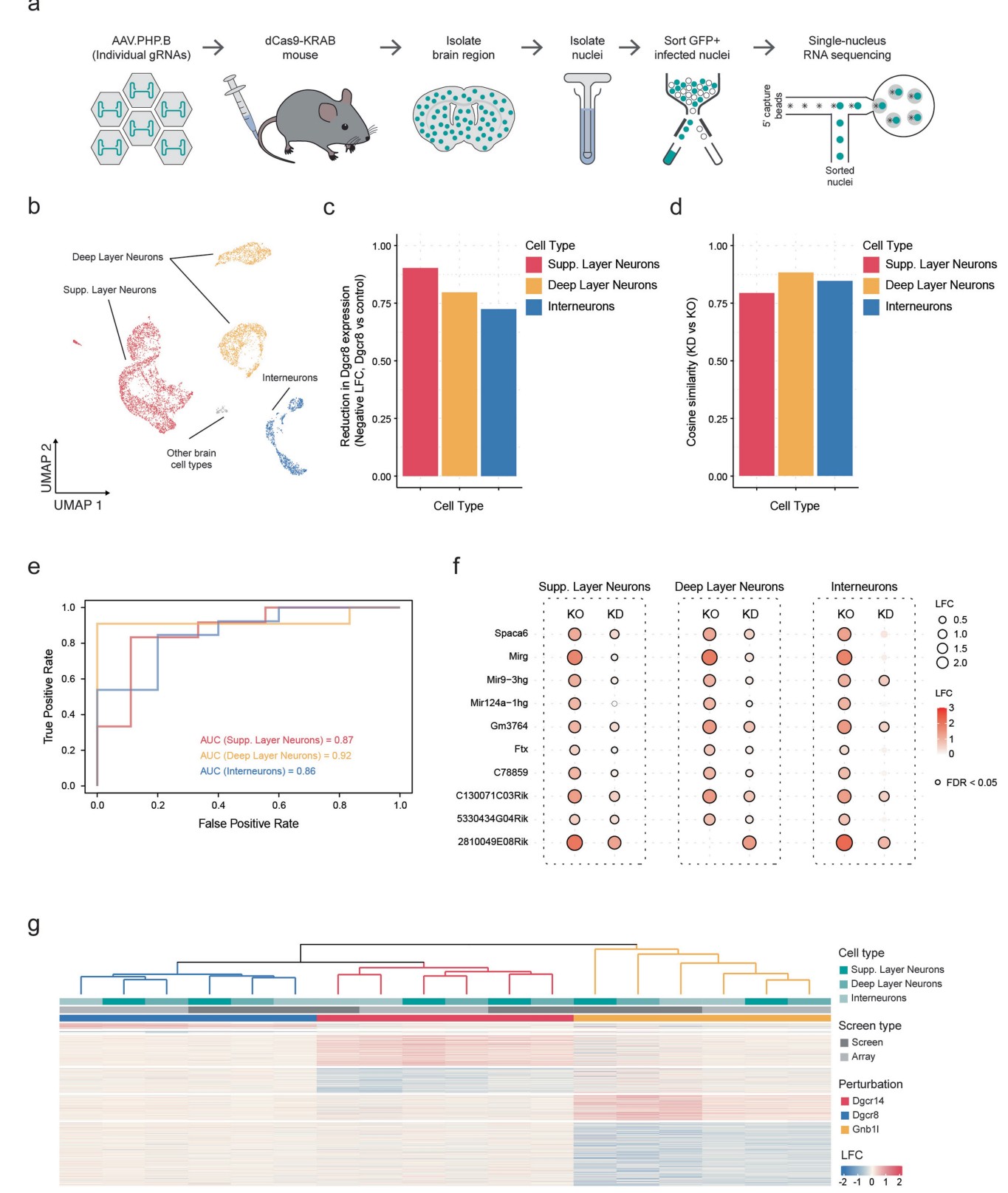

**Extended Data Fig. 7 | Modelling haploinsufficiency with CRISPRi.**
**a**. Schematic representation of arrayed CRISPRi experiments. **b**. UMAP embedding of ~8,000 AAV.PHP.B-infected nuclei isolated from the dCas9-KRAB mouse prefrontal cortex. **c**. LFC values of the *Dgcr8* mRNA between *Dgcr8* perturbation and control nuclei across cell types. **d**. Cosine similarity of gene expression values between *Dgcr8* perturbation and control nuclei across cell types. **e**. Area under the curve (AUC) showing the reliability of differential expression identification (DE) in knock-out (KO, screen profiles)

and knock-down (KD, CRISPRi) experiments. False positive and true positive rates comparison of snRNA-seq profiles of *Dgcr8* KO and KD. True positives are DE genes (LFC > 0.4 and FDR < 0.05) identified in the KO experiment. False positives are genes considered DE in the KO dataset, but not in the KD. **f**. LFC values (for both KO and KD experiments) of DEGs identified in the screen KO dataset across cell types. P-values were calculated with edgeR-LRT with FDR multiple comparison test correction. **g**. Hierarchical clustering of LFC profiles from perturbations across cell types and experiments (screen or arrayed).

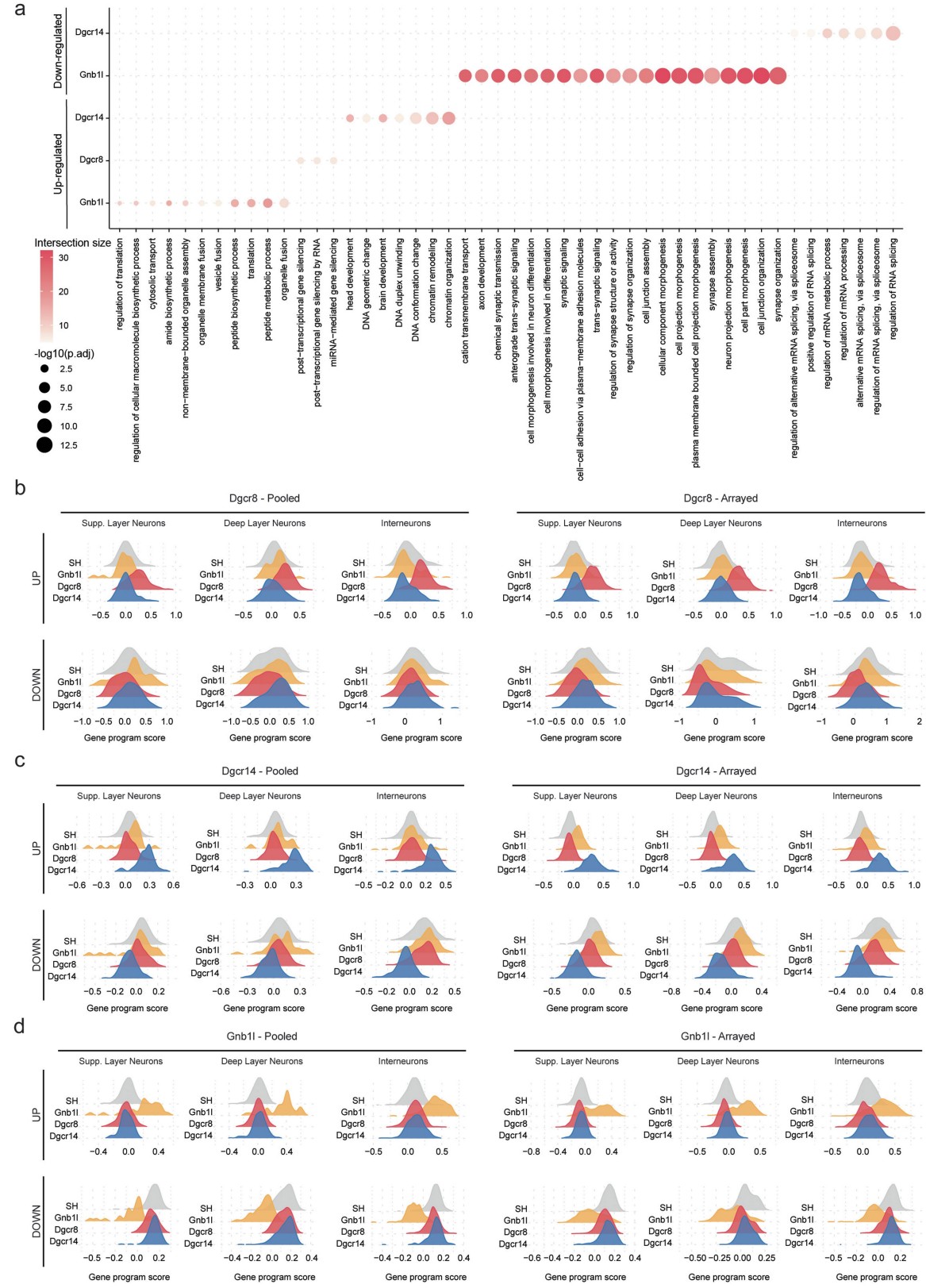

**Extended Data Fig. 8 | Gene program scores show perturbation-specific signatures across cell types and experiments. a**. Bubble plot showing biological processes (GO:BP) altered by perturbing *Dgcr8*, *Dgcr14*, or *Gnb1l*, for both up and down-regulated gene programs. P-values were adjusted with Bonferroni's multiple comparison test. **b–d**. Gene program scores for up-regulated (UP) and down-regulated (DOWN) genes in *Dgcr8*, *Dgcr14*, and *Gnb1l* perturbed neuron types across experiments (screen and array).

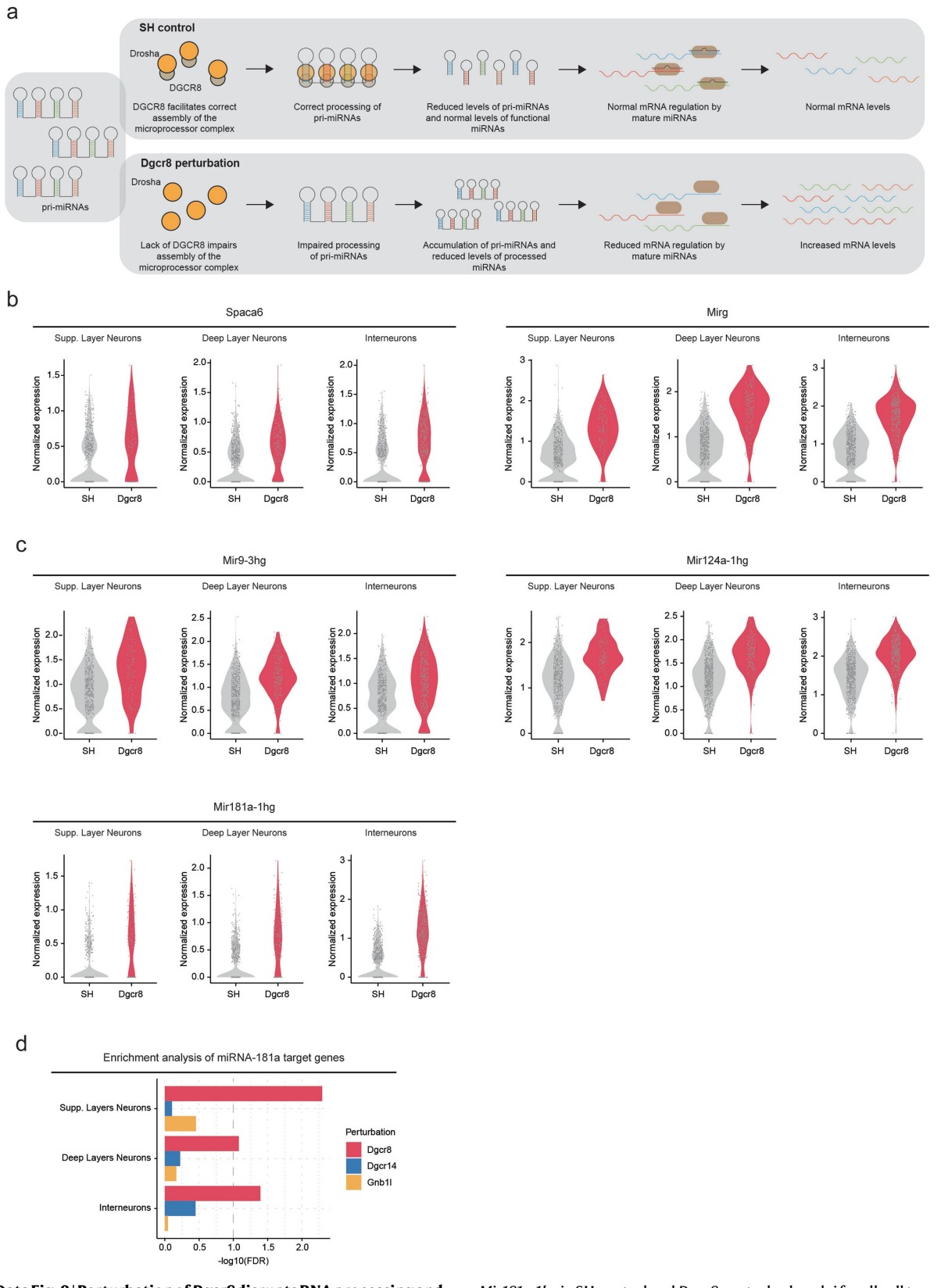

**Extended Data Fig. 9 | Perturbation of *Dgcr8* disrupts RNA processing and leads to the accumulation of pri-miRNAs. a**. Schematic representation of pri-miRNA processing mediated in SH control and Dgcr8-perturbed cells. **b**. Normalized expression of *Spaca6* and *Mirg* in SH control and *Dgcr8*-perturbed nuclei for all cell types. **c**. Normalized expression of *Mir9-3hg, Mir124a-1hg,* and *Mir181a-1hg* in SH control and *Dgcr8*-perturbed nuclei for all cell types. **d**. miRNA target enrichment analysis for *miRNA-181a* targets in up-regulated genes across perturbations and cell types. Dashed line indicates FDR = 0.1. P-values were corrected with FDR multiple comparison test.

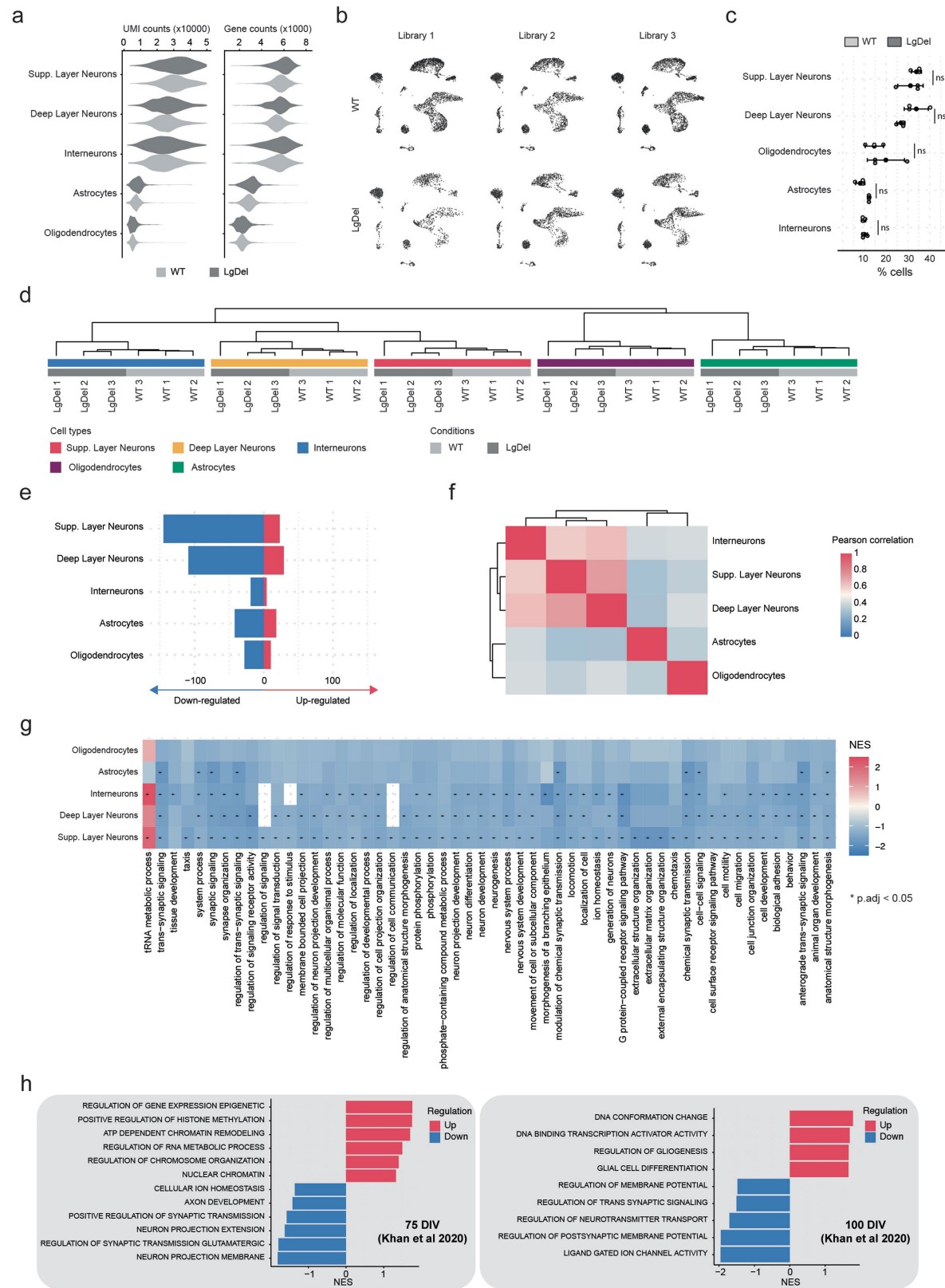

**Extended Data Fig. 10 | Single-nucleus prefrontal cortex atlas of the LgDel 22q11.2DS mouse model. a**. Average UMI and gene counts for each cell type in WT and LgDel snRNA-seq libraries. **b**. UMAP embeddings separated by individual samples from WT and LgDel snRNA-seq libraries. **c**. Cell type frequency in WT and LgDel samples. Data are presented as mean values +/− SD (n = 3 biologically independent animals, Two-sided Student's t-test, FDR < 0.05, ns = non-significant). **d**. Hierarchical clustering of pseudobulk gene counts for each sample separated by cell type and condition. **e**. Number of up- and

down-regulated DEG (abs(LFC) > 0.5 and FDR < 0.01) in LgDel for each cell type. **f**. Heatmap with Pearson correlation of LFC LgDel profiles in different cell types. **g**. Extended heatmap with GSEA normalized enrichment score (NES) of biological processes (GO:BP) identified in all LgDel cell types. The asterisk (*) indicates adjusted p-value (p.adj) <0.05. P-values were adjusted with Bonferroni's multiple comparison test. **h**. Biological processes altered in human cerebral spheroids derived from 22q11.2 patients' cells. Data from Khan et al 2020.

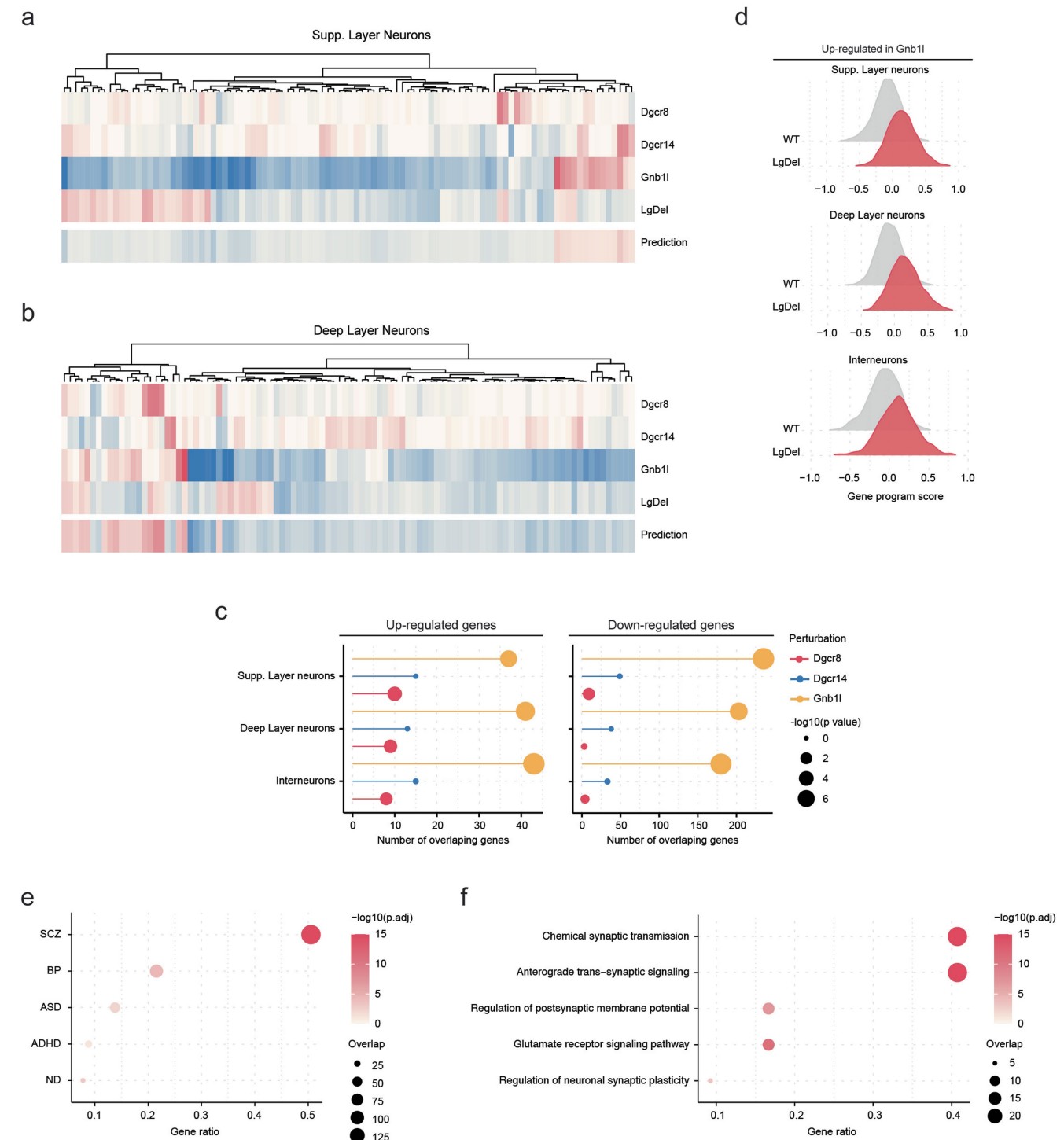

**Extended Data Fig. 11 | Individual perturbations partially explain LgDel transcriptional changes. a**. Heatmap showing the LFC values for the top 100 predicted genes in individual perturbations, LgDel, and the model prediction (LgDel = 0.03 Dgcr8 + 0.06 Gnb1l + 0.04 Dgcr14, dcor = 0.15) based on individual perturbations profiles in Superficial Layer Neurons. **b**. Heatmap showing the LFC values for the top 100 predicted genes in individual perturbations, LgDel, and the model prediction (LgDel = 0.18 Dgcr8 + 0.11 Gnb1l + (−0.13) Dgcr14, dcor = 0.37) based on individual perturbation profiles in Deep Layer Neurons. **c**. Number of genes similarly dysregulated in LgDel and individual perturbation gene programs. P-values calculated with hypergeometric test without multiple comparison correction **d**. Gene program score in WT and LgDel nuclei for the up-regulated program in *Gnb1l*-perturbed nuclei. **e**. DisGeNET enrichment analysis of genes commonly dysregulated in individual perturbations and LgDel transcriptional profiles (SCZ: Schizophrenia; BP: Bipolar Disorder; ADHD: Attention Deficit Hyperactivity Disorder; ASD: Autism Spectrum Disorder). P-values were adjusted with Bonferroni's multiple comparison test. **f**. Gene ontology analysis of schizophrenia-associated genes commonly dysregulated in individual perturbations and LgDel transcriptional profiles. P-values were adjusted with Bonferroni's multiple comparison test.

# Reporting Summary

## Statistics

For all statistical analyses, confirm that the following items are present in the figure legend, table legend, main text, or Methods section.

| n/a | Confirmed | |
|-----|-----------|---|
| ☐ | ☒ | The exact sample size (*n*) for each experimental group/condition, given as a discrete number and unit of measurement |
| ☐ | ☒ | A statement on whether measurements were taken from distinct samples or whether the same sample was measured repeatedly |
| ☐ | ☒ | The statistical test(s) used AND whether they are one- or two-sided *Only common tests should be described solely by name; describe more complex techniques in the Methods section.* |
| ☐ | ☒ | A description of all covariates tested |
| ☐ | ☒ | A description of any assumptions or corrections, such as tests of normality and adjustment for multiple comparisons |
| ☐ | ☒ | A full description of the statistical parameters including central tendency (e.g. means) or other basic estimates (e.g. regression coefficient) AND variation (e.g. standard deviation) or associated estimates of uncertainty (e.g. confidence intervals) |
| ☐ | ☒ | For null hypothesis testing, the test statistic (e.g. *F*, *t*, *r*) with confidence intervals, effect sizes, degrees of freedom and *P* value noted *Give P values as exact values whenever suitable.* |
| ☒ | ☐ | For Bayesian analysis, information on the choice of priors and Markov chain Monte Carlo settings |
| ☒ | ☐ | For hierarchical and complex designs, identification of the appropriate level for tests and full reporting of outcomes |
| ☐ | ☒ | Estimates of effect sizes (e.g. Cohen's *d*, Pearson's *r*), indicating how they were calculated |

*Our web collection on statistics for biologists contains articles on many of the points above.*

## Software and code

Policy information about availability of computer code

| | |
|---|---|
| Data collection | Data were collected using publicly available software, as referenced in the methods section. RNA and gRNA UMI count matrices were generated from raw sequencing data using CellRanger v5.0 (10x Genomics). |
| Data analysis | FANS data were analyzed with FlowJo v10.5.0. Analyses were done using publicly available R packages (BOWTIE 2 v2.3.5, Seurat v3.0, EdgeR v3.36.0, MASS v7.3-50, UWOT v0.1.8, Augur v1.0.0, g:Profiler v0.2.0, destiny v3.17, CRISPresso2 v2.0.20, Enrichr v2.1, fgsea v3.17). Custom made scripts are available through the Platt Lab GitHub (https://github.com/plattlab/AAV-Perturb-seq). See methods section for detailed information. |

For manuscripts utilizing custom algorithms or software that are central to the research but not yet described in published literature, software must be made available to editors and reviewers. We strongly encourage code deposition in a community repository (e.g. GitHub). See the Nature Portfolio guidelines for submitting code & software for further information.

## Data

Policy information about availability of data

All manuscripts must include a data availability statement. This statement should provide the following information, where applicable:
- Accession codes, unique identifiers, or web links for publicly available datasets
- A description of any restrictions on data availability
- For clinical datasets or third party data, please ensure that the statement adheres to our policy

Raw and processed sequencing data generated for this study are available through the Gene Expression Omnibus (GEO accession number GSE236519).

## Human research participants

Policy information about studies involving human research participants and Sex and Gender in Research.

| Reporting on sex and gender | N/A |
|---|---|
| Population characteristics | N/A |
| Recruitment | N/A |
| Ethics oversight | N/A |

Note that full information on the approval of the study protocol must also be provided in the manuscript.

# Field-specific reporting

Please select the one below that is the best fit for your research. If you are not sure, read the appropriate sections before making your selection.

☒ Life sciences          ☐ Behavioural & social sciences          ☐ Ecological, evolutionary & environmental sciences

For a reference copy of the document with all sections, see nature.com/documents/nr-reporting-summary-flat.pdf

# Life sciences study design

All studies must disclose on these points even when the disclosure is negative.

| Sample size | For the pooled screen, the brains of 15 LSL-Cas9 mice injected with a AAV library targeting 22q11.2 genes. This number of animals permits sufficient infected nuclei after sorting for GFP+ nuclei. To analyze the 22q11.2 animal model, 30,000 nuclei from 3 WT and 3 LgDel mice were sequence. |
|---|---|
| Data exclusions | Cells with low number of RNA counts (< 1000) or gene counts (< 500) were removed. Cells with 0 or more than 1 detected gRNA were removed. |
| Replication | Representation of gRNA molecules inside the gRNA library was confirmed by deep sequencing. Cas9 activity was confirmed by indel analysis. The truthfulness of our AAV-Perturb-seq experiments was confirmed by arrayed injections. Once the method was implemented, all attempts at replication were successful. |
| Randomization | LSL-Cas9 animals were randomly injected and kept in populations of 2 to 4 animals per cage. LgDel animals were genotype to confirm zygosity and alocated to experimental group WT (LgDel+/+) or LgDel (+/-) depending on the genotype. |
| Blinding | LSL-Cas9 mice selection was done in a blinded fashion. Animals were selected by the support team at the animal facility. Immunohistochemistry and the tri-color experiment data analysos were done in a blinded fashion. Single cell data analyses were not performed in a blinded fashion to support the development of new analysis methods. |

# Reporting for specific materials, systems and methods

We require information from authors about some types of materials, experimental systems and methods used in many studies. Here, indicate whether each material, system or method listed is relevant to your study. If you are not sure if a list item applies to your research, read the appropriate section before selecting a response.

## Materials & experimental systems

| n/a | Involved in the study |
|-----|----------------------|
| ☒ ☐ | Antibodies |
| ☐ ☒ | Eukaryotic cell lines |
| ☒ ☐ | Palaeontology and archaeology |
| ☐ ☒ | Animals and other organisms |
| ☒ ☐ | Clinical data |
| ☒ ☐ | Dual use research of concern |

## Methods

| n/a | Involved in the study |
|-----|----------------------|
| ☒ ☐ | ChIP-seq |
| ☐ ☒ | Flow cytometry |
| ☒ ☐ | MRI-based neuroimaging |

# Eukaryotic cell lines

Policy information about cell lines and Sex and Gender in Research

| | |
|---|---|
| Cell line source(s) | HEK293T cells were acquired from Sigma-Aldrich. |
| Authentication | HEK293T cells have been authenticated by the original vendors using short tandem repeat analysis. |
| Mycoplasma contamination | HEK293T cells were checked for mycoplasma every 3 months and tested negative throughout the study. |
| Commonly misidentified lines (See ICLAC register) | No misidentified cell lines were used in this study. |

# Animals and other research organisms

Policy information about studies involving animals; ARRIVE guidelines recommended for reporting animal research, and Sex and Gender in Research

| | |
|---|---|
| Laboratory animals | Male LSL-Cas9, dCas9-KRAB, and LgDel mouse models between 6 and 8 weeks of age. Mice were kept under specific pathogen-free conditions on a standard light cycle, temperature, and humidity environment. |
| Wild animals | No wild animals were used in this study. |
| Reporting on sex | We used male mice as brain disorders such as ASD and Schizophrenia tend to have an higher prevalence in males. |
| Field-collected samples | No field-collected samples were used in this study. |
| Ethics oversight | ETH Animal Welfare Office; University Basel Veterinary Office; Basel-Stadt Cantonal Veterinary Office (Switzerland) |

Note that full information on the approval of the study protocol must also be provided in the manuscript.

# Flow Cytometry

## Plots

Confirm that:

☒ The axis labels state the marker and fluorochrome used (e.g. CD4-FITC).

☒ The axis scales are clearly visible. Include numbers along axes only for bottom left plot of group (a 'group' is an analysis of identical markers).

☒ All plots are contour plots with outliers or pseudocolor plots.

☒ A numerical value for number of cells or percentage (with statistics) is provided.

## Methodology

| | |
|---|---|
| Sample preparation | Nuclei isolated from mouse brain tissue was processed as described in the methods section |
| Instrument | SONY MA 900 |
| Software | FlowJo |
| Cell population abundance | We sorted a minimum of 50,000 infected nuclei per condition. |
| Gating strategy | Nuclei was first gated with a DNA dye (Ruby dye, ThermoFisher). Single-nuclei were sorted based on GFP expression. |

☒ Tick this box to confirm that a figure exemplifying the gating strategy is provided in the Supplementary Information.

