## [Peer Review File · Nature]

Manuscript Title: Transcriptional linkage analysis with in vivo AAV-Perturb-seq

Reviewer Comments & Author Rebuttals

Reviewer Reports on the Initial Version:

Referees' comments:

Referee #1 (Remarks to the Author):

General overview of the manuscript

Santhina et al. develop a novel method to introduce genetic perturbations via AAV vectors (instead of lentiviral vectors) to in vivo mouse models to enable high throughput phenotyping. The phenotyping is done using single-cell RNA sequencing, thus, they name their method "AAV-Perturb-seq". They specifically apply their AAV-Perturb-seq method to dissect the phenotypic landscape underlying the 22q11.2 deletion syndrome genes in the adult mouse brain prefrontal cortex. The authors applied their method to study the 29 expressed genes in the minimal critical region of mouse 22q11.2 deletion syndrome region, generating an in vivo Perturb-seq dataset comprising 60,000 cells with assigned gRNAs. The finding that individual genetic perturbations produce unique transcriptional disturbances that are shared across cell types is an intriguing observation that would have been difficult to glean via other methods. Santhina et al. identify four 22q11.2-linked genes, which explain the transcriptional signatures observed in a 22q11.2 deletion mouse model. They hypothesize/conclude that these four genes, when heterozygous, result in broad dysregulation of a class of genes that involves dysfunctional RNA processing. The combination of pooled low MOI in vivo Perturb-seq, arrayed low MOI in vivo Perturb-seq, and comparison to a deletion mouse model is notably thorough. On the whole, it's a well written paper demonstrating how their method works, and specifically the utility of it by using it to further investigate the specific drivers behind the 22q11.2 deletion syndrome. The work is generally very clearly communicated, displayed, and analyzed.

Overall, this is a compelling demonstration of in vivo Perturb-seq with a vector platform that promises broad application and impact.

We do have some concerns regarding interpretation of results, certain claims of novelty over previous methods, clarity in description of the vector platform, and potential for gRNA missassignment that should be addressed. The comments are divided into major and minor concerns, and other specific comments and recommendations have been subdivided by sections below.

Major concerns

The repeated claim that “a major driver of the 22q11.2 deletion syndrome transcriptional phenotype found in mature neurons emerges after development” is not well-supported. The finding that perturbation of these genes in adulthood produces transcriptional phenotypes does not, on its own, suggest that these phenotypes are due to disruptions in adult biological processes in 22q11.2 deletion syndrome. It only indicates that these genes remain required for normal transcription in adulthood. The phenotype/s could still (and likely do) arise during development in individuals with the 22q11.2 deletion.

A clear demonstration of the benefits of the AAV approach described here over the previous lentiviral-based approach would solidify the contribution of the present work. How many new tissues and cell types are accessible with AAV over original lentiviral in vivo Perturb-seq? Seems all of the broad cell types studied here using AAV were also accessible to the lentiviral approach described by Jin et al., 2020 Science? What about expression time delays for AAV vs. lentivirus post-infection?

Will AAV Perturb-seq work with virally delivered Cas9 (e.g. SaCas9) so that it can be applied outside of Cas9 knock in models? Or would the packaging limit of AAV require split vector systems?

Is the “novel analysis approach” demonstrated in extended data figure 3 similar in concept to the one described by Adamson et al. 2016 Cell and applied again since then, e.g. by Tian et al., 2021 Nature Neuroscience? A clear demonstration (or minimally, discussion) of the improvements from this approach vs. the previous validated ones applied to this data would be useful.

What does the distribution of gRNA UMIs per cell look like for each gRNA and how was the cutoff to assign gRNAs to cells determined? Extended data figure 2 currently only shows the median. Is the uncertainty in gRNA assignment reflected in the statistical analysis pipeline?

Related to this, the description on line 896 stating: “For each nucleus, we remove all gRNAs with less than 60 total reads, less than 1 captured molecule, or when the gRNA-UMI counts represented less than 10% of all gRNAs identified in the nucleus. The gRNA counts, as well as the number of gRNAs detected in a given nucleus and their identity were appended to the nuclei metadata in the Seurat object” seems problematic. Less than one UMI is not a filter, and less than 10% of all gRNAs identified in the nucleus seems arbitrary. A clear visualization (e.g. histogram) of the UMIs per cell for each of the 29 gRNAs and how that information was used to assign gRNAs to cells would increase the clarity/transparency of the analysis.

Having the annotated sequences of constructs and a clear visualization of the gRNA capture strategy in the extended data would be ideal as they constitute a major contribution of the work. For example Line 819 does not provide sufficient detail: “To capture gRNA molecules, we altered the reverse transcription (RT) reaction to additionally include a gRNA-constant-region-targeting RT primer (0.15 μ M, AAGCAGTGGTATCAACGCAGAGTACCAAGTTGATAACGGACTAGCC).” - Was one of the Repogle et al capture sequences first added to the gRNA in the AAV vector to enable direct gRNA sequencing? Clear descriptions of the vector designs could also explain the currently puzzling differences in gRNA capture rates between 5' and 3' capture approaches.

Minor concerns

Line 44: Consider rephrasing “genotype-phenotypic landscapes” to “genotype-phenotype landscapes” as stated later in the MS or “complex relationships between genotype and phenotypic landscapes”

Line 207: Consider changing “in the Cas9 mouse brain” to “in mouse brain cells expressing Cas9”

Line 457: Consider changing the term “brain defects” to “alterations in neuronal functioning” or similar for sensitivity to affected families.

Line 923: What is the rationale for >5 genes being a real effect?

Comments divided by specific sections

Abstract

“Here, we introduce AAV-mediated direct in vivo single-cell CRISPR screens, termed AAV-Perturb-seq, a tunable and broadly applicable method for high-throughput and high-resolution phenotyping of genetic perturbations in vivo” - should the bolded word be “applicable”?

In vivo single-nucleus pooled CRISPR screening in the adult brain enabled through systemic administration of AAV.PHP.B and 5' gRNA capture

I recommend a better naming system for these two different capture methods/constructs. The reason I suggest this is because in addition to you using two different RNA-seq capture methods (3' and 5'), these are also two different constructs. Calling just the capture methods different (3' vs. 5') from these two differently-designed constructs is a bit misleading and confusing, since they are also two different constructs. A big difference is that the U6-gRNA part of the construct is on the 3' or 5' end of the larger AAV construct - this is almost just as important as the actual capture method itself being different.

Perturbation of Dgcr8, Dgcr14, Gnb1l, or Ufd1l result in strong transcriptional changes in prefrontal cortex neurons

You say that you developed a novel data analysis pipeline, but from what I understand, you pseudobulk your single-cell data, and then use edgeR-LTR to calculate a differential expression metric between your control cells and your perturbed cells. What about this is novel? This is a fine way to analyze your data, but I would not say it is novel (this has been done before). Highly recommend taking out the descriptor “novel” for this analysis.

I would say it is not remarkable that the four genes that came out of your analysis are all within the 1.5 Mb minimal region believed to be critical in 22q11.2-related disorders - doesn't this make a lot of sense? It is great that you have these hits in your screen, but this is in line with what would be expected if this experiment works.

For Figure 2D - what genes were used for this dimensionality reduction? Was this using the entire transcriptome or a subset of DEGs? If a subset of DEGs were used, I would clarify this (and this would make the UMAP visualization less unexpected).

Can you calculate statistical significance for the difference in Augur scores you calculate? It is currently very unclear how/if these scores are truly that different between perturbations.

"...Dgcr8, Dgcr14, Gnb1l, and Ufd1l affect largely non-overlapping transcriptional programs across neuron types." - your current analyses are not comprehensive enough to make this statement. The three observations you mention do not exclude the possibility that there are overlapping transcriptional programs. What transcriptional programs are shared between these differently perturbed cells? If you make this statement, an analysis has to be done to show that they do not affect overlapping transcriptional programs. In its current state, your analysis shows that these perturbations result in different transcriptional responses based on perturbation introduced, but you have not shown that they do not also have overlapping transcriptional programs.

Transcriptional response strength is not due to gene editing efficiency but the function of the gene

Recommend changing this title to "Strength of transcriptional response is directly due to gene function, and is not a consequence of gene editing efficiency". I think this gets your main message across more clearly.

For the 50% of nuclei that you found had a true perturbation - were the gRNAs that led to these true perturbations different from gRNAs that did not lead to successful perturbations? Is it random, or is it related to certain successful or unsuccessful gRNAs?

The second paragraph of this section is a bit contradictory to your first paragraph, as you say you deduced a filtering method to filter out cells that are unperturbed. You start this paragraph by saying that unperturbed cells essentially confound your ability to robustly identify perturbed cells. I would restructure this section to flow better - it is a bit confusing to say you solved this problem, but then you start the next paragraph saying this is a problem again? I think it is great and wonderful that you made single AAV preps and tested them, but it doesn't quite make sense to me that you did this in response to a concern over unsuccessful perturbations confounding your ability to identify perturbed cells (since you say you found a way to solve this issue via your analysis strategy).

For subtle transcriptional changes not detected by sn-RNA-seq - could you elaborate on this point or provide an example? This potential explanation is a bit vague and doesn't add much without further explanation, a concrete example, or a suitable reference.

Your arrayed experiments are a beautiful recapitulation of your pooled experiment, really nice!

Perturbation of 22q11.2 DS genes results in the disruption of distinct sets of biological processes

“Given the accumulation of these pre-miRNAs, we hypothesized that reduced levels of the mature miRNAs should lead to up-regulation of their target genes.” - this part needs to be explained better - it would benefit from a visual schematic of what is being perturbed, what is accumulating and what is being up-regulated in response. It is very unclear the way it is written what is occurring here (hence, a visual would be helpful).

Single-nucleus prefrontal cortex atlas of a 22q11.2 DS mouse model

Extended data figure 11c - can you report statistical significance or non-significance for this box plot to show that the cell type numbers are in fact the same?

LgDel and individual gene perturbations share disease-associated risk alleles

This section describes a really nice analysis and is a very nice addition to your manuscript.

Discussion

The regression model you used was capable of predicting 40% of the variance observed in the LgDel mouse model - I do not think it is appropriate to say that this means “approximately half of the transcriptional phenotype observed in LgDel model mice neurons could be explained by the perturbation of three genes in adult animals”. This is too vague and not specific. You should say what you found - which is that 40% of the variance can be explained by the perturbation of these three genes. This is not the same as saying half of the transcriptional phenotype can be explained by the perturbation of these three genes.

Figures

Figure 1d would be better as a pie chart (or other nice visualization of the gRNA library contents)

Figure 2b - legend should say “Number of DEGs” (plural), not “Number of DEG”

Figure 2e - add statistical significance testing values (i.e. p-values) to plot

Figure 2d - I would abbreviate “Sup. Layer Neurons” to “Supp. Layer Neurons” (just add a p). I would use this abbreviation throughout the manuscript (supp. instead of sup.)

Figure 3a - I would have progression go from left to right on the bottom panel as well (instead of in a clockwise direction)

Figure 3d-f - add statistical significance p-values from statistical significance testing

Figure 4 legend title - would change based on my comment above (that 40% of the variance is explained, not of the transcriptional signature overall)

Figure 4 g, h - add statistical significance p-values from statistical significance testing

Figure 5 - I would abbreviate “Sup. Layer Neurons” to “Supp. Layer Neurons” (just add a p)

Extended Data Figure 1f - why do the first and third FACS plots not look the same? Aren't you looking at nuclei in both cases? This needs to be reconciled, or the figure legend needs to explain

what exactly is being shown on these FACS plots. It is currently unclear.

Referee #2 (Remarks to the Author):

KEY RESULTS

The authors set out to develop approaches for linking genetic variants to rich phenotypes in vivo. To do so, they propose and implement an adaptation of the Perturb-seq screening approach using AAVs instead of lentiviruses to deliver guide RNAs. This enables them to efficiently deliver genetic perturbations to living mouse brains and read out the results via single-nucleus RNA sequencing. They apply this technology to study genes associated with 22q11.2 deletion syndrome, which is caused by a microdeletion containing some 37 genes. With AAV-Perturb-seq, the authors can engineer individual knockouts of each of these genes in mouse brains. They can then compare the resulting transcriptional phenotypes to those seen in a mouse model of the disease containing the full 22q11.2 deletion, and thus explore how the individual gene phenotypes might influence the overall one.

VALIDITY

The two most pressing concerns are: (1) There is a significant cell-type-dependent rate of undetected gRNAs. This raises questions both about the fidelity of the approach and its broader applicability to diverse cell types, which is one of the key stated advantages. (2) 22q11.2 is a haploinsufficiency syndrome, but it is being studied here using what are presumably homozygous deletions of the relevant genes (given the observed high rate of editing and the lethality of one of the perturbations). It's unclear how relevant the full knockout phenotypes are to understanding the biology of the disease. Some additional concerns are addressed in more specific comments below.

ORIGINALITY AND SIGNIFICANCE

The primary technical advance in the paper is the development of the AAV-Perturb-seq technology. Though Perturb-seq has been done in the past within the mouse brain using lentiviral vectors (Jin et al., Science, 2020), AAV vectors have significant advantages for some tissues and for therapeutic uses. [Side note: As mentioned below in my comments below, the authors should more directly state the advantages of the AAV approach for a general audience.] The technology is novel and appears promising based on the authors' sincere efforts to characterize its effectiveness. The application to 22q11.2 deletion syndrome is also new, as is the attempt to explain a polygenic in vivo phenotype using this type of approach (to my knowledge).

DATA, METHODOLOGY, AND SUGGESTED IMPROVEMENTS

As an opening comment, the paper is well-written, easy-to-understand, and pleasant to look at. Though I have some nitpicks about various things I found confusing below, these were the exception to the rule. The authors should be credited for rigorously testing and reporting the performance of their method, which allowed me to answer a lot of questions I had as I read the manuscript.

My main reservations are that the method has a "version 1" quality to it due to the issues with gRNA calling. Given competition in this space from more refined lentiviral approaches, a stronger case needs to be made that the AAV technology can accomplish things that those approaches cannot. As I

understand it, one instance of that is neurological diseases, as AAV easily penetrates the brain, though perhaps the authors can make a better argument. The second challenge is that after reading the manuscript quite carefully, I'm still not sure how conclusive the findings about 22q11.2 deletion syndrome really are. I would be interested to hear whether the authors can address some of these questions and concerns.

Fidelity of gRNA calling

Though the AAV technology is an interesting adaptation, I am concerned that the fidelity of gRNA calling is a potential barrier for future applications. Extended Data Fig. 4a/b suggest that there is a significant cell-type-dependent underdetection issue. E.g. in neurons, the average number of gRNAs detected per nucleus (i.e. infection events per cell) looks to be about 1, but 30-40% of these sorted, GFP+ nuclei (therefore containing the construct) have no gRNA detected. If my thinking is correct, that suggests the true number of infection events per cell is more like $1/0.6$ or $1/0.7 = 1.42 - 1.67$, i.e. the cells are superinfected on average. Some of these additional undetected infection events will be in cells already identified as containing multiple gRNAs, but it suggests a reasonable fraction of the cells identified as containing single gRNAs may in fact contain more than one. The rate of detection in some of the other cell types also appears to be low enough that it cannot be relied upon—I'm assuming these cell types simply don't express very many copies of the gRNA. Though obviously there is plenty of exciting work to do in neurons, this is potentially a significant barrier for the technology to be applied in less tractable cell types and I'd be interested to see it discussed more openly. How does this not affect sensitivity of differential expression etc.? It's a bit hard not to draw contrasts here with the lentiviral approach used in the Jin et al. *in vivo* Perturb-seq paper, which seems to have more robust calling.

As a minor presentational point, it would be helpful to show histograms of gRNA UMI counts broken down by cell type. The means/medians are not that informative about dropout.

Relevance of homozygous null phenotypes to understanding 22q11.2 deletion syndrome

22q11.2 deletion syndrome is a haploinsufficiency syndrome. In contrast, the CRISPR approach here appears to mostly be generating homozygous nulls based on Extended Data Fig. 6c. (Incidentally, it is unclear to me what the error bars on Fig. 2e and 6c indicate.) Indeed, the gene *Upf1l* is discarded from analysis because the perturbation is overtly toxic on short timescales, suggesting homozygous deletion of an essential gene.

The goal of the linear modeling is to examine how an additive combination of effects from many genes might lead to the 22q11.2 DS phenotype. Though I think this is an interesting approach, the problem is frankly it is not clear whether the observed phenotypes, which I suspect reflect full knockouts, are the relevant ones to include in this analysis. For example, some of the single gene deletions in the Perturb-seq experiment appear to have stronger phenotypes on their own (in terms of number of differentially expressed genes) than the entire *LgDel* heterozygous deletion. And the role of *Upf1l* cannot be examined at all because it had to be removed.

I think this issue really deserves to be acknowledged in the main text as a challenge for this type of approach and I'd be interested to hear the authors' comments on why this is not a major issue for

the interpretation of the later analyses. Did you by chance make any efforts to identify heterozygotes in the experiment?

Bias induced by LDA fitting procedure

In the LDA-based filtering procedure, I understand the logic of discarding perturbed cells that are essentially “mis-labeled.” It seems odd, however, to apply the same logic to the control cells, where presumably the label is correct. It might improve the quality of results to only discard the perturbed cells that fail this filter. Indeed, the fact that the computational procedure discards 50% of cells while validation experiments (Extended Data Fig. 6C) suggest most cells are edited seems consistent with the filtering procedure being too conservative and/or emphasizing outlier phenotypes. For example, I’m wondering whether heterozygous deletions are disproportionately discarded, since they will likely have intermediate phenotypes between control cells and full knockouts.

Details about pseudobulk differential expression procedure

I’m a bit confused about what is used as a replicate in edgeR. The methods section mentions that analysis and pseudo-bulk aggregation for the main Perturb-seq experiment is done within independent “sequencing lanes.” Is each “sequencing lane” treated as a replicate? Or is each animal treated as a replicate? How many cells are averaged in each pseudobulk profile? How were independent animals grouped for sequencing and what is the definition of a sequencing lane?

I was surprised that there was no discussion of batch effects by animal. If you overlay animal in Fig. 1e is there structure? (Or is this not possible because samples were merged in a way that this information was lost?)

Given the attrition from guide calling and filtering, it seems likely some of the analyses are not adequately powered to detect weak effects. It’s a practical reality that some hypotheses will not be testable in this challenging in vivo context, but it seems clear comparing Fig. 1e/f to Fig. 2a that the strength of phenotype is at least somewhat proportional to the number of cells in each class. What I’m wondering more is how this affects the ability to detect weaker transcriptional phenotypes even in the well-represented cell types. Given the observed high correlations between transcriptional phenotypes across different neuronal subtypes, have you tried aggregating all neurons to increase power? Do any of the weaker perturbations then reveal phenotypes? It’s a bit surprising that genes like *Ranbp1*, which is also essential in many cell types, do not yield phenotypes.

Minor points

More details on the mouse genotypes and numbers used should be included in the main text as the experiments are described. In some cases such as the arrayed confirmation experiments it is hard to tell even in the methods how many mice are involved.

It is unclear to me what “edgeR-LTR” is. If you Google this term you get nothing. How is it different from normal edgeR?

As a very lightweight analysis, have you looked at the relative representation of each guide across cell type? From 1f they look roughly conserved, but I'm just wondering if any cell-type-dependent differences in representation manifest (as a result of e.g. overt toxicity of any guides).

"Across all cell types, only genes within the 1.5 Mb deleted locus exhibited a negative LFC, while adjacent genes had no expression changes": I think this needs to be rephased to state the intended point more clearly. Extended Data 11e shows that there are other downregulated genes.

A silly comment: given how targeted the library is and the citation to the literature, the "Remarkably" comment on pg. 8 about the 4 genes with phenotypes lying within the minimal region is perhaps unnecessary

Referee #3 (Remarks to the Author):

Santinha and colleagues use in vivo pooled CRISPR screens with single-cell transcriptomics to identify key genes and cell types implicated in 22q11.2 DS. Compared to prior approaches (e.g. in vivo perturb-seq from Xin et al., PMID: 33243861), this work expands in two important areas: 1) AAV-mediated perturbations (which is superior to lentivirus for many in vivo applications) and 2) improved single-cell transcript and guide capture using 5' chemistry.

Using this new platform, the group identifies 4 key genes driving 22q11.2, the relevant developmental timepoints, and functional deficits in RNA processing. Overall, this paper is exciting for both the novel biological insights into 22q11.2 DS and from a technical standpoint with single-cell in vivo CRISPR screens. This new platform should be of broad applicability to mouse models of disease and identification of disease mechanisms. For this second reason in particular, I am supportive of its publication in a top-tier journal. However, I have a few minor comments that I believe should be addressed prior to publication:

* I commend the authors on the very careful validation studies (using individual AAVs in separate mice). The indel data (Cas9 editing) is very convincing. However, it seems like a bit of a missed opportunity to delve deeper into the RNA processing/splicing defects. For example, the authors mention "a disrupted balance between RNA transcription and splicing resulting from perturbation of Dgcr14". This should be demonstrated beyond DEG and GO analyses. Unfortunately, single-cell/nucleus RNA-seq does not provide splicing data. Might the authors instead use long-read sequencing? It would be helpful to understand whether Dgcr14 loss causes widespread changes in splicing (i.e. all transcripts) or whether there is a specific enrichment for long transcripts, neuron/synapse transcripts, etc.

* The authors find that the phenotype is not really developmentally-driven but rather an issue with mature neurons. Would it be possible in the LgDel mouse to rescue using the 4 genes (or CRISPR activators)? This would provide the most convincing demonstration that the 4 identified genes are sufficient for treating the disease (and outline a gene therapeutic approach for future human clinical studies). This may be beyond the scope of the paper (and thus should not be required for

publication) but would certainly be quite exciting.

* The authors should comment further on the gene *Pi4ka*, which seems to meet their significance criterion in astrocytes (Fig 2b). The other genes seem to have neuron-focused disruptions so this seems like it might be of interest and perhaps functioning via distinct mechanisms.

* The technique is misnamed. Perturb-seq is a 3'-based capture method. In this work, the authors specifically test 3' and 5' methods head-to-head and adopt the 5' method (analogous to Mimitou et al., reference 17). Given that, and to avoid confusion, please rename the method AAV-ECCITE-seq, as ECCITE-seq (ref 17) is the first work to describe the 5' capture strategy used.

* The introduction mentions "a very narrow range of developmental time points, tissues, and cell types conducive to lentiviral infection in vivo." However, no citations are provided for this statement. I thought the primary use of AAV was superior spread (targeting more cells in the brain) than lenti. The authors should provide citations to support these claims — especially "narrow range of developmental time points".

AAV-mediated single-nucleus CRISPR screening enables functional dissection of the 22q11.2 deletion syndrome in vivo

We truly appreciate the constructive feedback and enthusiasm expressed by all reviewers. It was a pleasure to read and respond to each point, which overall resulted in an improved manuscript with enhanced robustness and clarity. We are confident that the revised work addresses all concerns raised by the reviewers and brings new biological and methodological insights as well as computational and experimental capabilities.

To facilitate the reviewers' work while navigating this document, below we summarize the most important additions and provide a table that maps old to new figures, and then respond point-by-point to each reviewer concern. We also include a revised text where all changes from the original version are highlighted.

Summary of changes in the revised manuscript

New experiments

- 1) AAV-Perturb-seq experiment using CRISPRi-mediated knockdown of *Dgcr8* in vivo to model haploinsufficiency (Reviewer Comment 2.4)
- 2) Astrocyte-specific AAV-Perturb-seq experiment to demonstrate the applicability of the technology in less-abundant cell types (Reviewer Comment 3.3)
- 3) Splicing analysis of *Dgcr14*-perturbed neurons using bulk RNA-seq performed on sorted nuclei (Reviewer Comment 3.1)

New analyses

- 1) AAV-Perturb-seq analyses for the new experiments 1 and 2 described above (Reviewer Comment 2.4 and 3.3)
- 2) Alternative splicing analysis for new experiment 3 described above (Reviewer Comment 3.1)
- 3) Computational stratification of cellular zygosity states using diffusion maps (Reviewer Comment 2.4)
- 4) Exploration and justification of our gRNA filtering/calling strategy (Reviewer Comment 1.6)
- 5) Comparison between gRNA filtering/calling strategies in our work compared to in vivo Perturb-seq described in Jin et al (Reviewer Comment 2.2)

- 6) Transcriptional phenotypes detected with all neuron types together (Reviewer Comment 2.8).
- 7) Overlap between DEGs or biological processes associated with each perturbation in neurons (Reviewer Comment 1.18).
- 8) Statistical analyses in gene programs expression plots (Reviewer Comment 1.33 and 1.35)
- 9) Evidence that LDA filters WT-like nuclei without removing weak signals (Reviewer Comment 2.5)

New main figures

None

Major changes to main figures

- 1) Figure 1d was removed. Subsequent panels were shifted by one letter.

New Extended Data figures

- 1) Extended Data Figure 8: Computational dissection of zygoty in array perturbed nuclei
- 2) Extended Data Figure 9: Experimental modelling of haploinsufficiency with CRISPRi

Major changes to Extended Data figures

- 1) Extended Data Figure 1: correction of FACS gating strategy in panel F
- 2) Extended Data Figure 2: change 3' and 5' labels. Introduce pAS006 and pAS088 labels to plasmids.
- 3) Extended Data Figure 4: new panel C with gRNA UMI counts per cell type
- 4) Extended Data Figure 10: panel A was previously in Extended Data Figure 7g
- 5) Extended Data Figure 11: new panel D with intersection of DEGs and biological processes between perturbations
- 6) Extended Data Figure 12: panel A is now a schematic explaining Dgcr8-mediated pri-miRNA processing. Subsequent panels were shifted by one letter.

Mapping old to new figures (frame shifts)

In this document, we will refer to both the old and new figure name to avoid confusion.

Old figure	New figure
Figure 1e	Figure 1d
Figure 1f	Figure 1e
Figure 1g	Figure 1f
Extended Data Figure 4c	Extended Data Figure 4d
Extended Data Figure 4d	Extended Data Figure 4e
Extended Data Figure 4e	Extended Data Figure 4f
Extended Data Figure 4f	Extended Data Figure 4g
Extended Data Figure 6a	Extended Data Figure 6b
Extended Data Figure 6b	Extended Data Figure 6c
Extended Data Figure 6c	Extended Data Figure 6a
Extended Data Figure 7g	Extended Data Figure 10a
Extended Data Figure 8	Extended Data Figure 10
Extended Data Figure 9	Extended Data Figure 11
Extended Data Figure 10	Extended Data Figure 12
Extended Data Figure 11	Extended Data Figure 13
Extended Data Figure 12	Extended Data Figure 14
Extended Data Figure 13	Extended Data Figure 15

Referees' comments:

Referee #1 (Remarks to the Author):

General overview of the manuscript

Santhina et al. develop a novel method to introduce genetic perturbations via AAV vectors (instead of lentiviral vectors) to in vivo mouse models to enable high throughput phenotyping. The phenotyping is done using single-cell RNA sequencing, thus, they name their method "AAV-Perturb-seq". They specifically apply their AAV-Perturb-seq method to dissect the phenotypic landscape underlying the 22q11.2 deletion syndrome genes in the adult mouse brain prefrontal cortex. The authors applied their method to study the 29 expressed genes in the minimal critical region of mouse 22q11.2 deletion syndrome region,

generating an in vivo Perturb-seq dataset comprising 60,000 cells with assigned gRNAs. The finding that individual genetic perturbations produce unique transcriptional disturbances that are shared across cell types is an intriguing observation that would have been difficult to glean via other methods. Santhina et al. identify four 22q11.2-linked genes, which explain the transcriptional signatures observed in a 22q11.2 deletion mouse model. They hypothesize/conclude that these four genes, when heterozygous, result in broad dysregulation of a class of genes that involves dysfunctional RNA processing. The combination of pooled low MOI in vivo Perturb-seq, arrayed low MOI in vivo Perturb-seq, and comparison to a deletion mouse model is notably thorough. On the whole, it's a well written paper demonstrating how their method works, and specifically the utility of it by using it to further investigate the specific drivers behind the 22q11.2 deletion syndrome. The work is generally very clearly communicated, displayed, and analyzed.

Overall, this is a compelling demonstration of in vivo Perturb-seq with a vector platform that promises broad application and impact.

We thank the referee for taking the time to review our manuscript. We appreciate that they share our enthusiasm about the potential and promise of our approach.

We do have some concerns regarding interpretation of results, certain claims of novelty over previous methods, clarity in description of the vector platform, and potential for gRNA missassignment that should be addressed. The comments are divided into major and minor concerns, and other specific comments and recommendations have been subdivided by sections below.

Major concerns

Reviewer comment 1.1: The repeated claim that “a major driver of the 22q11.2 deletion syndrome transcriptional phenotype found in mature neurons emerges after development” is not well-supported. The finding that perturbation of these genes in adulthood produces transcriptional phenotypes does not, on its own, suggest that these phenotypes are due to disruptions in adult biological processes in 22q11.2 deletion syndrome. It only indicates that these genes remain required for normal transcription in adulthood. The phenotype/s could still (and likely do) arise during development in individuals with the 22q11.2 deletion.

We acknowledge and agree with the referee's view. We are not suggesting that the disease phenotype emerges in adulthood. The developmental nature of 22q11.2 deletion syndrome is well established. What we are stating is that there is more to the disease than abnormal development. Our study centered on developing a broadly applicable new technology (AAV-Perturb-seq) and using it to systematically investigate the role of all expressed 22q11.2 deletion syndrome genes individually in mature cells, which has never been done before. Our results show that a small subset of 22q11.2 genes are involved in cellular processes in mature neurons, which when knocked out, result in transcriptional phenotypes very similar to those we found via snRNA-seq in LgDel model. The fact that part (approximately 40%) of the LgDel transcriptional changes in mature neurons is explained by perturbation of three genes (after normal development) is striking and unexpected. We acknowledge that more work is required to fully resolve the mechanism and determine if correcting the loss of these genes rescues the phenotype. This is our goal for future studies. Taken together, we entirely agree with the reviewer on the fundamental importance of development in 22q11.2 deletion syndrome, but also want to highlight that the loss of certain 22q11.2 genes in mature neurons may also be contributing to the phenotype. We have revised the abstract and discussion to clarify the importance of development, acknowledge the limitations of our study, and better articulate the new insights that our study provides.

Reviewer comment 1.2: A clear demonstration of the benefits of the AAV approach described here over the previous lentiviral-based approach would solidify the contribution of the present work. How many new tissues and cell types are accessible with AAV over original lentiviral in vivo Perturb-seq? Seems all of the broad cell types studied here using AAV were also accessible to the lentiviral approach described by Jin et al., 2020 Science? What about expression time delays for AAV vs. lentivirus post-infection?

The same point regarding a comparison of AAV vs LV was raised by all reviewers (**Reviewer Comment 2.1 and 3.5**) and we provided a detailed answer in **Reviewer Comment 3.5**. As this is of general interest, we included our response in the **Supplementary Note 1**.

Reviewer comment 1.3: Will AAV Perturb-seq work with virally delivered Cas9 (e.g. SaCas9) so that it can be applied outside of Cas9 knock in models? Or would the packaging limit of AAV require split vector systems?

There is nothing fundamentally incompatible between AAV-Perturb-seq and applications in non-genetically modified animals. As the reviewer points out, performing gene editing outside of Cas9 knock in models requires either a smaller Cas9 (e.g., SaCas9) to ensure all of the necessary elements fit in one vector or a dual or split AAV approach. These alternative approaches are well established and there are several studies applying them in vivo in

multiple tissues. For example, here are a selection of references for smaller Cas9s¹, dual AAVs², and split AAVs^{3,4}. The only unique consideration for performing AAV-Perturb-seq in these contexts is controlling the multiplicity of infection. This could be achieved through delivering a limiting amount of the vector containing the gRNA (as we demonstrated in this study), but then a much higher amount of the second vector containing everything else. There are also straightforward ways to only label cells/nuclei that receive both vectors, which could be achieved by, for example, encoding a different half of split eGFP-KASH in each vector. This approach has also been shown before⁵. Taken together, given the extensive literature support of in vivo gene editing in non-genetically modified animals and the compatibility between AAV-Perturb-seq and these vectors, we think it is fair to conclude that AAV-Perturb-seq would work outside of Cas9 knock in models. We have written a supplementary note to include this perspective (**Supplementary Note 6**).

Reviewer comment 1.4: Is the “novel analysis approach” demonstrated in extended data figure 3 similar in concept to the one described by Adamson et al. 2016 Cell and applied again since then, e.g. by Tian et al., 2021 Nature Neuroscience? A clear demonstration (or minimally, discussion) of the improvements from this approach vs. the previous validated ones applied to this data would be useful.

This is an important point, and we agree that we could have better articulated the similarities and differences between our pipeline and previous work.

We believe that there are two main steps that should be carefully considered while analyzing single-cell CRISPR screening data: 1) filtering of gRNA-labelled cells; and 2) identifying gene expression changes mediated by individual perturbations. After exploring pipelines published previously (**Rebuttal Table 1**), we designed an approach that we believe best reflects current knowledge in the field. We aimed to apply strict metrics and thresholds to increase confidence in our findings, with the tradeoff that we may miss tenuous signals that can be confused with noise. Below, we discuss the two critical steps and clarify differences between our approach and previous methods.

Rebuttal Table 1: Cell filtering and transcriptional signal detection in previously published scRNA-seq CRISPR screens.

Analysis step	Adamson, et al. (Perturb-seq) ⁶	Other examples in literature	Our study	Benefit of our approach
Removal of non-perturbed cells	Not applicable (subsequent publication from the same authors look directly at expression of target genes ^{7,8})	Linear models ⁹ , cosine similarity ¹⁰ , or Local Outlier Factor ¹¹ between control and perturbed cells with a subset of genes with p-value < 0.05.	LDA with a subset of gene with FDR < 0.05.	By using FDR rather than p-values, we make sure that the filter is supported by true DE genes.
Identification of expression changes (informative genes)	Kolmogorov-Smirnov test and random forest	Linear models ⁹ and cell-type specific genes ⁶ .	Aggregate nuclei into artificial pseudobulk samples followed by DE with edgeR-LRT.	Recent findings indicate that pseudobulk methods outperform single-cell methods ¹² .
Association of transcriptional changes to biological processes	Pre-define lists of genes associated with expected processes. Combination of random forest and gene clustering.	Dimensional reduction methods ¹³ or WGCNA ^{11,14} to identify informative components (i.e. , groups of genes). All detected genes used as input.	Gene ontology analysis with DEGs for each perturbation and cell type.	Focus on high confidence DEGs supported by LFC and FDR values.

1) Filter non-perturbed but gRNA-labelled cells. On average, Cas9-gRNA complexes generate an indel in approximately 80% of infected cells (**Figure 2e**), meaning that approximately 20% of cells were not gene edited. Furthermore, a fraction of the edited cells will contain approximately 33% of in-frame deletions that should not lead to loss of function (see **Reviewer Comment 1.20**). The transcriptome of these “non-perturbed” cells should remain unchanged and should be indistinguishable from the control group. Different methods have been applied to remove non-perturbed cells and we will now describe them and explain why we are confident our method is robust. Specifically, in one of the publications indicated by the reviewer (Adamson et al⁶), they did not perform any filtering to remove non-perturbed cells. In later publications from the same labs (Jonathan Weissman and Britt Adamson^{7,8}), they used CRISPR inhibition or activation to regulate the mRNA expression of target genes without the introduction of mutations. Such methods lead to direct down or up regulation of target transcripts, which can be used to remove cells that do not present the desired regulation. In our case, the introduction of indels into a target gene typically leads to

loss of function mutations without necessarily affecting the gene's mRNA level⁹, and thus we cannot use a mRNA quantification-based method. As a side point, we now show that AAV-Perturb-seq is compatible with CRISPRi (**Reviewer Comment 2.4**).

Other publications using Cas9 to induce mutations have explored the perturbation inference problem using linear models⁹, cosine similarity¹⁰, and Local Outlier Factor¹¹. While we have no reasons to criticize these methods, but we think there are improvements that can be made when selecting the most informative genes for subsequent analysis. Ideally, one wants to use a subset of features that are enough to distinguish perturbation X from control, but not so big that it prompts overfitting by including more variables (genes) than observations (single cells). Across these publications, authors use a linear model, Kolmogorov–Smirnov test, and Student's t-test with a cut-off at p-value < 0.05 to identify DEGs between control and all gRNA X-labelled cells, and later use this DEGs subset as input for the filtering algorithms (**Rebuttal Table 1**). Importantly, note that the authors use p-value and not adjusted p-values. We believe that this way of feature selection is influenced by two factors that reduce their power. First, given the large scale of single-cell data (*i.e.*, the high number of cells in each group), tests looking at probability distributions, such as the Kolmogorov-Smirnov test, tend to find many genes with $p < 0.05$ expression changes without biological significant changes due to small differences detected by high sample counts. Second, scRNA-seq is affected by high dropout rates, especially among lowly expressed genes, which gives to all detected genes a non-zero fold change value that reflects technical limitations rather than biological signal. By using a relaxed threshold (no threshold on fold change values and non-adjusted p-value of 0.05), these methods use features that may not be truly differentially expressed, thus introducing noise into the filtering process.

Considering this computational step of perturbation inference, the fundamental difference between us and other publications is that we only rely on confidently assigned DEGs by thresholding by FDR rather than p-value. Moreover, after we implemented our approach, a publication by Papalexi et al¹⁵ corroborated our assumptions whereby they demonstrate that MIMOSCA and MUSIC (which use p-value thresholding) fail to properly filter unperturbed control cells.

We believe that our approach, which is restricted to truly differential expressed genes, is superior at realistically identifying perturbed cells.

2) Identify significant and robust gene expression changes. After removing the noise introduced by non-perturbed cells, one should focus on identifying transcriptional changes that explain the consequences of perturbations in the target gene and its biological relevance. Previous work has employed different techniques that can be grossly grouped into dimensional reduction methods (WGCNA, PCA, NMF) and single-cell-specific differential

expression analysis. For instance, Adamson et al⁶ used a combination of Kolmogorov-Smirnov (KS) test and random forest. We will comment on KS test below. Regarding the random forest test, its main advantage is also its limitation. Specifically, random forest allows for the identification of the most important features (genes) used by the model to classify distinct perturbations. However, it assumes that all perturbations lead to distinct transcriptional phenotypes as it associates a specific set of informative genes to each perturbation. Tian et al¹¹, Jin et al¹⁴, and others, have used dimensional reduction methods to group the entire transcriptional space (around 5000 genes in single-cell experiments) into smaller groups, or components. These typically cluster genes with similar expression patterns. However, it is not obvious which genes are the most relevant ones inside the groups.

The most used approach to identify informative genes rely on methods commonly used to find DEGs from single-cell data, such as linear models^{9,16}, logistic regression¹⁵, KS test⁶, and Mann–Whitney U test⁸. A recent publication by Squair et al¹² presents an exceptional and exhaustive work to compare the performance of different DE methods across multiple scRNA-seq datasets. The authors' two most preeminent findings are: 1) commonly used methods tend to have high false discovery rates and overestimate the number of DEGs and 2) the generation of artificial pseudobulks of cells from the same experimental group to use as input for bulk RNA-seq DE methods outperforms approaches that compare groups using individual cells (single-cell common methods mentioned above). Thus, we chose to implement a pseudobulk approach. We also evaluated a single-cell specific DE method (logistic regression). However, as reported before by others, this method has a bias towards highly expressed genes (**Extended Data Figure 4f, previous Extended Data Figure 4e**).

3. Pseudobulk performs superior to single-cell specific methods in a true positive control.

The LgDel model carries a heterozygous deletion in chromosome 16 (equivalent to human 22q11.2 locus). We hypothesized that this can be used as a control to test DE methods, as we know that genes within the locus are being expressed from only one copy, and thus, in general, their expression should be reduced to approximately 50%. In **Rebuttal Figure 1** we show that, on average, our pseudobulk detects LFC values of approximately -1 (50% reduction) across deleted genes and cell types, while the logistic regression test typically applied to single-cell data presents smaller LFC values and high variance.

Overall, we are confident that our strict approach to this problem increases the likelihood of retrieving true biological significance. We agree that calling our analysis “novel” may have been an overstatement and removed it from the main text.

We wrote a supplementary note based on this discussion (**Supplementary Note 4**).

Rebuttal Figure 1: Log fold change values of heterozygous deleted genes in LgDel mouse model calculated with our pseudobulk approach (left) and logistic regression, a method typically used for single-cell data (right).

Reviewer comment 1.5: What does the distribution of gRNA UMIs per cell look like for each gRNA and how was the cutoff to assign gRNAs to cells determined? Extended data figure 2 currently only shows the median. Is the uncertainty in gRNA assignment reflected in the statistical analysis pipeline?

We answer **Reviewer comment 1.5** and **Reviewer comment 1.6** together. Please find below our systematic dissection of gRNA capture and filtering approaches.

Reviewer comment 1.6: Related to this, the description on line 896 stating: “For each nucleus, we remove all gRNAs with less than 60 total reads, less than 1 captured molecule, or when the gRNA-UMI counts represented less than 10% of all gRNAs identified in the nucleus. The gRNA counts, as well as the number of gRNAs detected in a given nucleus and their identity were appended to the nuclei metadata in the Seurat object” seems problematic. Less than one UMI is not a filter, and less than 10% of all gRNAs identified in the nucleus seems arbitrary. A clear visualization (e.g. histogram) of the UMIs per cell for each of the 29 gRNAs and how that information was used to assign gRNAs to cells would increase the clarity/transparency of the analysis.

The following information relates to **Reviewer comment 1.5** and **Reviewer comment 1.6**. We respond to this comment from four angles.

1) In **Figure 3b** we show the results of an arrayed experiment (*i.e.*, the gRNAs were never pooled), which correlate extremely well with the pooled data. If our filtering steps were not

robust, we would not see this level of correlation. This is our strongest evidence that our filtering steps are appropriate and robust.

2) Thank you for highlighting that less than one UMI is not a filter. This was a typo. We removed all gRNAs with only 1 UMI count (*i.e.*, we only consider gRNAs with two or more UMI counts). The reasoning for this is below. This has been changed in the main text.

3) We will now walk you through our filtering steps and provide justification for each. Before we begin, to provide context on how this work is done in the field, here is a summary of gRNA filtering strategies published previously.

Rebuttal Table 2: summary of gRNA filtering strategies published previously.

Publication	Sequencing strategy	Perturbation capture	Label strategy	Filter
Dixit, et al. (Perturb-seq) ⁹	DROP-seq, 3P	Indirect	Barcode	Focus on reads rather than UMIs. Keep gRNAs with proportion over 20%.
Adamson, et al. (Perturb-seq) ⁶	DROP-seq, 3P	Indirect	Barcode	Remove events by coverage. Keep events with more than 3 UMIs and 50 reads.
Datlinger, et al. (CROP-seq) ¹⁷	DROP-seq, 3P	Indirect	gRNA	Keep the most abundant gRNA, if the proportion of the dominant gRNA is 3x higher than the sum of all other gRNAs.
Jaitin, et al. (CRISP-seq) ¹⁸	MARS-seq, 3P	Indirect	Barcode	Keep all gRNA with more than 1 UMI.
Replogle, et al. ⁸	10Gen, 5P	Direct	gRNA	Variable by gRNA following a bimodal distribution.
Mimitou, et al. (ECCITE-seq) ¹⁵	10x Gen, 5P	Direct	gRNA	Keep the most abundant gRNA. If more than 1 gRNA has high counts, the cell is labelled as doublet. It is not clear what is the quantitative meaning of "high counts".
Frangieh, et al. (Perturb-CITE-seq) ¹⁶	10xGen, 3P	Indirect	Barcode	Keep gRNAs with more than 20 UMIs, more than 200 reads and proportion over 20%.
Jin, et al. (In vivo Perturb-seq) ¹⁴	10xGen, 3P	Indirect	Barcode	Keep the most abundant gRNA if its UMIs count is 1.3x higher than the second most abundant gRNA.

Our strategy to assign gRNAs to cells was designed to consider three important details of single-cell data generation and introduced filtering steps to minimize the risk for each of them. Each filter alone cannot perfectly stratify true from false signal, but collectively result in a robust enrichment of signal over noise:

3.1) Reads per UMI coverage filter to correct for chimeric molecules. First reported in Dixit et al⁹, chimeric sequences are created during PCR amplification of gRNA molecules. Such events may create molecules composed by a combination of cell barcode, UMI and gRNA that wasn't originally present in the cDNA library. Therefore, chimeric sequences could only emerge in our dataset from template swapping during PCR amplification, which we mitigate computationally. As these molecules appear later in the single-cell library preparation protocol, they are less abundant and generate fewer reads. To remove chimeric molecules, we calculate coverage, namely the number of reads per UMI ($READ_count / UMI_count$), and remove molecules with low coverage applying a threshold based on the coverage distribution (**Rebuttal Figure 2a-b**). As can be seen in **Rebuttal Figure 2b**, there is a bimodal distribution of the coverage which supports the removal of molecules with coverage lower than 60 (less than 60 reads per molecule). It should be noted that coverage depends on sequencing depth and should be re-calculated for every new sequencing data.

3.2) Total UMI count filter to correct for chimeric molecules and ambient RNA. For all gRNAs, we found a high number of events with only one UMI count within and across many cells (**Rebuttal Figure 2c**). Such a phenomenon would not be expected given the low multiplicity of infection at which we infect the brain (**Figure 1b**). We and others^{9,19} therefore accept that these events result from ambient RNA contamination or chimeric sequences created during library preparation. In such a scenario, a possible contaminating gRNA molecule is expected to have very few UMI counts, while an expressed gRNA is more likely to have many UMIs (as there are multiple RNA copies inside the nucleus). As can be seen in **Rebuttal Figure 2c**, there is a disproportional high number of molecules with 1 UMI count, which supports this filtering step.

3.3) UMI proportion filter to correct for chimeric molecules and ambient RNA. The number of detected UMIs in a nucleus correlates directly with the number of reads. A nucleus with a higher number of UMIs is more likely to also have a higher number of UMIs coming from contaminating gRNA and mRNA. A gRNA associated with a small number of UMIs that represent a small proportion of the total gRNA-UMIs identified in a nucleus most likely represents a chimeric read or RNA cross contamination. Proportion-based filters have been used previously to address this issue^{15,19}. We incorporate this concept in our workflow but increased the stringency of the threshold compared to published work (**Rebuttal Table 2**). For instance, in the in vivo Perturb-seq (Jin et al¹⁴) paper, they filter out gRNAs from nuclei where the most abundant gRNA UMI count is 1.3x higher than the UMI counts for the second most abundant gRNA. Such a threshold only considers information about the two most expressed gRNAs and selects a gRNA label based on the highest expressed gRNA, even if there is a second gRNA with high counts. For instance, one cell containing 10 and 14 UMIs

for two different gRNAs would be labeled as only having one gRNA. On the other hand, we remove events only from nuclei where the gRNA UMI count is at least 10x higher. This extra stringency gives the confidence that we are removing events from nuclei where there is a large difference on the number of UMIs per gRNA. For instance, one cell containing 10 and 14 UMIs for two different gRNAs would be labeled as having both gRNAs, whereas another cell containing 2 and 20 UMIs for two different gRNAs would be labeled as only having one gRNA. With gRNAs not having the same capture efficiency, we strongly believe that this extra stringency is important for confidently identifying cells with single infections/perturbations.

4) We can assess the impact of these filtering steps through modifying their thresholds and comparing how the outputs of the analysis change.

4.1) We applied a range of thresholds to the number of UMI- and proportion-based filters and compared gRNA-nuclei assignment (**Rebuttal Figure 3a**). For each filter combination (umi_X_prop_Y, X = {1, 2, 3} and Y = {0, 0.1, 0.2}) we identified putatively expressed gRNAs for each nucleus, use this information for gRNA assignment, and labeled nuclei expressing unique gRNAs (MOI = 1) in at least one of the filter combinations. We then calculated the pairwise label accuracy between any two filter combinations (**Rebuttal Figure 3a, top heatmap**). We observed a high accuracy between thresholds, indicating that nuclei tend to be labelled with the same gRNA across threshold strategies.

4.2) As the accuracy was not 100%, we investigated which nuclei had different assignments. We repeated the analysis from point 1 keeping only nuclei that were labelled with a gRNA for both thresholds (*i.e.*, removing nuclei that were not assigned to a gRNA by one of the thresholds in the pairwise comparison). This analysis highlights whether thresholding changes the gRNA associated to a given nuclei or serves to remove nuclei with low-confidence assignments. We observed an accuracy of 100% between any two pairs of thresholds, which strongly suggests that thresholding only removes nuclei with low-confidence assignments without changing gRNA assignment itself for high-confidence nuclei (**Rebuttal Figure 3a, bottom heatmap**).

4.3) We next investigated how thresholding affects the transcriptional signal associated with a perturbation by performing DE analysis between SH control and *Dgcr14*-perturbed nuclei using each threshold. As seen in **Rebuttal Figure 3b**, the filter threshold combination used throughout our work (umi_1_prop_0.1) yielded the higher number of significant DEGs (FDR < 0.05), which indicates our ability to remove potentially low-confidence nuclei that introduce noise. Nevertheless, we observed that all thresholds lead to similar expression changes (LFC directionality) and mainly affect the LFC magnitude and consequently the statistical power (FDR) to call a gene differentially expressed (**Rebuttal Figure 3b**).

These results indicate that our gRNA thresholding removes background noise from subsequent analyses (increasing our power to detect biological pathways) without affecting gRNA association to truly informative cells.

In conclusion, we recognize that single cell data is noisy, and we took systematic and conservative steps to ensure the robustness of our workflow, especially regarding gRNA assignment. Our workflow goes beyond current gold standards because we use carefully selected conservative filters to remove noise and increase our confidence in signal. Finally, the overall workflow and our findings were independently validated in the arrayed experiment where we tested perturbations one by one and compared them to the pooled data (**Figure 3b**), which provides the strongest possible evidence that our workflow for processing pooled AAV-Perturb-seq data is robust.

As we believe this discussion is important for the general audience, we integrated it in **Supplementary Note 3**.

Rebuttal Figure 2: Filtering strategy of gRNA counts. **a.** Correlation between sequencing reads and UMI counts associated with gRNAs. Color code indicates the coverage filtering threshold. **b.** Histogram showing gRNA counts coverage. Dotted line indicates the coverage threshold. **c.** Histogram of UMIs counts per gRNA across all gRNAs. Red line indicates UMI count higher than 1.

Rebuttal Figure 3: Evaluation of filtering thresholds in the transcriptional signature of *Dgcr14*-perturbed nuclei. **a.** Heatmap with pairwise accuracy of gRNA assignments between any two filtering thresholds. Accuracy was calculated using all nuclei (top) nuclei that are labelled with a gRNA for both thresholds (bottom). **b.** Log fold change values (LFC) calculated based on nuclei assigned to a unique gRNA in each thresholding group.

Reviewer comment 1.7: Having the annotated sequences of constructs and a clear visualization of the gRNA capture strategy in the extended data would be ideal as they constitute a major contribution of the work. For example Line 819 does not provide sufficient detail: “To capture gRNA molecules, we altered the reverse transcription (RT) reaction to additionally include a gRNA-constant-region-targeting RT primer (0.15 μ M, AAGCAGTGGTATCAACGCAGAGTACCAAGTTGATAACGGACTAGCC).” - Was one of the Repogle et al capture sequences first added to the gRNA in the AAV vector to enable direct gRNA sequencing? Clear descriptions of the vector designs could also explain the currently puzzling differences in gRNA capture rates between 5’ and 3’ capture approaches.

We agree that having annotated sequences of constructs and primers and clear schematics will facilitate the implementation of AAV-Perturb-seq by others. We have included this in the in **Supplementary Note 2: 5’ and 3’ gRNA molecules capture**. For each capture approach, the new schematics include the full plasmid maps (pAS006 and pAS088 for 3’ and 5’ capture, respectively), a representation of the AAV genome in the plasmids, the encoded RNA transcripts that are expressed in the infected cells, annotated RT primer binding sites that initiate reverse transcription of mRNA (3’ capture) and gRNA (5’ capture), and the PCR amplification steps for enriching gRNA sequences within the cDNA library and appending the necessary adaptors and indexes for Illumina sequencing. We also include a table with all primer sequences, as well as the sequences of hU6 and the gRNA constant region.

We will also respond to the reviewers comment on the RT primer design and gRNA capture rates. As the gRNA targeting RT primer (same sequence used by Replogle et al⁸) binds to our gRNA constant region, we did not alter this sequence, which was originally introduced by Hsu et al²⁰.

The reviewer comments on the differences in gRNA capture rates between 5' and 3' capture approaches. We believe this comes from: 1) the hU6 pol-III promoter leads to high RNA expression when compared with pol-II promoters²¹, and consequently more RNA molecules are available to be captured during single-nucleus library preparation; 2) our 3' capture approach construct (pAS006) expresses pol-II transcripts containing a WPRE sequence that mediates mRNA nuclear export, leading to reduced numbers of mRNA molecules containing the gRNA in the nucleus available for capture (and this should not be occurring with our 5' capture approach construct (pAS088) containing the pol-III U6 promoter to express gRNAs that should not be exported from the nucleus) ; and 3) AAV is known to lead to higher transgene expression than LV, also likely leading to gRNA molecules and better gRNA capture (see **Reviewer Comment 3.5** for a systematic comparison between AAV and LV).

Minor concerns

Reviewer comment 1.8: Line 44: Consider rephrasing “genotype-phenotypic landscapes” to “genotype-phenotype landscapes” as stated later in the MS or “complex relationships between genotype and phenotypic landscapes”

This has been rectified in the main text.

Reviewer comment 1.9: Line 207: Consider changing “in the Cas9 mouse brain” to “in mouse brain cells expressing Cas9”

This has been rectified in the main text.

Reviewer comment 1.10: Line 457: Consider changing the term “brain defects” to “alterations in neuronal functioning” or similar for sensitivity to affected families.

This is a important point, we appreciate you bringing it to our attention. We have rectified it throughout the main text.

Reviewer comment 1.11: Line 923: What is the rationale for >5 genes being a real effect?

We use a threshold of at least 5 DEGs because it is the minimum number we can use for the LDA filtering step without bias the filtering. We confirmed this in **Reviewer Comment 2.8** where we showed that lowering number of DEGs led to random filtering and removal of all nuclei from further analysis.

Comments divided by specific sections

Abstract

Reviewer comment 1.12: “Here, we introduce AAV-mediated direct in vivo single-cell CRISPR screens, termed AAV-Perturb-seq, a tunable and broadly applicable method for high-throughput and high-resolution phenotyping of genetic perturbations in vivo” - should the bolded word be “applicable”?

Yes, thank you. This was corrected in the main text.

In vivo single-nucleus pooled CRISPR screening in the adult brain enabled through systemic administration of AAV.PHP.B and 5' gRNA capture

Reviewer comment 1.13: I recommend a better naming system for these two different capture methods/constructs. The reason I suggest this is because in addition to you using two different RNA-seq capture methods (3' and 5'), these are also two different constructs. Calling just the capture methods different (3' vs. 5') from these two differently-designed constructs is a bit misleading and confusing, since they are also two different constructs. A big difference is that the U6-gRNA part of the construct is on the 3' or 5' end of the larger AAV construct - this is almost just as important as the actual capture method itself being different.

This is a fair point. We now refer to these constructs as pAS006 (gRNA expression cassette embedded within the mRNA) and pAS088 (independent gRNA expression cassette). We emphasize in the main text, methods, supplementary note, and **Extended Data Figure 2** that

pAS006 was tested in combination with 3' sequencing technology while pAS088 is used with 5' sequencing.

Perturbation of Dgcr8, Dgcr14, Gnb1l, or Ufd1l result in strong transcriptional changes in prefrontal cortex neurons

Reviewer comment 1.14: You say that you developed a novel data analysis pipeline, but from what I understand, you pseudobulk your single-cell data, and then use edgeR-LTR to calculate a differential expression metric between your control cells and your perturbed cells. What about this is novel? This is a fine way to analyze your data, but I would not say it is novel (this has been done before). Highly recommend taking out the descriptor "novel" for this analysis.

This is a fair point. We used the term "novel" to signify that we were bringing in a new strategy to analyze single-cell CRISPR screens, but not specific analytical methods in general. Our intention was to highlight that we applied, for the first time, pseudobulk methods for the analysis of pooled screen data. Previous publications have all used single-cell-specific methods, linear models, and dimensionality reduction algorithms. Supported by rigorous benchmarking studies¹², we believe that such single-cell-specific approaches may underperform due to the high level of uncertainty and intrinsic noise typically associated with single-cell datasets. We further elaborate on the different computational methods for pooled single-cell CRISPR screen data analysis in **Reviewer Comment 1.6**.

In sum, to avoid misleading readers we have removed "novel" from the text and better articulated what is new with our approach.

Reviewer comment 1.15: I would say it is not remarkable that the four genes that came out of your analysis are all within the 1.5 Mb minimal region believed to be critical in 22q11.2-related disorders - doesn't this make a lot of sense? It is great that you have these hits in your screen, but this is in line with what would be expected if this experiment works.

While we believe the result is remarkable, because of how it makes sense and further supports the findings, we can agree that the word may be overly sensational and have therefore removed it from the text.

Reviewer comment 1.16: For Figure 2D - what genes were used for this dimensionality reduction? Was this using the entire transcriptome or a subset of DEGs? If a subset of DEGs were used, I would clarify this (and this would make the UMAP visualization less

unexpected).

We apologize for the lack of clarity. To generate the UMAP plots in **Figure 2d**, we used all DEGs identified for all perturbations in the pooled screen. We have revised the figure legends and methods to better describe the inputs for dimensionality reduction.

In addition to revising the text to increase clarity, we will justify why we believe our approach, which is in-line with what is currently done in the field, is appropriate.

1) Data clustering and visualization typically requires selection of the most informative features. Specifically, for scRNA-seq data, this step is important for two reasons: 1) most genes add redundancy to the data and are not informative as they are lowly expressed contributing to technical noise or their expression variance across cells is low; and 2) we aim to have fewer variables (genes) than samples (cells) to avoid overfitting. Different approaches have been taken to select the most informative genes. For instance, the Seurat pipeline²² applies polynomial regression to identify a user-defined number of variable genes (2000 by default) that are used as variables for principal component analysis (PCA). In a second step, the user defines the number of informative principal components to be used for clustering and UMAP visualization. Other methods to select informative genes use published curated datasets (transcriptomics, epigenomics, protein markers) to identify lists of relevant features²³. Relevant features can be identified by differential analysis when the cells' labels are known. As nuclei in our dataset have labels for both cell type and perturbations, we can use DE analysis to identify the most informative genes and use these directly to calculate UMAP coordinates. Note that our approach does not require any previous knowledge of transcriptional changes induced by perturbation and thus can be scaled to unknown phenotypes. Similar approaches have been used to visualize single-cell perturbation data. Adamson et al⁶ used Kolmogorov-Smirnov test to identify DEG, followed by low rank independent component analysis (low rank ICA) and t-SNE plots. Papalexi et al¹⁵ first identified DEG using logistic regression and use those genes for linear discriminant analysis (LDA) followed by UMAP. Taken together, feature selection is a common, and often necessary, for visualizing single cell data.

2) UMAP visualization confirms that individual perturbations lead to distinct transcriptional phenotype. Our approach uses a combination of all DEGs from all perturbations. We calculate UMAP coordinates with all passing-filter perturbations and control nuclei at the same time, which enables UMAP to yield information not only about the differences between control and perturbed cells, but also differences between perturbations.

To conclude, we consider our approach appropriate and in-line with current best practices to represent single-cell data in a 2D UMAP visualization. Together with clustering of DEGs (**Figure 2c**), Augur scores (**Extended Data Figure 5b**), and Hotelling's T-squared statistics (**Extended Data Figure 4g, previous Extended Data Figure 4f**), UMAP reinforces the different transcriptional phenotypes detected between perturbation groups, as well as the strong clustering of nuclei with the same perturbation.

Reviewer comment 1.17: Can you calculate statistical significance for the difference in Augur scores you calculate? It is currently very unclear how/if these scores are truly that different between perturbations.

The Augur score calculation results in one data point per perturbation and cell type, which makes it difficult to calculate statistical significance between any two individual points. While this analysis does not allow for rigorous comparisons across perturbations, we nonetheless think it is informative as it provides orthogonal evidence of the strong transcriptional phenotypes induced by the four perturbations of interest. We have revised the methods section to clarify this limitation.

Reviewer comment 1.18: "...Dgcr8, Dgcr14, Gnb1l, and Ufd1l affect largely non-overlapping transcriptional programs across neuron types." - your current analyses are not comprehensive enough to make this statement. The three observations you mention do not exclude the possibility that there are overlapping transcriptional programs. What transcriptional programs are shared between these differently perturbed cells? If you make this statement, an analysis has to be done to show that they do not affect overlapping transcriptional programs. In its current state, your analysis shows that these perturbations result in different transcriptional responses based on perturbation introduced, but you have not shown that they do not also have overlapping transcriptional programs.

We believe the reviewer raises a pertinent question, which highlights the ambiguity in the term "transcriptional program". Below we clarify this ambiguity by looking at overlaps between perturbations at the level of differentially expressed genes and biological pathways.

First, in our analysis we start by focusing on the identification of transcriptional differences between control and perturbed nuclei by pseudobulk DE analysis. Differential gene expression is the first step towards identifying differences in the transcriptional phenotypes between conditions. In **Figure 2c** and **Figure 3 c-f** we show that each perturbation is

associated with a unique set of DEGs. In other words, we do not detect overlapping DEGs between *Dgcr8*, *Dgcr14*, and *Gnb1l* perturbations (**Rebuttal Figure 4, left**).

Second, while we may not see overlap at the level of DEGs, there may be an overlap at the level of biological pathways. In **Figure 3c** we show a summary of biological functions associated with each perturbation and in **Extended Data Figure 10b (previous Extended Data Figure 8)** we present an exhaustive list of biological processes for each perturbation phenotype. As with DEGs, we do not detect any overlap in altered biological pathways between *Dgcr8*, *Dgcr14*, and *Gnb1l* perturbations (**Rebuttal Figure 4, right**).

Taken together, at the DEG and biological pathway level we do not observe an overlap between perturbations, which supports that the phenotypes are perturbation-specific. We have revised the manuscript to include the two new figures showing a lack in overlap at the DEG and biological pathway level (**Extended Data Figure 11d**).

Rebuttal Figure 4: Overlap between DEGs lists (left) and identified biological processes (right) associated with each perturbation in neurons.

Transcriptional response strength is not due to gene editing efficiency but the function of the gene

Reviewer comment 1.19: Recommend changing this title to “Strength of transcriptional response is directly due to gene function, and is not a consequence of gene editing efficiency”. I think this gets your main message across more clearly.

We agree that the title could be clearer and appreciate the suggestion. This new section title was implemented in main text to read: **Altered transcriptional phenotypes are due to gene function and not a consequence of gene editing efficiency.**

Reviewer comment 1.20: For the 50% of nuclei that you found had a true perturbation - were the gRNAs that led to these true perturbations different from gRNAs that did not lead to successful perturbations? Is it random, or is it related to certain successful or unsuccessful gRNAs?

We asked ourselves the exact same question previously and performed several experiments and analyses to address it.

1) Our first thought was that good/successful gRNAs were perturbing cells (leading to nuclei being labeled as perturbed after LDA filtering) while bad/unsuccessful gRNAs were not (leading to nuclei not being labeled as perturbed after LDA filtering). We performed the experiment shown in **Figure 2e** to address this. In this experiment, we quantified the gene editing efficiency of 8 gRNAs, half coming from nuclei labeled as perturbed and the other half coming from nuclei labeled as not perturbed. The indel analysis plot shows that the gene editing efficiency for all gRNAs is very similar, centering around 80% (which is a very high gene editing efficiency for an in vivo experiment). From this we can conclude that the success of a gRNA was not determining whether nuclei were being labeled as perturbed or not.

2) Our second thought was that while the DNA was being mutated, maybe the resulting mutations were non-random and not leading to an expected fraction of frame-shifting, loss of function mutations. However, if we look at the spectrum of mutations, we see a typical DNA repair pattern where more than two thirds of the mutations are frame-shifting, knockout mutations (**Extended Data Figure 6a, previous Extended Data Figure 6c**). Given this spectrum of mutations, and the number of in-frame mutations, we think this is the main reason why not all cells containing a gRNA are perturbed (explaining the reviewer's comment of why 50% of gRNA-containing nuclei are not labeled as perturbed).

3) It is also important to mention that the possibility of an unsuccessful gRNA within a library is not new or unique to our methodology. How we and others account for this is by building redundancy into the library using multiple gRNAs per gene. In our work, we used 2 gRNAs per gene. If gRNA success was driving nuclei being labeled as perturbed or not, we would expect to see large differences in nuclei passing filter for different gRNAs. Therefore, we plotted the percentage of nuclei passing filter for each gRNA targeting *Dgcr8*, *Dgcr14*,

Gnb1l, and *Ufd1l* across cell types (**Rebuttal Figure 5**). Overall, we observed that the percentage of retained nuclei is comparable for the two gRNAs targeting the same gene. The exception being the *Dgcr8* targeting set, in which the second gRNA (*Dgcr8_2*) tends to lead to a higher proportion of perturbed cells. We think these results are within the expected gRNA activity ranges and support the design of libraries containing multiple gRNA targeting the same gene.

Rebuttal Figure 5: Percentage of nuclei passing filter for each gRNA across cell types.

Taken together, the success/efficiency or particular design strategy of a gRNA does not explain nuclei being labeled as perturbed or not for the subset that we investigated deeply, and the redundancy built into our library should account for any unsuccessful gRNAs.

We suspect that not all gRNA-containing nuclei are labeled as perturbed because of the mosaic nature of gene editing, where gRNA-containing cells can have wild-type or non-frame-shifting, loss of function mutations. For example, the situation of approximately 50% non-perturbed nuclei could easily arise when the editing rates are ~80%, the in-frame mutation rate is as approximately one third, and there is typical noise in the single cell data. Overall, this suggests that our nuclei labels are appropriately describing the transcriptional changes resulting from perturbation of the genes of interest.

Reviewer comment 1.21: The second paragraph of this section is a bit contradictory to your first paragraph, as you say you deduced a filtering method to filter out cells that are unperturbed. You start this paragraph by saying that unperturbed cells essentially confound your ability to robustly identify perturbed cells. I would restructure this section to flow better - it is a bit confusing to say you solved this problem, but then you start the next paragraph saying this is a problem again? I think it is great and wonderful that you made single AAV preps and tested them, but it doesn't quite make sense to me that you did this in response to a concern over unsuccessful perturbations confounding your ability to identify perturbed

cells (since you say you found a way to solve this issue via your analysis strategy).

We appreciate this suggestion and have implemented this in the revised text.

Reviewer comment 1.22: For subtle transcriptional changes not detected by sn-RNA-seq - could you elaborate on this point or provide an example? This potential explanation is a bit vague and doesn't add much without further explanation, a concrete example, or a suitable reference.

It is well known that single-cell and single-nucleus RNA-seq are limited when it comes to detecting subtle changes in the transcriptome. Some examples of why include technical noise, limited sensitivity for low-abundance transcripts, and high gene dropout rates²⁴. As scRNA-seq is not an absolute technique, *i.e.*, it does not capture all available RNA molecules inside a cell, low-abundance transcripts tend to be less represented in the dataset which can confound data analysis and signal identification²⁴. Furthermore, single-cell data suffers from high gene dropout rates with a significant fraction of genes not detected in individual cells (REF). These limit the identification of true subtle gene expression dynamics of individual cells. We tried to minimize the impact of these variables by performing pseudobulk DE identification (which is less sensitive to gene dropouts as counts are aggregated) and focusing on truly differential expressed genes (FDR < 0.05). To provide more information and clarity, we have revised the text to elaborate the point as well as included two references directing readers to published reviews that fully delve into single-cell limitations and mitigation strategies. 1) Armand et al²⁵, who exhaustively discuss the use of single-cell techniques in brain tissue; and 2) Luecken et al²⁴, who focus on best practices for faithful analysis of single cell data.

Reviewer comment 1.23: Your arrayed experiments are a beautiful recapitulation of your pooled experiment, really nice!

Thank you! We strongly believe that these arrayed experiments strongly support the robustness of the pooled method.

Perturbation of 22q11.2 DS genes results in the disruption of distinct sets of biological processes

Reviewer comment 1.24: "Given the accumulation of these pre-miRNAs, we hypothesized that reduced levels of the mature miRNAs should lead to up-regulation of their target

genes.” - this part needs to be explained better - it would benefit from a visual schematic of what is being perturbed, what is accumulating and what is being up-regulated in response. It is very unclear the way it is written what is occurring here (hence, a visual would be helpful).

We agree that this could better explained and depicted. We have created and included in **Extended Data Figure 12a (new addition to previous Extended Data Figure 10)** a visual schematic to explain the consequences of *Dgcr8* perturbation in the pri-miRNA processing process and consequent alterations of transcript levels of the miRNA target genes. We have also revised the text to better explain the process.

Rebuttal Figure 6: Schematic representation of pri-miRNA processing mediated in SH control and *Dgcr8*-perturbed cells.

Single-nucleus prefrontal cortex atlas of a 22q11.2 DS mouse model

Reviewer comment 1.25: Extended data figure 11c - can you report statistical significance or non-significance for this box plot to show that the cell type numbers are in fact the same?

We now report the statistical results (t-test) in the figure and legend.

LgDel and individual gene perturbations share disease-associated risk alleles

Reviewer comment 1.26: This section describes a really nice analysis and is a very nice addition to your manuscript.

Thank you!

Discussion

Reviewer comment 1.27: The regression model you used was capable of predicting 40% of the variance observed in the LgDel mouse model - I do not think it is appropriate to say that this means “approximately half of the transcriptional phenotype observed in IgDel model mice neurons could be explained by the perturbation of three genes in adult animals”. This is too vague and not specific. You should say what you found - which is that 40% of the variance can be explained by the perturbation of these three genes. This is not the same as saying half of the transcriptional phenotype can be explained by the perturbation of these three genes.

We agree that the alternative suggestion is clearer and more direct. This suggestion was implemented throughout the manuscript (abstract, results, discussion, figure legends).

Figures

Reviewer comment 1.28: Figure 1d would be better as a pie chart (or other nice visualization of the gRNA library contents)

We agree that the current implementation is not ideal. We explored different visualizations, include the nice suggestion of a pie chart, but we did not find a solution we were happy with. Thus, we removed this panel from the figure and make reference to the library design in the main text and methods section.

Reviewer comment 1.29: Figure 2b - legend should say “Number of DEGs” (plural), not “Number of DEG”

Thank you for catching this. This was corrected.

Reviewer comment 1.30: Figure 2e - add statistical significance testing values (i.e. p-values) to plot

Given that the Indel (%) for each condition is so obviously different than the control condition, we do not think it is necessary to include the p-values in the plot. We nonetheless created the plot as shown below. If the reviewers and editor insist, we will use the version with the p-values below.

Rebuttal Figure 7: Deep sequencing-based gene editing (indel) analysis for four gRNAs targeting 22q11.2 genes with strong transcriptional phenotypes (*Dgcr8*, *Dgcr14*, *Gnb1l*, and *Ufd1l*) and four gRNAs targeting genes without significant transcriptional phenotypes (*Comt*, *Med15*, *Ranbp1*, and *Pi4ka*). Student's t-test, *** FDR < 0.001.

Reviewer comment 1.31: Figure 2d - I would abbreviate "Sup. Layer Neurons" to "Supp. Layer Neurons" (just add a p). I would use this abbreviation throughout the manuscript (supp. instead of sup.)

This was implemented throughout the text and figures.

Reviewer comment 1.32: Figure 3a - I would have progression go from left to right on the bottom panel as well (instead of in a clockwise direction)

This was implemented.

Reviewer comment 1.33: Figure 3d-f - add statistical significance p-values from statistical significance testing

This was implemented in the figure legend of **Figure 3d-f** and **Extended Data Figure 11** (previous **Extended Data Figure 9**).

Reviewer comment 1.34: Figure 4 legend title - would change based on my comment above (that 40% of the variance is explained, not of the transcriptional signature overall)

This was corrected to read: **Figure 4.** Transcriptional signatures found in LgDel model neurons are partially explained by perturbation of *Dgcr8*, *Dgcr14*, and *Gnb1l*.

Reviewer comment 1.35: Figure 4 g, h - add statistical significance p-values from statistical significance testing

This was implemented in the figure legend.

Reviewer comment 1.36: Figure 5 - I would abbreviate "Sup. Layer Neurons" to "Supp. Layer Neurons" (just add a p)

This was implemented throughout the text and figures.

Reviewer comment 1.37: Extended Data Figure 1f - why do the first and third FACS plots not look the same? Aren't you looking at nuclei in both cases? This needs to be reconciled, or the figure legend needs to explain what exactly is being shown on these FACS plots. It is currently unclear.

We modified the figure to clarify our gating strategy using the Vibrant Ruby dye to identify single nucleus and GFP to isolate infected nuclei.

Referee #2 (Remarks to the Author):

KEY RESULTS

The authors set out to develop approaches for linking genetic variants to rich phenotypes in vivo. To do so, they propose and implement an adaptation of the Perturb-seq screening approach using AAVs instead of lentiviruses to deliver guide RNAs. This enables them to efficiently deliver genetic perturbations to living mouse brains and read out the results via single-nucleus RNA sequencing. They apply this technology to study genes associated with 22q11.2 deletion syndrome, which is caused by a microdeletion containing some 37 genes. With AAV-Perturb-seq, the authors can engineer individual knockouts of each of these genes

in mouse brains. They can then compare the resulting transcriptional phenotypes to those seen in a mouse model of the disease containing the full 22q11.2 deletion, and thus explore how the individual gene phenotypes might influence the overall one.

VALIDITY

The two most pressing concerns are: (1) There is a significant cell-type-dependent rate of undetected gRNAs. This raises questions both about the fidelity of the approach and its broader applicability to diverse cell types, which is one of the key stated advantages. (2) 22q11.2 is a haploinsufficiency syndrome, but it is being studied here using what are presumably homozygous deletions of the relevant genes (given the observed high rate of editing and the lethality of one of the perturbations). It's unclear how relevant the full knockout phenotypes are to understanding the biology of the disease. Some additional concerns are addressed in more specific comments below.

ORIGINALITY AND SIGNIFICANCE

The primary technical advance in the paper is the development of the AAV-Perturb-seq technology. Though Perturb-seq has been done in the past within the mouse brain using lentiviral vectors (Jin et al., Science, 2020), AAV vectors have significant advantages for some tissues and for therapeutic uses. [Side note: As mentioned below in my comments below, the authors should more directly state the advantages of the AAV approach for a general audience.] The technology is novel and appears promising based on the authors' sincere efforts to characterize its effectiveness. The application to 22q11.2 deletion syndrome is also new, as is the attempt to explain a polygenic in vivo phenotype using this type of approach (to my knowledge).

DATA, METHODOLOGY, AND SUGGESTED IMPROVEMENTS

As an opening comment, the paper is well-written, easy-to-understand, and pleasant to look at. Though I have some nitpicks about various things I found confusing below, these were the exception to the rule. The authors should be credited for rigorously testing and reporting the performance of their method, which allowed me to answer a lot of questions I had as I read the manuscript.

We thank the reviewer for their thorough and thoughtful response as well as highlighting the value of our work.

Reviewer comment 2.1: My main reservations are that the method has a "version 1" quality to it due to the issues with gRNA calling. Given competition in this space from more refined lentiviral approaches, a stronger case needs to be made that the AAV technology can

accomplish things that those approaches cannot. As I understand it, one instance of that is neurological diseases, as AAV easily penetrates the brain, though perhaps the authors can make a better argument. The second challenge is that after reading the manuscript quite carefully, I'm still not sure how conclusive the findings about 22q11.2 deletion syndrome really are. I would be interested to hear whether the authors can address some of these questions and concerns.

This is an important point raised by all reviewers, and we respond to it thoroughly in **Reviewer Comment 3.5**. We also included the response in **Supplementary Note 1**.

Fidelity of gRNA calling

Reviewer comment 2.2: Though the AAV technology is an interesting adaptation, I am concerned that the fidelity of gRNA calling is a potential barrier for future applications. Extended Data Fig. 4a/b suggest that there is a significant cell-type-dependent underdetection issue. E.g. in neurons, the average number of gRNAs detected per nucleus (i.e. infection events per cell) looks to be about 1, but 30-40% of these sorted, GFP+ nuclei (therefore containing the construct) have no gRNA detected. If my thinking is correct, that suggests the true number of infection events per cell is more like $1/0.6$ or $1/0.7 = 1.42 - 1.67$, i.e. the cells are superinfected on average. Some of these additional undetected infection events will be in cells already identified as containing multiple gRNAs, but it suggests a reasonable fraction of the cells identified as containing single gRNAs may in fact contain more than one. The rate of detection in some of the other cell types also appears to be low enough that it cannot be relied upon—I'm assuming these cell types simply don't express very many copies of the gRNA. Though obviously there is plenty of exciting work to do in neurons, this is potentially a significant barrier for the technology to be applied in less tractable cell types and I'd be interested to see it discussed more openly. How does this not affect sensitivity of differential expression etc.? It's a bit hard not to draw contrasts here with the lentiviral approach used in the Jin et al. in vivo Perturb-seq paper, which seems to have more robust calling.

We agree with the reviewer that gRNA calling is a critical step in single-cell CRISPR screening, and we took many steps to ensure robustness. We address this reviewer's concern from five perspectives:

1) Arrayed experiments confirm robustness of gRNA calling. In **Figure 3** we show the results of an arrayed experiment (i.e., the gRNAs were never pooled), which correlate extremely well with the pooled data. If our gRNA calling method was not robust, we would not see this

level of correlation. This is our strongest evidence that our gRNA calling is accurate and robust.

2) Systematic filtering excludes known sources of noise. In our response to Reviewer 1 who brought up similar concerns (see **Reviewer comment 1.6**) we provide a thorough explanation and justification for each step and threshold involved with gRNA calling. We were conservative when it came to thresholding, thus prioritizing the use of nuclei to which we were able to assign gRNAs with high certainty and discarded nuclei with low confidence gRNA assignments. This was at the cost of increasing the number of non-assigned nuclei (MOI = 0). See our response to **Reviewer comment 1.6** for a full description.

3) Independent experimental observations support single cell findings: Experimentally, our 5' capture protocol increases the number of gRNA positive nuclei and the number of sequenced molecules, when compared directly with a 3' approach (**Extended Data Figure 2**). Improvements in molecule capture increase the gap between true signal and noise and facilitate gRNA calling. Moreover, the tricolor (**Extended Data Figure 1a-d**) revealed a distribution of single, double, and triple infected cells that closely resembles the MOI distribution calculated by our gRNA sequencing data. Our indel analysis (**Figure 2e and Extended Data Figure 6a, previous Extended Data Figure 6c**) experiment also further supports gRNA filtering (see **Reviewer Comment 2.5**) We think that these orthogonal observations strongly support accurate, or at least the new gold standard in, gRNA detection.

4. Our method is at least as robust as that in Jin et al and yields more cells with single perturbations. To clarify how our filtering efforts compare with others, we performed a side-by-side comparison between our gRNA calling strategy and the one described in Jin et al, Science, 2020 (**Rebuttal Figure 8a-b**). It should be considered that while both experiments are performed in vivo, there are large differences between the studies, including the deliver method (lentivirus vs AAV), developmental time of injection (in utero vs adult), and what was being sequenced (cells vs nuclei).

In the Jin et al paper, the authors describe their gRNA filtering strategy as *“if a cell had one perturbation with >1.3x the number of UMIs assigned to it than the next best perturbation based on the 10x sequence, that cell was assigned to that perturbation in the 10x data; otherwise, the cell was declared to have multiple perturbations”*. We were able to replicate the authors' MOI analysis (**Rebuttal Figure 8c left and Jin et al Figure S4.d**). Applying the Jin et al filtering strategy to our data revealed a similar MOI distribution, with a higher proportion of single-infected nuclei in our screen (68% against 52% single infected cells)

(**Rebuttal Figure 8c, right**). These results indicate that our experimental protocol leads to a higher number of single-perturbed cells, but at the cost of more cells with more than one infection, when applying the same (Jin et al) gRNA calling strategy.

While these results sound interesting, we have a few criticisms regarding the filtering strategy used in Jin et al. We believe the filter is not the most appropriate to differentiate between single and multiple infected cells. For instance, if a cell has two gRNAs assign to it with 10 and 14 UMI counts each, using the Jin et al method this cell will be called single infected, which does not seem correct given the high number and similarity of counts for both gRNAs. Our method would label the same cell as having two gRNAs. Using our method, to be considered singly infected it would require >10x more UMIs for one gRNA compared to another (we further justify this and other thresholds for gRNA calling in **Reviewer Comment 1.4**).

We also did the reverse exercise where we applied our filtering strategy to the Jin et al dataset. We found that as much as 60% of all cells in the Jin et al dataset had low-confidence gRNA assignments and were thus labeled as having an MOI of 0 (**Rebuttal Figure 8d, left**). This contrasts with our dataset where we see approximately 30% (**Rebuttal Figure 8d, right**). This observation is also consistent when considering individual cell types (**Rebuttal Figure 8e-f**). We suspect that our improved ability to detect, and thus more robustly call, gRNAs is because transgene expression is higher in AAVs compared to LVs (further elaborated upon in our thorough comparison of AAVs and LVs in **Reviewer Comment 3.5**).

Taken together, we can conclude that our gRNA calling strategy is at least as robust as that described in Jin et al while also yielding more cells with single infections.

5) Non-neuronal cells generally yield fewer UMIs and studying these cell types deeply requires targeted approaches. In our dataset we observed that the number of total UMIs detected in a cell directly correlates with the number of gRNA-associated UMIs (**Rebuttal Figure 8g**). Consequently, as the reviewer points out, the percentage of cells to which we cannot assign a gRNA confidently is higher for non-neuronal cells. However, this is a general trend in the field as it has been consistently reported that non-neuronal brain cells (astrocytes, microglia, and oligodendrocytes) tend to present fewer UMI and gene counts in both scRNA-seq and snRNA-seq²⁶. Therefore, without sequencing an outrageous number of cells or implementing additional experimental steps, any future pooled experiment by us or anyone else will suffer from the same fate. If one is interested in studying less abundant and harder to sequence cell types, we believe it is best to perform a cell type-specific screen. This could be achieved using a cell type-specific delivery approach, cell type-specific promoter, or physical enrichment (FACS, MACS) of specific cell types. In the revised text, we

comment on this concept and provide new data demonstrating a cell type-specific AAV-Perturb-seq experiment in astrocytes (see **Reviewer comment 3.3**).

Taken together, while we recognize the limitations of scRNA-seq and the difficulties of studying certain cell types, we carefully constructed our pipeline to ensure sensitivity and specificity, which we argue sets a new standard in the field. We also demonstrate a new approach to perform AAV-Perturb-seq in seemingly any targeted cell type, using astrocytes as a proof of concept. In sum, we have multiple lines of evidence demonstrating that AAV-Perturb-seq is broadly applicable and robust.

a

Jin et al, Science 2020

-Assign MOI = 1 if the UMI counts of the most abundant gRNA is 1.3x higher than the number of UMIs associated with the second most abundant gRNA.
-Assign MOI > 1 to all cells that don't meet the above requirement

b

Santinha et al

-Remove UMIs below coverage threshold.
-Remove events with 1 UMI count.
-Remove events that contribute less than 10% to the total number of events in a given cell

c

d

e

f

g

Rebuttal Figure 8: Comparison of gRNA filtering strategies. **a.** Filter strategy used by Jin et al in in vivo Perturb-seq. **b.** Filter strategy applied in our study. **c.** Percentage of gRNAs detected per cell (or nucleus) for all cell types combined using the filter strategy presented by Jin et al. **d.** Percentage of gRNAs detected per cell (or nucleus) for all cell types combined using the filter strategy presented in Santinha et al. **e.** MOI for each cell type using the filter strategy presented by Jin et al. Top: dataset from Jin et al. Bottom: dataset from Santinha et al. **f.** MOI for each cell type using the filter strategy presented in Santinha et al. Top: dataset from Jin et al. Bottom: dataset from Santinha et al.

Reviewer comment 2.3: As a minor presentational point, it would be helpful to show histograms of gRNA UMI counts broken down by cell type. The means/medians are not that informative about dropout.

We generated UMI counts broken down by cell type and have implemented it in our analysis of the pooled screen dataset. By visualizing UMI counts distributions per cell type, we show a robust ability to detect gRNAs across different cell types. This figure was added to the revised manuscript (**Extended Data Figure 4c**).

Rebuttal Figure 9: Number of total UMI counts per nucleus across cell types.

Reviewer comment 2.4: Relevance of homozygous null phenotypes to understanding 22q11.2 deletion syndrome

*22q11.2 deletion syndrome is a haploinsufficiency syndrome. In contrast, the CRISPR approach here appears to mostly be generating homozygous nulls based on Extended Data Fig. 6c. (Incidentally, it is unclear to me what the error bars on Fig. 2e and 6c indicate.) Indeed, the gene *Upf1l* is discarded from analysis because the perturbation is overtly toxic on short timescales, suggesting homozygous deletion of an essential gene.*

The goal of the linear modeling is to examine how an additive combination of effects from many genes might lead to the 22q11.2 DS phenotype. Though I think this is an interesting approach, the problem is frankly it is not clear whether the observed phenotypes, which I suspect reflect full knockouts, are the relevant ones to include in this analysis. For example,

some of the single gene deletions in the Perturb-seq experiment appear to have stronger phenotypes on their own (in terms of number of differentially expressed genes) than the entire LgDel heterozygous deletion. And the role of Upf1 cannot be examined at all because it had to be removed.

I think this issue really deserves to be acknowledged in the main text as a challenge for this type of approach and I'd be interested to hear the authors' comments on why this is not a major issue for the interpretation of the later analyses. Did you by chance make any efforts to identify heterozygotes in the experiment?

We understand and acknowledge the concerns raised by reviewers about the difficulties in controlling zygosity in gene editing experiments. This is a widely known and recognized challenge with gene editing in general. The only widely used approaches for controlling zygosity are through deriving single cell clones or transgenic animals, working in haploid cells, or through haploidizing targeted loci. None of these strategies are compatible with high throughput in vivo gene editing experiments. Results from such experiments must therefore be followed by orthogonal analyses and/or experimental validation, which we now include within the revised manuscript using computation and an orthogonal experiment, respectively.

1) Computational stratification of zygosity cell states in perturbed nuclei. Our indel analysis data (**Figure 2e and Extended Data Figure 6a, previous Extended Data Figure 6c**) indicate that a significant number (~80%) of infected cells (*i.e.*, cells expressing Cas9 and a gRNA) carry a mutation in the target gene. Addressing a specific point of the reviewer: the error bars in these plots are independent animals injected with a pure AAV expressing the indicated gRNA. We propose three potential scenarios for gRNA-labelled cells: 1) the cells are infected but not edited and are thus wild-type (WT); 2) the cells are infected and acquire a heterozygous mutation; or 3) the cells are infected and acquire a homozygous mutation. In this context, we are using heterozygous and homozygous as generalized labels that could reflect more complex genotypes. For example, a “heterozygous” cell could be bi-allelically edited but only acquired a loss of function mutation in one allele. For answering the reviewer's question, we think this approach is appropriate, but in the future, it would be interesting to couple the precise cellular genotype to the transcriptional readout. Considering that the nuclei can exist in one of three zygosity states, we can then proceed with stratification.

Given the success of diffusion maps in stratifying cell states, we repurposed the method to attempt to stratify the transcriptional states underpinning changes in zygosity. We extracted gene expression data from control and perturbed nuclei and performed an alignment in

pseudotemporal space using diffusion maps²⁷. We focus here on the arrayed data here because there are more cells/signal for developing a new computational workflow, but in **Reviewer Comment 2.5** we perform a similar analysis on pooled data with similar results. Our analysis of the arrayed data revealed that perturbed nuclei disperse continuously (and without bifurcation) across pseudotime (**Rebuttal Figure 10a-c, new Extended Data Figure 8a-c**), suggesting that the transition from heterozygous to homozygous likely reflects a change in the magnitude of differential expression rather than a change in the identity of differentially expressed genes. While the continuous nature of the data makes it impossible to unambiguously separate zygosity states, we can reasonably stratify the data leveraging the fact that the density plots show an apparent bi-modal distribution (likely capturing heterozygous and homozygous nuclei) (**Rebuttal Figure 10d-i, new Extended Data Figure 8d-i**). Considering these zygosity labels, we found that heterozygous and homozygous groups have undistinguishable expression of the genes previously used to define transcriptional phenotypes and that LFC values calculated using only heterozygous or homozygous nuclei highly correlates to LFC values obtained previously using all perturbed-labelled nuclei (**Rebuttal Figure 10j-o, new Extended Data Figure 8j-o**). These results strongly suggest that heterozygous and homozygous mutations in *Dgcr8*, *Dgcr14*, and *Gnb1l* result in a continuous phenotype where assessment of both or either zygosity captures the impact of the perturbation (and thus are relevant to haploinsufficiency).

2) Modeling haploinsufficiency with CRISPRi. The LgDel model carries a heterozygous deletion in the mouse equivalent of the human 22q11.2 locus (mouse chromosome 16). In **Figure 4c** we show that genes with a deleted copy present a reduction of approximately 50% (LFC = -1) in their transcript levels when compared with wild-type litter control animals. We hypothesized that CRISPRi mediated knock-down may reduce target gene expression to levels observed in a heterozygous situation and thus may be used to simulate the phenotype generated by haploinsufficiency.

To explore this hypothesis, we prepared gRNAs targeting the promoter region of *Dgcr8* and control sequences following previously established rules for the design of CRISPRi gRNAs²⁸. We injected AAV.PHP.B carrying gRNA and hSyn mediated GFP-KASH into a dCas9-KRAB mouse line (JAX stock #030000) and followed our AAV-Perturb-seq workflow (**Rebuttal Figure 11a [new Extended Data Figure 9a] and Extended Data Figure 7a**). Single-nucleus clustering revealed the presence of superficial and deep layer excitatory neurons and interneurons (**Rebuttal Figure 11b, new Extended Data Figure 9b**) similarly to our previous datasets (**Extended Data Figure 7b**). Across cell types, CRISPRi-mediated *Dgcr8* mRNA reduction was comparable to the values observed for LgDel, indicating our ability to model heterozygous expression (**Rebuttal Figure 11c, new Extended Data Figure 9c**). We also confirmed that CRISPRi (KD) and CRISPR (KO) mediated *Dgcr8* perturbations led to

analogous transcriptional phenotypes. Support for this came from: 1) high similarity between LFC values (**Rebuttal Figure 11d, new Extended Data Figure 9d**); 2) genes considered DE by both experiments (**Rebuttal Figure 11e, new Extended Data Figure 9e**); and 3) discovery of DE genes relevant to *Dgcr8*'s known biological function, *i.e.*, genes associated with the process “gene silencing by miRNA” (**Rebuttal Figure 11f, new Extended Data Figure 9f**).

All in all, our additional computational (zygosity stratification with diffusion maps) and experimental (modeling haploinsufficiency with CRISPRi) work provide strong evidence supporting our original conclusions. In the revised manuscript, we have included this additional work, which we believe further increases the value of the paper as it provides a new approach to assessing zygosity-dependent cell states and extends the applicability of AAV-Perturb-seq to CRISPRi technology. In the discussion of the revised manuscript, we also acknowledge that we, along with seemingly all other non-clonal gene editing approaches, cannot perfectly model haploinsufficiency and therefore such results require further validation. Nonetheless, we believe that AAV-Perturb-seq provides a powerful way to prioritize such efforts.

Rebuttal Figure 10 (new Extended Data Figure 8): Computational stratification of zygosity cell states in perturbed nuclei. a-c. Diffusion maps stratification of SH control and perturbed nuclei. **d-e.** Histogram and density plots highlighting the position of SH control and nuclei across the first diffusion component (DC). **g-h.** Stratification of perturbed nuclei into three zygosity states. **j-o.** Expression of DEGs calculated with all perturbed nuclei together (j, l, and n) and correlation of LFC between zygosity states and all nuclei (k, m, and o).

Rebuttal Figure 11 (new Extended Data Figure 9): Modeling haploinsufficiency with CRISPRi. **a.** Schematic representation of arrayed CRISPRi experiments. **b.** UMAP embedding of ~8 000 AAV.PHP.B-infected nuclei isolated from the Cas9-KRAB mouse prefrontal cortex. **c.** LFC values of the *Dgcr8* mRNA between *Dgcr8* perturbation and control nuclei across cell types. **d.** Cosine similarity of gene expression values between *Dgcr8* perturbation and control nuclei across cell types. **e.** Area under the curve (AUC) showing the reliability of differential expression identification (DE) in knock-out (KO, screen profiles) and knock-down (KD, CRISPRi) experiments. False positive and true positive rates comparison of snRNA-seq profiles of *Dgcr8* KO and KD. True positives are DE genes (LFC > 0.4 and FDR < 0.05) identified in the KO experiment. False positives are genes considered DE in the KO dataset, but not in the KD. **f.** LFC values (for both KO and KD experiments) of DEGs identified in the screen KO dataset across cell types.

Reviewer comment 2.5: Bias induced by LDA fitting procedure

In the LDA-based filtering procedure, I understand the logic of discarding perturbed cells that

are essentially “misabeled.” It seems odd, however, to apply the same logic to the control cells, where presumably the label is correct. It might improve the quality of results to only discard the perturbed cells that fail this filter. Indeed, the fact that the computational procedure discards 50% of cells while validation experiments (Extended Data Fig. 6C) suggest most cells are edited seems consistent with the filtering procedure being too conservative and/or emphasizing outlier phenotypes. For example, I’m wondering whether heterozygous deletions are disproportionately discarded, since they will likely have intermediate phenotypes between control cells and full knockouts.

We appreciate the reviewer’s concerns about applying the same discarding logic to control and perturbed nuclei. This was a misunderstanding, because, as the reviewer suggested, we always keep all control nuclei and only discard nuclei with a perturbation label. As shown in **Extended Data Figure 6c (previous Extended Data Figure 6b)**, such a filtering procedure increases the strength of the perturbation-mediated transcriptional signature (magnitude of LFC values) without biasing the overall trend of the signal, which indicates that the filtering maintains perturbation-specific phenotypes. In the revised text, we modified the methods section **Perturbation and nuclei filtering** to clarify the filtering steps.

The reviewer brings up an interesting point about the possibility that the threshold is too stringent, which would enrich for “outlier” phenotypes. However, we believe that our thresholding does a good job balancing noise from over fitting based on the following three points. First, discarding approximately 50% of cells can easily make sense when gene editing rates are approximately 80% (**Figure 2e**), in-frame deletions are observed 33% of the time as expected (**Extended Data Figure 6a, previous Extended Data Figure 6c**), and general noise typical in single cell data. Second, in **Extended Data Figure 6c (previous Extended Data Figure 6b)** we show that removing non-perturbed cells leads to an increase in the magnitude of LFC values without affecting the overall signal as seen by the high correlation of LFC values calculated before and after filtering. Third, as shown in **Reviewer Comment 2.4**, we plotted the nuclei from the pooled screen in a pseudotime alignment calculated with diffusion maps and color-coded nuclei based on LDA filtering results. As seen in **Rebuttal Figure 12**, discarded nuclei overlap strongly with the SH control nuclei, indicating that nuclei with a milder transcriptional response (presumably heterozygous) are not removed by our LDA filtering. Taken together, there is no evidence that our LDA procedure is biasing the results towards outlier phenotypes, whereby our results are in line with expected gene editing outcomes, there is consistency in disrupted transcriptional responses pre and post filtering, and our workflow clearly captures cells spanning the continuous range from control to inferred heterozygous and homozygous cells.

Rebuttal Figure 12: Pseudotemporal alignment of Interneurons nuclei from the pooled screen dataset. For each perturbation, SH control (top, grey), perturbation-labelled and discarded (middle, red), and perturbation-labelled and kept after LDA filtering (bottom, blue).

Reviewer comment 2.6: Details about pseudobulk differential expression procedure

I'm a bit confused about what is used as a replicate in edgeR. The methods section mentions that analysis and pseudo-bulk aggregation for the main Perturb-seq experiment is done within independent "sequencing lanes." Is each "sequencing lane" treated as a replicate? Or is each animal treated as a replicate? How many cells are averaged in each pseudobulk profile? How were independent animals grouped for sequencing and what is the definition of a sequencing lane?

We appreciate the comment and will take the opportunity to address this lack of clarity. A sequencing lane is a single-cell library Chromium 10xGenomics individual lane. As for our experimental methods, we randomly pooled the tissue of 15 male LSL-Cas9 AAV-library injected animals into 3 tissue grinders for nuclei isolation. This allowed us to perform all isolations together which consequently reduces the time between isolation and sorting. Then, we sorted all nuclei together and divided it into 9 individual samples, where each sample was handled independently for single-cell library preparation. Each single cell library was then loaded in an individual single-cell lane, which are processed at the same time and sequenced together. We then use the lane information to create pseudobulk groups of nuclei, which serve as replicates required for edgeR. The replicates in this sense are artificial but biologically independent because each is comprised of a unique pool of nuclei. Our data shows that the number of perturbed nuclei per replicate is higher for neuron types than for non-neuronal cells and is directly correlated with the total number of nuclei sequenced for each cell type (**Rebuttal Figure 13a-b**). We also observed a balanced contribution of each replicate to the total number of nuclei of any given perturbation (**Rebuttal Figure 13c**) without apparent over-representation of nuclei coming from specific lanes.

In the revised text, we clarified this information in the methods section.

Rebuttal Figure 13: Pseudobulk differential expression. **a.** Mean number of nuclei per lane (n=9). **b.** Correlation between the mean number of nuclei per lane and the mean number of nuclei per cell type. **c.** Relative contribution of each lane to the total nuclei count in each perturbation.

Reviewer comment 2.7: I was surprised that there was no discussion of batch effects by animal. If you overlay animal in Fig. 1e is there structure? (Or is this not possible because samples were merged in a way that this information was lost?)

As mentioned in **Reviewer Comment 2.6**, we divided sorted nuclei into 9 single-cell libraries, which we call lanes. We cannot overlay individual animals, but we can do it for individual sequencing lanes. As shown below, we did not detect any strong batch effect between lanes.

Rebuttal Figure 14: Relative representation of cell types in each perturbation.

*Reviewer comment 2.8: Given the attrition from guide calling and filtering, it seems likely some of the analyses are not adequately powered to detect weak effects. It's a practical reality that some hypotheses will not be testable in this challenging in vivo context, but it seems clear comparing Fig. 1e/f to Fig. 2a that the strength of phenotype is at least somewhat proportional to the number of cells in each class. What I'm wondering more is how this affects the ability to detect weaker transcriptional phenotypes even in the well-represented cell types. Given the observed high correlations between transcriptional phenotypes across different neuronal subtypes, have you tried aggregating all neurons to increase power? Do any of the weaker perturbations then reveal phenotypes? It's a bit surprising that genes like *Ranbp1*, which is also essential in many cell types, do not yield phenotypes.*

We fully agree with the reviewer that there are practical limitations to scRNA-seq and this is increasingly complicated in vivo. As the reviewer mentions, this could make it difficult to detect weak effects. However, it is worth noting that the concept that more input (cells in our case) gives you more sensitivity is a general feature of any (omics) method. Nonetheless, it is worth exploring whether the lack of strong transcriptional phenotypes was due to something technical rather than a true biological result. We explored this from multiple angles.

1) In **Reviewer Comments 1.6 and 2.1** we rule out that the lack of transcriptional phenotypes was due to technical features such as the efficiency of a gRNA or gRNA calling thresholds.

2) As the reviewer suggested, we performed an analysis where we increased the number of nuclei per condition by aggregating all neuronal cells. This results in an overall increase in the number of nuclei for each perturbation from 400 to 1300 on average. **Rebuttal Figure 15a** shows the number of DEGs detected for each perturbation in this aggregate analysis. This analysis confirms our original findings (**Figure 2b**) where *Dgcr8*, *Dgcr14*, *Gnb1l*, and *Ufd1l* show a strong transcriptional phenotype. While this aggregate analysis revealed that one additional gene (*Txnrd2*) passed our standard DEG and adjusted p-value threshold, essentially none of the nuclei belonging to this perturbation passed the nuclei filter (**Rebuttal Figure 15b**), indicating it was likely a false positive. Additionally, the LFC values associated with these DEGs presented low magnitude (max 0.18, in contrast to max 0.72 for the other four perturbations) and they were not associated with disrupted biological processes. These results are in line with our cell type specific observations and suggest that the number of nuclei (albeit within the range of nuclei that we have across cell types in our dataset) is not the limiting factor to detect possible weaker phenotype.

3) Another way to assess potential biases due to unequal nuclei numbers is through down sampling-based normalization, where stronger phenotypes that emerged due to over sampling would disappear. We therefore normalized the number of nuclei in each condition by randomly selecting 300 nuclei per perturbation. We chose 300 as it is the size of the smallest group of perturbed neuronal nuclei (*Aifm3* perturbation). Differential expression analysis revealed transcriptional phenotypes only for the 4 significant genes, supporting our previous results obtained with both data divided by cell type (**Figure 2b**) as well as with all neuronal nuclei together (**Rebuttal Figure 15c**).

4) In **Reviewer Comment 2.4**, we show that AAV-Perturb-seq captures a range of transcriptional phenotypes including intermediate (*i.e.*, heterozygous) cell states.

5) Regarding the specific comment of *Ranbp1*, while it has been shown that the gene is essential for mouse development²⁹, published work as well as our own data strongly suggests that this gene is not essential for mouse neurons *in vivo*. In **Figure 2e** and **Extended Data Figure 6a (previous Extended Data Figure 6c)** we show that our gRNA efficiently induces indels in the *Ranbp1* gene (indel efficiency of 80%), ruling out the possibility of a bad gRNA. Animals injected with this *Ranbp1*-targeted gRNA do not show any apparent phenotype up to 5 weeks post-injection. For comparison, we observed that animals injected with gRNA targeting the essential gene *Ufd1l* exhibited apparent motor and/or coordination defects three weeks after injection and had to be sacrificed. Moreover, a recent CRISPR screen cataloging mouse neuronal essential genes did not identify *Ranbp1* as a hit³⁰. In sum,

Ranbp1 may be essential for development but not necessarily for neuronal survival in vivo, and a transcriptional phenotype resulting from Ranbp1 perturbation should not be assumed. To the broader point, this is where we see the value in generating the direct in vivo data rather than assuming results derived from other model systems or experimental setups are universally true.

All in all, while we cannot exclude the possibility that some weak signals are missed, we have shown that AAV-Perturb-seq captures a range of transcriptional phenotypes including intermediate (*i.e.*, heterozygous) cell states as well as avoids major biases due to differences in cell number and other technical factors.

Rebuttal Figure 15: Transcriptional phenotype with all neuron types together. a. Number of DEGs for all perturbations calculated with all neuron types together. Dashed line indicates 5 DEGs with an adjusted p-value (p_{adj}) lower than 0.05. **b.** Percentage of nuclei passing LDA filtering when aggregating all neuron types together. **c.** Number of DEGs for all perturbations calculated with 300 neuronal nuclei per perturbation. Dashed line indicates 5 DEGs with an adjusted p-value (p_{adj}) lower than 0.05.

Minor points

Reviewer comment 2.9: More details on the mouse genotypes and numbers used should be included in the main text as the experiments are described. In some cases such as the

arrayed confirmation experiments it is hard to tell even in the methods how many mice are involved.

We clarified this in the main text and methods section.

Reviewer comment 2.10: It is unclear to me what “edgeR-LTR” is. If you Google this term you get nothing. How is it different from normal edgeR?

We apologize for the error and thank the reviewer for this finding. Indeed, edgeR-LTR was a typo, it is wrong, and it should read edgeR-LRT, which stands for likelihood ratio test.

The edgeR package offers two different statistical tests to discover differential expressed genes: quasi-likelihood F-test (QLFTest) and likelihood ratio test (LRT)³¹. In the context of scRNA-seq, it has been reported that edgeR-LRT outperforms QLFTest and other DE methods at detecting DEGs. These observations informed our choice of using the edgeR-LRT test in our analysis.

The typo has been correct in the main text and methods section.

Reviewer comment 2.11: As a very lightweight analysis, have you looked at the relative representation of each guide across cell type? From 1f they look roughly conserved, but I’m just wondering if any cell-type-dependent differences in representation manifest (as a result of e.g. overt toxicity of any guides).

We have performed the suggested analysis. As the reviewer anticipated, the relative representation of each perturbation across cell types is roughly conserved without any apparent strong changes.

Rebuttal Figure 16: Relative representation of each perturbation across cell types.

Reviewer comment 2.12: “Across all cell types, only genes within the 1.5 Mb deleted locus exhibited a negative LFC, while adjacent genes had no expression changes”: I think this needs to be rephased to state the intended point more clearly. Extended Data 11e shows that there are other downregulated genes.

This was corrected in the main text and now states: “First, focusing on genes previously targeted in our pooled screen, we found that only those within the 1.5 Mb deleted locus exhibited a significant negative LFC, while adjacent genes had minimal expression changes (**Figure 4c**). This observation was consistent across cell types and confirms that locus heterozygosity leads to approximately 50% reduction in the expression of affected genes”.

Reviewer comment 2.13: A silly comment: given how targeted the library is and the citation to the literature, the “Remarkably” comment on pg. 8 about the 4 genes with phenotypes lying within the minimal region is perhaps unnecessary

Reviewer 1 had the same comment and we reproduce our response here: While we believe the result is remarkable, because of how it makes sense and further supports the findings, we can agree that the word may be overly sensational and have therefore removed it from the text.

Referee #3 (Remarks to the Author):

Santinha and colleagues use in vivo pooled CRISPR screens with single-cell transcriptomics to identify key genes and cell types implicated in 22q11.2 DS. Compared to prior approaches (e.g. in vivo perturb-seq from Xin et al., PMID: 33243861), this work expands in two important areas: 1) AAV-mediated perturbations (which is superior to lentivirus for many in vivo applications) and 2) improved single-cell transcript and guide capture using 5' chemistry.

Using this new platform, the group identifies 4 key genes driving 22q11.2, the relevant developmental timepoints, and functional deficits in RNA processing. Overall, this paper is exciting for both the novel biological insights into 22q11.2 DS and from a technical standpoint with single-cell in vivo CRISPR screens. This new platform should be of broad applicability to mouse models of disease and identification of disease mechanisms. For this second reason in particular, I am supportive of its publication in a top-tier journal. However, I

have a few minor comments that I believe should be addressed prior to publication:

We thank the reviewer for sharing our enthusiasm about the power of single-cell in vivo CRISPR screens and how this is broadly enabled with the use of AAV vectors. We appreciate the positive feedback regarding the novelty of our work and the potential impact of this new platform for interrogating disease mechanisms.

*Reviewer comment 3.1: * I commend the authors on the very careful validation studies (using individual AAVs in separate mice). The indel data (Cas9 editing) is very convincing. However, it seems like a bit of a missed opportunity to delve deeper into the RNA processing/splicing defects. For example, the authors mention “a disrupted balance between RNA transcription and splicing resulting from perturbation of Dgcr14”. This should be demonstrated beyond DEG and GO analyses. Unfortunately, single-cell/nucleus RNA-seq does not provide splicing data. Might the authors instead use long-read sequencing? It would be helpful to understand whether Dgcr14 loss causes widespread changes in splicing (i.e. all transcripts) or whether there is a specific enrichment for long transcripts, neuron/synapse transcripts, etc.*

We agree with the reviewer that exploring splicing defects upon perturbation of *Dgcr14* is an exciting avenue. Below we dive into our experimental and computational efforts to do so.

As the reviewer points out, as scRNA-seq techniques capture only 5' or 3' mRNA regions, it does not provide sufficient information for performing splicing analysis. However, bulk RNA-seq protocols are able to sequence the entire transcript and thus provide sufficient information for splicing analysis³². This, together with the fact that *Dgcr14* perturbation leads to similar transcriptional phenotypes across neuron types, opens the possibility to deliver a *Dgcr14*-targeting gRNA (the same high efficiency gRNA used in the arrayed experiments) together with hSyn (to enrich for neurons), sort nuclei to enrich for perturbed cells, and perform bulk RNA-seq. Thus, we repeated the same experimental approach used in the arrayed perturbations experiments (**Extended Data Figure 7a and Rebuttal Figure 17a-b**) and performed bulk RNA-seq of sorted neuronal nuclei.

We conducted alternative splicing analysis on 6 nuclear RNA-seq samples, 3 animals injected with *Dgcr14*-targeting gRNA and 3 control animals injected with safe harbor-targeted gRNAs. We used STAR 2.7.10b for alignment followed by ASpli 2.6.0³², an event-based approach to differential alternative splicing (AS) analysis. Both methods were chosen for their superior performance in recent benchmarks³³. Although the analysis resulted in a list of hundreds of potential hits (**Rebuttal Table 3**), a more thorough examination of these hits revealed that the presence of a large proportion of nascent pre-mRNA transcripts in nuclear

RNA-seq samples confounds the accurate computational categorization and detection of AS events. In fact, as we use nuclei rather than all cells, most mRNAs are still in their nascent form and do not contribute to the splicing signal. The presence of a high level fluctuating baseline component of read coverage along the entire range of the locus originating from pre-mRNA transcripts leads the model to incorrectly call the local fluctuation in bin coverage signal between the conditions as true difference. Although there are few instances in which visual inspection suggests the existence of a true observed difference in the splicing signal between the conditions, most plots randomly examined failed to show a true signal.

We would like to point out that to the extent of our knowledge, the performance of other AS splicing methods in general would suffer from this same issue given that if, for example, we include the entire pre-mRNA as an additional transcript (one large exon) in the annotation, then we render the inference of the alternative splicing event from bin coverage signal or junction reads an ill-posed problem. It is worth noting that long read sequencing would not solve this. Taken together, even though we find hundreds of hits with some meaningful information (**Rebuttal Figure 17c-e**), given the caveats and limitations of splicing analysis performed on nuclei in general, we do not feel confident publishing these findings at this time.

With further work it may be possible to overcome these limitations. Given that adult mouse brain tissue requires the isolation of nuclei rather than cells, we believe that an experimental design to look at splicing events in adult animals needs to take the high levels of pre-mRNA into account when sequencing RNA from nuclei. This work would require the development of a new computational method. Alternatively, new experiments could be performed in a developmental context where whole cells can be isolated from very young mice. This would either require the development and use of genetically engineered (*Dgcr14*^{+/-}) mouse models, or an alternative methodology where gRNA is delivered intracerebroventricularly in utero or in neonates so that perturbations happen well before whole single cells are isolated.

To conclude, while we are enthusiast about pursuing this avenue, initial efforts reveal disappointing limitations to current AS methods that impede us to pursue such experiments in the time frame and scope of our current work. Possible experimental approaches to circumvent these drawbacks would take many of months of implementation to yield data that we feel confident with.

Rebuttal Table 3: Top 20 AS events (IR = intron retention; ES = exon skipping).

gene_name	event	logFC	fdr	regulation	width
Eif5b	IR	-1.886	8.48E-51	down	57570
Abcc8	IR	1.765	1.45E-35	up	75511
Mpp3	IR	1.528	1.70E-26	up	28810
Rack1	IR	-1.021	4.92E-22	down	6103
Sod2	IR	-1.983	1.41E-21	down	33218
Iqgap1	IR	2.497	2.03E-21	up	114392
Cadm2	ES	-1.798	1.58E-13	down	965488
Arsa	IR	-1.51	2.85E-13	down	4950
Fam133b	IR	-1.223	4.77E-12	down	26406
Mrpl28	IR	-1.729	2.29E-10	down	3114
Tmed4	IR	-0.8303	7.26E-09	down	4502
Sh2b3	IR	-1.558	2.21E-08	down	22159
Dock9	ES	-1.774	7.48E-08	down	255792
Ptk2	ES	-0.7972	8.20E-08	down	218179
Gne	IR	5.163	8.81E-08	up	50103
Zfp280d	ES	2.93	6.85E-07	up	88918
Kcnma1	ES	-1.915	8.41E-07	down	725061
Macroh2a1	ES	-0.7044	2.14E-06	down	62743
Arhgef2	IR	-0.6122	3.43E-06	down	42087

Rebuttal Figure 17: RNA splicing in Dgcr14-perturbed nuclei. a. Schematics of AAV genome used in the splicing experiment. **b.** Schematic representation of the splicing experiment. **c.** Relative proportion of significant alternative splicing events (FDR < 0.05, 157 events). **d.** Events up and down regulated for each splicing event category. **e.** Gene length of genes with detected alternative splicing (AS) events and genes with no detectable event.

*Reviewer comment 3.2: * The authors find that the phenotype is not really developmentally-driven but rather an issue with mature neurons. Would it be possible in the LgDel mouse to rescue using the 4 genes (or CRISPR activators)? This would provide the most convincing demonstration that the 4 identified genes are sufficient for treating the disease (and outline a gene therapeutic approach for future human clinical studies). This may be beyond the scope of the paper (and thus should not be required for publication) but would certainly be quite exciting.*

Thank you for your comment and suggestion for this potential follow-up experiment. We agree that performing rescue experiments in the LgDel mouse model would be an extremely

exciting direction to pursue and could potentially provide valuable therapeutic insights. We do believe that CRISPRa-based multiplexed activation of the 4 identified genes in the disease model mouse would be possible (from a gene editing perspective) and a great way to assess phenotypic rescue. However, as you mentioned, we also believe that this experiment is beyond the scope of the current manuscript. In the future, we hope to provide these results in another study.

*Reviewer comment 3.3: * The authors should comment further on the gene Pi4ka, which seems to meet their significance criterion in astrocytes (Fig 2b). The other genes seem to have neuron-focused disruptions so this seems like it might be of interest and perhaps functioning via distinct mechanisms.*

While the schematic makes it look like Pi4ka is close to the threshold, it falls below it, and we therefore did not investigate this in the original manuscript. However, we agree with the reviewer that there may be some signal worth exploring deeper, and the possibility of identifying distinct cellular mechanisms is certainly intriguing. To address this thoroughly, as well as expand the utility of AAV-Perturb-seq, we devised an astrocyte-specific AAV-Perturb-seq approach that should provide a rich dataset to investigate astrocyte-specific mechanisms.

An in vivo cell type-specific screen could in principle be achieved using cell type-specific delivery or expression as well as through physical enrichment of the cell type of interest. We chose to use a cell type-specific promoter that should lead to higher levels of expression of eGFP-KASH in astrocytes, thus allowing for enrichment of infected astrocyte nuclei using FACS. In a new set of experiments, we swapped the CBh promoter in the AAV genome for GfaABC1D, a promoter that was documented to have high levels of expression in astrocytes (**Rebuttal Figure 18a**). We cloned our original gRNA library used in the unbiased screen (CBh screen), produced AAV.PHP.B and followed AAV-Perturb-seq's experimental pipeline (**Rebuttal Figure 18b**). Single-nucleus sequencing of ~ 35 000 infected cells revealed the presence of astrocytes as well as other brain cells (**Rebuttal Figure 18c**). Although we expected more of an enrichment in astrocytes based on prior published findings describing the GfaABC1D promoter³⁴, the presence of other cell types is not abnormal when using such promoters. For example, it was shown previously that GfaABC1D can also be expressed in neurons³⁵. Nevertheless, when looking at cell type composition in the datasets, we observed that 17% of all cells are astrocytes in the GfaABC1D experiment, as opposed to 6% in the CBh screen (**Rebuttal Figure 18d**), resulting in a sufficient number of nuclei to proceed with an astrocyte-focused AAV-Perturb-seq analysis.

Directing our attention to astrocytes, we detected a gRNA in 80% of nuclei, with ~45% expressing a unique gRNA (**Rebuttal Figure 18e**). This was comparable to our previous results for the CBh screen shown in **Extended Data Figure 4a**. Analysis of cells with MOI 1 (without filtering unperturbed cells) indicated the presence of a transcriptional phenotype in astrocytes perturbed in *Dgcr8* (3 DEGs) and *Dgcr14* (16 DEGs) (**Rebuttal Figure 18f**). These findings confirm our original result indicating that the perturbed genes do not elicit an overtly strong transcriptional response in astrocytes.

Although the number of DEGs for *Dgcr8*-perturbed astrocytes did not meet our standards for LDA filtering and signal enrichment, and therefore we could not stratify perturbed and non-perturbed cells, we thought we may be able to proceed with unfiltered data. As noted previously, the transcriptional signature before and after filtering tends to correlate (**Extended Data Figure 6c, previous Extended Data Figure 6b**), making this approach sensible with the caveat of increased noise due to the inclusion of unperturbed cells. Considering this approach, we were curious to investigate whether the transcriptional phenotype observed in astrocytes was similar to our previous results in neurons, and therefore calculated the correlation between astrocytes and neuron types perturbed in *Dgcr8* and *Dgcr14*. We demonstrate that astrocytes have a perturbation-specific transcriptional signal that is comparable with our previous observations in neuron types. Support for this comes from high perturbation-specific correlation (**Rebuttal Figure 18g**) and overlap of DEGs (**Rebuttal Figure 18h**) between cell-type pseudobulk profiles. Overall, the combination of the results from the unbiased CBh screen and the astrocyte specific GfaABC1D screen suggests that mutations in *Dgcr8* and *Dgcr14* lead to perturbation-specific transcriptional phenotypes that are shared across brain cells beyond neuron types.

While this new experiment and analysis is of a sufficient quality to be published, because of the lack of expected astrocyte specificity of the GfaABC1D promoter and the corresponding poor astrocyte enrichment in our data, we are not thrilled with the outcome (and propagating the idea that the promoter is cell type specific) and would therefore have not included this work in the revised manuscript. In the future, it would be valuable to perform additional experiments with alternative cell type specific promoters, but we believe this goes beyond the timeline and expectations of the current manuscript. In sum, setting aside the lack of reproducibility of prior published findings regarding GfaABC1D, we have shown that focusing AAV-Perturb-seq on less abundant cell types is in principle possible.

As we believe the results of this experiment may be relevant to others designing cell type-specific experiments, we included this answer in the **Supplementary Note 4**.

Rebuttal Figure 18: Astrocyte-specific pooled screen. a. Schematic representation of the AAV genome engineered to express. **b.** Schematic representation of arrayed CRISPRi experiments. **c.** UMAP embedding of ~35 000 AAV.PHP.B-infected nuclei isolated from the mouse prefrontal cortex. **d.** Abundance of cell types in single-nucleus datasets generated from brain cells infected with CBh and GfaABC1D AAVs. **e.** Percentage of gRNAs detected

per nucleus in Astrocytes. **f.** Number of DEGs for all perturbations in Astrocytes. **g.** Pearson correlation of transcriptional profiles induced by different perturbations across neuron types and Astrocytes. **h.** Heatmap of DEGs (columns) in *Dgcr8* and *Dgcr14* perturbations in neuron types and Astrocytes (rows).

*Reviewer comment 3.4: * The technique is misnamed. Perturb-seq is a 3'-based capture method. In this work, the authors specifically test 3' and 5' methods head-to-head and adopt the 5' method (analogous to Mimitou et al., reference 17). Given that, and to avoid confusion, please rename the method AAV-ECCITE-seq, as ECCITE-seq (ref 17) is the first work to describe the 5' capture strategy used.*

We understand the reviewer's comment, which is correct. Indeed 5' capturing was first introduced in ECCITE-seq³⁶ and Replogle et al (the authors didn't name the technology)⁸. We named our method AAV-Perturb-seq for two main reasons.

1) ECCITE-seq is usually associated with protein quantification. ECCITE-seq was an iteration from CITE-seq, a method created to perform multi-modal RNA and protein quantification from the same cell. While the method introduced 5' gRNA capturing, it is more often associated with protein quantification experiments. By naming our method AAV-ECCITE-seq we could mislead reader to believe that our method supports protein quantification.

2) Perturb-seq is commonly used when referring to single cell CRISPR screening experiments. While there were three single-cell CRISPR screening papers published at approximately the same time (Dixit, et al. (Perturb-seq)⁹, Datlinger, et al. (CROP-seq)¹⁷ Jaitin, et al. (CRISP-seq)¹⁸), for whatever reason they are now most commonly referred to as Perturb-seq experiments in the literature. We do not necessarily think that this is fair given the well-known errors in the Perturb-seq paper¹⁹ and that the vector and methods in the CROP-seq paper are the most commonly used today. Nonetheless, because the field has decided that this is how they want to refer to single cell CRISPR screening experiments, we thought it was best to go along with convention.

While our very strong preference is to name our approach AAV-Perturb-seq for the reasons above, if all of the reviewers and the editor think another name is more appropriate (eg AAV-ECCITE-seq) we would be willing to change it. Apart from naming the method, we agree that it is important to acknowledge the lineage of 5' capture innovation and have clarified this in the revised text.

*Reviewer comment 3.5: * The introduction mentions "a very narrow range of developmental time points, tissues, and cell types conducive to lentiviral infection in vivo." However, no*

citations are provided for this statement. I thought the primary use of AAV was superior spread (targeting more cells in the brain) than lenti. The authors should provide citations to support these claims — especially “narrow range of developmental time points”.

We thank the reviewer for pointing this out as it tells us that we were taking for granted how much readers would appreciate the numerous advantages of AAV compared to lentivirus (LV). Several of the reviewers commented on this as well. Considering contemporary research, AAV is the vastly preferred modality for in vivo delivery for several very good reasons. These advantages, as well as disadvantages, are thoroughly covered by many recent reviews on in vivo delivery^{37,38}. Below, we highlight the main advantages and disadvantages relevant for single cell CRISPR screening.

The main advantages of AAV over LV are as follows:

1) Unlike LV, AAVs can be injected systemically and infect seemingly any organ and cell type in a tunable way. If LV is injected systemically, it is only capable of infecting a few cells in the liver³⁹. If LV is injected within a compartment (*e.g.*, via intraperitoneal, intrathecal, or intracerebroventricular injection) it mostly infects cells along the barrier without penetrating deeply into tissue⁴⁰. Thus, in vivo delivery of LV almost always requires direct injection into the organ of interest.

When injected systemically, AAVs can infect almost any tissue or cell type in a tunable way. Unlike LV, which is difficult to modify and target to specific cell types, AAV is very easy to modify and preferentially target to specific cell types thanks to natural serotypes and engineered/evolved capsid variants. If natural AAV serotypes are injected systemically, they show unique (*i.e.*, tunable) biodistributions⁴¹. AAV capsid proteins can further be engineered or evolved to enable the preferential targeting of (new) cell types of interest or passing of physiological barriers such as the blood brain barrier. In our study, we leveraged the evolved AAV capsid PHP.B⁴², which enabled us to achieve brain-wide infection with a simple systemic (tail vein) injection.

2) Unlike LV, AAVs do not generally require direct injection into tissue, thus avoiding the need for surgical procedures and complicated ethics approvals. Direct injection into tissue often requires a surgical procedure, which in the case of brain delivery is a craniotomy. The drawbacks of this are plentiful. Surgery, compared to systemic delivery, is labor intensive, requires expert knowledge, low throughput, leads to increased mortality in experimental mice, requires increased post-surgical monitoring, and involves a more elaborate ethics approval process (due to increase severity and stress to the animals).

3) Unlike LV, AAVs do not generally require direct injection into tissue, thus avoiding tissue damage and confounding alterations to cell states. Direct injection into tissue causes damage, resulting in altered cell states. For example, along the injection track created during cranial/brain injections, it is extremely common to find reactive astrocytes and activated microglia⁴³, which can confound phenotypes of interest especially when using single cell methods.

4) Unlike LV, AAV sparsely infects a large number of cells across a tissue. With direct injection of a virus into tissue, it is extremely difficult if not impossible to infect a large number of cells while controlling the MOI. Systemically injecting AAV made it simple for us to titrate the amount of virus to optimize the number of infected cells with a single infection (**Extended Data Figure 1**).

5) Transgene expression from AAV is known to be higher compared to LV, likely leading to higher rates of gRNA detection. While we did not test AAV and LV head-to-head for gRNA capture, it is well known that AAV is superior to LV in terms of transgene expression⁴⁴. We speculate that this contributes to our gRNA capture/calling rates being higher than what has been described previously with LV¹⁴.

6) LV has a narrow range of utility. Beyond what has already been discussed above, the only case that we are aware of where delivering LV in vivo is commonly done and useful is in the context of development where LV is injected before or shortly after birth. When LV is injected in utero or postnatally in the ventricle of the brain, it is possible to achieve brain-wide infection. Intracerebral ventricular injection into neonates can lead to brain-wide infection due to the fact that the blood brain barrier is immature. In utero injection into the brain of embryos within a pregnant mother can also lead to brain-wide infection due to radial glial progenitor cells of the ventricle wall being readily infected and differentiating to give rise to transgene-expressing daughter cells throughout the brain. This approach was used in the Jin et al paper. However, the concern with such an approach is that large gRNA libraries could be bottlenecked due to only a small number of cells being amenable to infection. This outcome may not be apparent when examining the number of infected cells at the endpoint because these could represent daughters of the originally infected progenitor cells. In contrast to these developmental perspectives, it is not possible to achieve brain-wide infection with lentivirus. Taken together, while there is some utility of LV we think it is fair to say that the diversity of options is severely limited.

The main disadvantages of AAV over LV are as follows:

1) While the packaging capacity of AAV (~4.7kb) is smaller than that of LV (~10kb), dual/split vector approaches offer efficient mitigation strategies. The main limitation of AAV over LV is packaging capacity. However, this can be overcome by using dual/split vector systems⁴. The concept here is that payloads can be split over multiple viruses, achieved either by splitting independent elements (eg, gRNA from Cas9) or using strategies that stitch (via split proteins, trans-splicing, split-inteins) the biomolecules (eg Cas proteins) back together again. These methods are widely used in the context of gene editing and other fields^{2,3}. While this is not necessary in our experimental setup where we are using a Cas9 transgenic animal, it should be straightforward to combine our approach with split vectors to enable AAV-Perturb-seq in non non-Cas9 transgenic animals (see **Reviewers Comment 1.3**).

Taken together, AAV is vastly superior to LV for in vivo delivery and AAV-Perturb-seq will therefore open new avenues for single cell CRISPR screening in vivo. In the revised text, we discuss these advantages and disadvantages as well as reference several recent reviews that fully compare the technologies. In addition, we also include this comparison in the **Supplementary Note 1**.

REFERENCES

1. Ran, F. A. *et al.* In vivo genome editing using *Staphylococcus aureus* Cas9. *Nature* **520**, 186–191 (2015).
2. Yang, Y. *et al.* A dual AAV system enables the Cas9-mediated correction of a metabolic liver disease in newborn mice. *Nat. Biotechnol.* **34**, 334–338 (2016).
3. Koblan, L. W. *et al.* In vivo base editing rescues Hutchinson–Gilford progeria syndrome in mice. *Nature* **589**, 608–614 (2021).
4. Lai, Y. *et al.* Efficient in vivo gene expression by trans-splicing adeno-associated viral vectors. *Nat. Biotechnol.* **23**, 1435–1439 (2005).
5. Chew, W. L. *et al.* A multifunctional AAV–CRISPR–Cas9 and its host response. *Nat. Methods* **13**, 868–874 (2016).

6. Adamson, B. *et al.* A Multiplexed Single-Cell CRISPR Screening Platform Enables Systematic Dissection of the Unfolded Protein Response. *Cell* **167**, 1867-1882.e21 (2016).
7. Norman, T. M. *et al.* Exploring genetic interaction manifolds constructed from rich single-cell phenotypes. *Science* **365**, 786–793 (2019).
8. Replogle, J. M. *et al.* Combinatorial single-cell CRISPR screens by direct guide RNA capture and targeted sequencing. *Nat. Biotechnol.* **2020 388 38**, 954–961 (2020).
9. Dixit, A. *et al.* Perturb-seq: Dissecting molecular circuits with scalable single cell RNA profiling of pooled genetic screens. *Cell* **167**, 1853 (2016).
10. Duan, B. *et al.* Model-based understanding of single-cell CRISPR screening. *Nat. Commun.* **10**, 2233 (2019).
11. Tian, R. *et al.* Genome-wide CRISPRi/a screens in human neurons link lysosomal failure to ferroptosis. *Nat. Neurosci.* **24**, 1020–1034 (2021).
12. Squair, J. W. *et al.* Confronting false discoveries in single-cell differential expression. *Nat. Commun.* **2021 121 12**, 1–15 (2021).
13. Ursu, O. *et al.* Massively parallel phenotyping of coding variants in cancer with Perturb-seq. *Nat. Biotechnol.* **40**, 896–905 (2022).
14. Jin, X. *et al.* In vivo Perturb-Seq reveals neuronal and glial abnormalities associated with autism risk genes. *Science* **370**, eaaz6063 (2020).
15. Papalexis, E. *et al.* Characterizing the molecular regulation of inhibitory immune checkpoints with multimodal single-cell screens. *Nat. Genet.* **53**, 322–331 (2021).
16. Frangieh, C. J. *et al.* Multimodal pooled Perturb-CITE-seq screens in patient models define mechanisms of cancer immune evasion. *Nat. Genet.* **2021 533 53**, 332–341 (2021).

17. Datlinger, P. *et al.* Pooled CRISPR screening with single-cell transcriptome readout. *Nat. Methods* 2017 143 **14**, 297–301 (2017).
18. Jaitin, D. A. *et al.* Dissecting Immune Circuits by Linking CRISPR-Pooled Screens with Single-Cell RNA-Seq. *Cell* **167**, 1883-1896.e15 (2016).
19. Hill, A. J. *et al.* On the design of CRISPR-based single-cell molecular screens. *Nat. Methods* **15**, 271–274 (2018).
20. Hsu, P. D. *et al.* DNA targeting specificity of RNA-guided Cas9 nucleases. *Nat. Biotechnol.* **31**, 827–832 (2013).
21. Knapp, D. J. H. F. *et al.* Decoupling tRNA promoter and processing activities enables specific Pol-II Cas9 guide RNA expression. *Nat. Commun.* **10**, 1490 (2019).
22. Stuart, T. *et al.* Comprehensive Integration of Single-Cell Data. *Cell* **177**, 1888-1902.e21 (2019).
23. Yang, P., Huang, H. & Liu, C. Feature selection revisited in the single-cell era. *Genome Biol.* **22**, 321 (2021).
24. Luecken, M. D. & Theis, F. J. Current best practices in single-cell RNA-seq analysis: a tutorial. *Mol. Syst. Biol.* **15**, e8746 (2019).
25. Armand, E. J., Li, J., Xie, F., Luo, C. & Mukamel, E. A. Single-Cell Sequencing of Brain Cell Transcriptomes and Epigenomes. *Neuron* **109**, 11–26 (2021).
26. Habib, N. *et al.* Massively parallel single-nucleus RNA-seq with DroNc-seq. *Nat. Methods* **14**, 955–958 (2017).
27. Haghverdi, L., Buettner, F. & Theis, F. J. Diffusion maps for high-dimensional single-cell analysis of differentiation data. *Bioinformatics* **31**, 2989–2998 (2015).
28. Gilbert, L. A. *et al.* Genome-Scale CRISPR-Mediated Control of Gene Repression and Activation. *Cell* **159**, 647–661 (2014).

29. Meechan, D. W. *et al.* Modeling a model: Mouse genetics, 22q11.2 Deletion Syndrome, and disorders of cortical circuit development. *Prog. Neurobiol.* **130**, 1–28 (2015).
30. Wertz, M. H. *et al.* Genome-wide In Vivo CNS Screening Identifies Genes that Modify CNS Neuronal Survival and mHTT Toxicity. *Neuron* **106**, 76–89.e8 (2020).
31. Robinson, M. D., McCarthy, D. J. & Smyth, G. K. edgeR: a Bioconductor package for differential expression analysis of digital gene expression data. *Bioinforma. Oxf. Engl.* **26**, 139–140 (2010).
32. Mancini, E., Rabinovich, A., Iserete, J., Yanovsky, M. & Chernomoretz, A. ASpli: integrative analysis of splicing landscapes through RNA-Seq assays. *Bioinformatics* **37**, 2609–2616 (2021).
33. Fenn, A. *et al.* Alternative splicing analysis benchmark with DICAST. 2022.01.05.475067 Preprint at <https://doi.org/10.1101/2022.01.05.475067> (2022).
34. Griffin, J. M. *et al.* Astrocyte-selective AAV gene therapy through the endogenous GFAP promoter results in robust transduction in the rat spinal cord following injury. *Gene Ther.* **26**, 198–210 (2019).
35. Taschenberger, G., Tereshchenko, J. & Kügler, S. A MicroRNA124 Target Sequence Restores Astrocyte Specificity of gfaABC1D-Driven Transgene Expression in AAV-Mediated Gene Transfer. *Mol. Ther. - Nucleic Acids* **8**, 13–25 (2017).
36. Mimitou, E. P. *et al.* Multiplexed detection of proteins, transcriptomes, clonotypes and CRISPR perturbations in single cells. *Nat. Methods* **16**, 409–412 (2019).
37. Asokan, A., Schaffer, D. V. & Jude Samulski, R. The AAV Vector Toolkit: Poised at the Clinical Crossroads. *Mol. Ther.* **20**, 699–708 (2012).
38. Mingozzi, F. & High, K. A. Therapeutic in vivo gene transfer for genetic disease using AAV: progress and challenges. *Nat. Rev. Genet.* **12**, 341–355 (2011).

39. Wilson, R. C. & Gilbert, L. A. The Promise and Challenge of In Vivo Delivery for Genome Therapeutics. *ACS Chem. Biol.* **13**, 376–382 (2018).
40. Artegiani, B. & Calegari, F. Lentiviruses allow widespread and conditional manipulation of gene expression in the developing mouse brain. *Development* **140**, 2818–2822 (2013).
41. Wang, D., Zhang, F. & Gao, G. CRISPR-Based Therapeutic Genome Editing: Strategies and In Vivo Delivery by AAV Vectors. *Cell* **181**, 136–150 (2020).
42. Chan, K. Y. *et al.* Engineered AAVs for efficient noninvasive gene delivery to the central and peripheral nervous systems. *Nat. Neurosci.* **20**, 1172–1179 (2017).
43. Casanova, F., Carney, P. R. & Sarntinoranont, M. Effect of Needle Insertion Speed on Tissue Injury, Stress, and Backflow Distribution for Convection-Enhanced Delivery in the Rat Brain. *PLOS ONE* **9**, e94919 (2014).
44. Hughes, S. *et al.* Lentiviral vectors as tools to understand central nervous system biology in mammalian model organisms. *Front. Mol. Neurosci.* **8**, (2015).

Reviewer Reports on the First Revision:

Referees' comments:

Referee #1 (Remarks to the Author):

We are generally satisfied and happy with the revisions. The data is a bit sparse in areas (i.e. 1 UMI per gRNA cutoff seems arbitrary and maybe your real UMIs are at a higher cutoff just from looking at the graphs, but it's honestly hard to tell because individual gRNA UMI counts are generally low). But the fact that your arrayed single guide validation data lines up very well with the pooled data supports the gRNA assignments. You did make changes to the text where we suggested which is appreciated. It is a very impressive and exciting piece of work!

Referee #2 (Remarks to the Author):

The manuscript remains solid, and the authors have clearly attempted to earnestly respond to reviewer comments. I am happy with most of their responses. In particular, I think the more direct engagement with the challenges of studying heterozygous mutations in the revised main text substantially improves the presentation of the paper and homes in on a key challenge for the field moving forward. I also think the authors have made a decent case that they set a new standard for in vivo studies, though challenges remain.

My only real reservation concerns the supplementary notes that have been introduced, which don't meet the quality standard set by the rest of the paper.

Points:

1) I do think the sample multiplexing strategy in which cells from distinct animals were randomly mixed potentially obscures a major confounder, but obviously this can't be fixed after the fact. Just a point worth remembering for future studies.

2) Though I appreciate the insight into the authors' thinking, the new Supplementary Notes are unpolished, in places quite speculative, and seem out of place in a scientific paper in their current form. E.g. "AAV is vastly superior to LV" is not scientific, nor is referring to "truly differentially expressed genes" when discussing the output of your computational pipeline relative to others. Some example points:

1: I am not an expert in this world but this would really need to be reviewed for accuracy if included. When I suggested providing background for a general audience I meant a few sentences summarizing the main advantages in the introduction of the paper with some references.

2/3: These seem more at home in methods or as traditional supplemental figures.

4: There are claims here that I consider open to debate, and some speculative comparisons to approaches in other studies. If you really want to go down this road, then you must do side-by-side comparisons *in multiple contexts* and include the analyses in the main text. (I do not think this is essential for this paper at all.) I think your main point is covered by reference to Squair et al., plus

Supplementary Note Figure 4 could be a supplemental figure.

6: Completely speculative. Delete or place the points in discussion.

7: Why is this not in main text if the point is important? It's not cited at all.

Overall I don't see why these notes are needed. Honestly I'd prefer that the authors just remove them, or shorten them substantially to focus on what they did (rather than what they might do or their opinions on what other people did). If the authors think it is essential to include them I can do another round of review to at least provide feedback on some of the points that I think are overstated.

Referee #3 (Remarks to the Author):

The authors have done an excellent job addressing review comments in-depth.

As I mentioned in my initial review, I still feel that AAV-Perturb-Seq is a misnomer and that the technique (which uses ECCITE-seq to do 5' direct capture) should instead be called AAV-ECCITE-seq to avoid further confusion in an already confusing field of CRISPR perturbomics techniques.

Referee #4 (Remarks to the Author):

For the authors:

In their revised manuscript entitled "AAV-mediated single-nucleus CRISPR screening enables functional dissection of the 22q11.2 deletion syndrome in vivo", Santinha et al apply an in vivo single-cell CRISPR screen, termed AAV-Perturb-seq, to systematically dissect the phenotypic landscape underlying 22q11.2 deletion syndrome genes in the adult mouse brain prefrontal cortex. They identified three 22q11.2-linked genes that explain approximately 40% of the transcriptional changes observed in a 22q11.2 deletion mouse model, suggesting that active disruption in adult neurons, rather than exclusively developmental process, dysregulates RNA processing and synaptic function in 22q11.2 deletion syndrome. This is an innovative paper and the first to apply pooled CRISPR screening to dissect the genes causally related to a highly penetrant psychiatric CNV in post-mitotic neurons. The authors systematically include a number of thoughtful empirical validations of key gene perturbation signatures. With further characterization of the downstream mechanisms linking the top three 22q11.2 genes, this manuscript could be suitable for publication in Nature.

Major Concerns:

1. Pooled Cas9 CRISPR screening in vivo of 22q11.2 genes (Fig. 1-4): The authors validated in vivo titration of AAV tail vein injection to independently express either mTagBFP, Venus, or mCherry under the control of a ubiquitous CBh promoter. The higher viral dose led to an increased percentage of total infected nuclei (~ 34%), but almost half received more than one AAV particle. Thus, the authors selected the dose of 5.0×10^9 AAV particles per animal to maximize the total

number of cells infected with a single AAV (Fig. 1). They also selected a 5'-based capture approach, combining independent gRNA expression (pAS088) with 5' capture sequencing, as it more sensitively capture mRNA and gRNA information in AAV-infected nuclei (Fig. 2). Their bioinformatic approach was extremely rigorous, requiring not just detection of the gRNA, but also confirmation of significantly perturbed expression of the target gene in individual cells. For the 29 of the 37 genes in the 22q11.2 locus expressed in adult mouse cortex, the authors designed two independent gRNAs and included five control gRNAs targeting mouse safe-harbor (SH) loci. They detected all gRNAs in the library, with an average of ~10 nuclei in microglia to 400 nuclei in interneurons; in total, they detected ~60,000 nuclei spanning 6 brain cell types and perturbation of all 22q11.2 genes expressed in the adult prefrontal cortex. Using a pseudobulk approach, the authors found significant transcriptional phenotypes in four gene perturbations across all neuron types (Dgcr8, Dgcr14, Gnb1l, and Ufd1l), as measured by the number of DEGs; all four are present within the 1.5 Mb minimal region believed to be critical in 22q11.2-related disorders (Fig. 3-4).

a. It is surprising that a number of 22q11.2 genes (e.g. Comt), with reported neuronal phenotypes, show no evidence of gene expression perturbations. Can the authors speculate how this might be?

b. How did the authors assess general toxicity of Cas9 in neurons?

c. A weakness of Cas9 screens can be an inability to detect critical roles of lowly expressed genes.

Can the authors test if there is a relationship between gene expression levels and the hits reported?

d. Can the authors speculate about the potential non-cell-autonomous impact of knockout cells on each other?

e. The authors include arrayed validation using eight individual AAV viruses of four 22q11.2 hit genes (Dgcr8, Dgcr14, Gnb1l, and Ufd1l) and four 22q11.2 genes with no apparent transcriptional phenotype (Comt, Med15, Ranbp1, and Pi4ka), which were then individually injected into distinct mice (SI Fig. 6,7). How similar/correlated were the pooled and arrayed gene program scores (SI Fig 11)?

f. Dosage: The authors attempted to computationally resolve the extent that CRISPR pool result in heterozygous or knockout gene deletions. Unexpectedly, they inferred that heterozygous and homozygous groups have undistinguishable expression of the genes previously used to define transcriptional phenotypes (SI. Fig. 8).

i. For at least one of four 22q11.2 hit genes (Dgcr8, Dgcr14, Gnb1l, and Ufd1l), it would be extremely informative to empirically test this in confirmed heterozygous and homozygous neurons.

ii. For Dgcr8, CRISPRi-mediated Dgcr8 mRNA reduction was comparable to the values observed for 22q11.2 DS (SI Fig. 9). What was the average and distribution of single cell knockdown of Dgcr8 observed by CRISPRi? What was the correlation of transcriptomic profiles of Dgcr8 CRISPRi to CRISPRko?

2. Physiological relevance (Fig. 4-5): The authors highlight cell-type-specific gene set enrichment of snRNAseq DEGs in 8 week old 22q11.2 (LgDel^{+/-} and LgDel^{+/+}) mice (Fig. 4d) and by pooled CRISPRko in 6-8 week old AAV mice (Fig. 5). This is an interesting analysis that could be much more comprehensively examined and validated.

a. Developmental impact: Perhaps beyond the scope of this report, but it would be extremely informative to contrast the impact of in vivo fetal and adult AAV-CRISPR knockout of 22q11.2 genes. Even if no new CRISPR experiments are performed, perhaps new or existing bulk or snRNAseq datasets from LgDel^{+/-} and LgDel^{+/+} mice could inform the extent of overlap of CRISPR ko DEGs across development.

b. Convergence: The authors report very little shared DEGs (SI Fig. 11), biological processes / gene ontology (SI Fig. 10-11) between the four 22q11.2 hit genes (Dgcr8, Dgcr14, Gnb1l, and Ufd1l). Is there any evidence instead that DEGs of these four genes are co-expressed and/or form interconnected downstream networks? If not, might this be explained by cell type specific effects in glutamatergic or GABAergic neurons?

c. Combinatorial perturbation: It is important to note that individual 22q11.2 neurons are never disrupted in isolation. It would be extremely informative to compare medium and very high titer AAV-CRISPR (to achieve knockout of all four 22q11.2 genes in the same cell), in order to explore additive effects between risk genes specifically in glutamatergic neurons and interneurons.

d. Rescue: Perhaps beyond the scope of this report, but it would be extremely valuable to attempt a CRISPRa rescue of the four genes in heterozygous 22q11.2 mice. This would allow testing of point-of-no-return and direct assessment of whether these effects are developmental or indeed specific to post-mortem neurons.

e. Although the authors show an enrichment of SCZ, BP, ASD, ADHD genes by gene ontology, it's unclear how these disease gene lists were generated. Is there an enrichment of 22q11.2 DEGs for the genes predicted to be genetically regulated by GWAS loci or rare genes with mutations associated with ASD, SZ or other psychiatric disorders? Can the authors also test for enrichment of postmortem disease signatures as reported by the CommonMind Consortium, NIH HBCC and/or PsychEncode UCLA (Geschwind) analyses?

f. Are gene expression differences validated in existing 22q11.2 postmortem or hiPSC datasets (ie. hiPSC PMID: 27846841 & 32989314?) Is there any evidence to support a recent report that mitochondrial defects exist in 22q11.2 neurons (PMID: 30833507)?

Author Rebuttals to First Revision:

AAV-mediated single-nucleus CRISPR screening enables functional dissection of the 22q11.2 deletion syndrome in vivo

Referees' comments:

Referee #1 (Remarks to the Author):

We are generally satisfied and happy with the revisions. The data is a bit sparse in areas (i.e. 1 UMI per gRNA cutoff seems arbitrary and maybe your real UMIs are at a higher cutoff just from looking at the graphs, but it's honestly hard to tell because individual gRNA UMI counts are generally low). But the fact that your arrayed single guide validation data lines up very well with the pooled data supports the gRNA assignments. You did make changes to the text where we suggested which is appreciated. It is a very impressive and exciting piece of work!

We thank the reviewer for their time and thoughtful suggestions.

Referee #2 (Remarks to the Author):

The manuscript remains solid, and the authors have clearly attempted to earnestly respond to reviewer comments. I am happy with most of their responses. In particular, I think the more direct engagement with the challenges of studying heterozygous mutations in the revised main text substantially improves the presentation of the paper and homes in on a key challenge for the field moving forward. I also think the authors have made a decent case that they set a new standard for in vivo studies, though challenges remain.

My only real reservation concerns the supplementary notes that have been introduced, which don't meet the quality standard set by the rest of the paper.

Points:

1) I do think the sample multiplexing strategy in which cells from distinct animals were randomly mixed potentially obscures a major confounder, but obviously this can't be fixed after the fact. Just a point worth remembering for future studies.

2) Though I appreciate the insight into the authors' thinking, the new Supplementary Notes are unpolished, in places quite speculative, and seem out of place in a scientific paper in their current form. E.g. "AAV is vastly superior to LV" is not scientific, nor is referring to "truly differentially expressed genes" when discussing the output of your computational pipeline relative to others. Some example points:

1: I am not an expert in this world but this would really need to be reviewed for accuracy if included. When I suggested providing background for a general audience I meant a few sentences summarizing the main advantages in the introduction of the paper with some references.

2/3: These seem more at home in methods or as traditional supplemental figures.

4: There are claims here that I consider open to debate, and some speculative comparisons to approaches in other studies. If you really want to go down this road, then you must do side-by-side comparisons **in multiple contexts** and include the analyses in the main text. (I do not think this is essential for this paper at all.) I think your main point is covered by reference to Squair et al., plus Supplementary Note Figure 4 could be a supplemental figure.

6: Completely speculative. Delete or place the points in discussion.

7: Why is this not in main text if the point is important? It's not cited at all.

Overall I don't see why these notes are needed. Honestly I'd prefer that the authors just remove them, or shorten them substantially to focus on what they did (rather than what they might do or their opinions on what other people did). If the authors think it is essential to include them I can do another round of review to at least provide feedback on some of the points that I think are overstated.

We thank the reviewer for their insights and enthusiasm.

To address their final minor concerns, we implemented what the reviewer suggested and removed the supplementary note and re-wrote parts of the manuscript to include essential information.

Referee #3 (Remarks to the Author):

The authors have done an excellent job addressing review comments in-depth.

As I mentioned in my initial review, I still feel that AAV-Perturb-Seq is a misnomer and that the technique (which uses ECCITE-seq to do 5' direct capture) should instead be called AAV-ECCITE-seq to avoid further confusion in an already confusing field of CRISPR perturbomics techniques.

We thank the reviewer for their appreciation of our hard work to address their concerns.

As for naming the method, indeed the nomenclature around CRISPR perturbomics techniques is confusing and there is little consensus about proper terminology. There are many methods with different chemistries offering different pros and cons and opportunities for multimodal measurements. After repeating a thorough literature review, we found that the term Perturb-seq predominates when it comes to describing methods and experiments where CRISPR screens are combined with single-cell transcriptomics. We believe the term has evolved from describing a precise method to a catch-all for anything CRISPR screening plus scRNA-seq. ECCITE-seq on the other hand is almost exclusively mentioned in the context of indexing epitopes for multi-modal omics. This concept is in the name: expanded CRISPR-compatible cellular indexing of transcriptomes and epitopes by sequencing (ECCITE-seq).

Without having a perfect term to use, we are left to make an imperfect decision. We believe that selecting the predominant catch-all term in the literature (Perturb-seq) is slightly more informative to readers overall than ECCITE-seq, which we believe will mislead readers into assuming we also performed epitope indexing. However, because the reviewer is correct in that ECCITE-seq was the first to describe 5' direct capture of guide RNAs, we have revised the text to make this emphatically clear.

Referee #4 (Remarks to the Author):

For the authors:

In their revised manuscript entitled “AAV-mediated single-nucleus CRISPR screening enables functional dissection of the 22q11.2 deletion syndrome in vivo”, Santinha et al apply an in vivo single-cell CRISPR screen, termed AAV-Perturb-seq, to systematically dissect the phenotypic landscape underlying 22q11.2 deletion syndrome genes in the adult mouse brain prefrontal cortex. They identified three 22q11.2-linked genes that explain approximately 40% of the transcriptional changes observed in a 22q11.2 deletion mouse model, suggesting that active disruption in adult neurons, rather than exclusively developmental process, dysregulates RNA processing and synaptic function in 22q11.2 deletion syndrome. This is an innovative paper and the first to apply pooled CRISPR screening to dissect the genes causally related to a highly penetrant psychiatric CNV in post-mitotic neurons. The authors systematically include a number of thoughtful empirical validations of key gene perturbation signatures. With further characterization of the downstream mechanisms linking the top three 22q11.2 genes, this manuscript could be suitable for publication in Nature.

We thank the reviewer for their enthusiasm, careful review of the manuscript, and suggestions for expanding the utility of the technology.

Major Concerns:

1. Pooled Cas9 CRISPR screening in vivo of 22q11.2 genes (Fig. 1-4): The authors validated in vivo titration of AAV tail vein injection to independently express either mTagBFP, Venus, or mCherry under the control of a ubiquitous CBh promoter. The higher viral dose led to an increased percentage of total infected nuclei (~ 34%), but almost half received more than one AAV particle. Thus, the authors selected the dose of 5.0×10^9 AAV particles per animal to maximize the total number of cells infected with a single AAV (Fig. 1). They also selected a 5'-based capture approach, combining independent gRNA expression (pAS088) with 5' capture sequencing, as it more sensitively capture mRNA and gRNA information in AAV-infected nuclei (Fig. 2). Their bioinformatic approach was extremely rigorous, requiring not just detection of the gRNA, but also confirmation of significantly perturbed expression of the target gene in individual cells. For the 29 of the 37 genes in the 22q11.2 locus expressed in

adult mouse cortex, the authors designed two independent gRNAs and included five control gRNAs targeting mouse safe-harbor (SH) loci. They detected all gRNAs in the library, with an average of ~10 nuclei in microglia to 400 nuclei in interneurons; in total, they detected ~60,000 nuclei spanning 6 brain cell types and perturbation of all 22q11.2 genes expressed in the adult prefrontal cortex. Using a pseudobulk approach, the authors found significant transcriptional phenotypes in four gene perturbations across all neuron types (Dgcr8, Dgcr14, Gnb1l, and Ufd1l), as measured by the number of DEGs; all four are present within the 1.5 Mb minimal region believed to be critical in 22q11.2-related disorders (Fig. 3-4).

a. It is surprising that a number of 22q11.2 genes (e.g. Comt), with reported neuronal phenotypes, show no evidence of gene expression perturbations. Can the authors speculate how this might be?

The reviewer raises an interesting point that we have thought very carefully about. The lack of transcriptional phenotypes for some perturbations could be due to technical or biological aspects. Below we discuss our experimental efforts to rule out technical reasons and speculate on possible biological explanations.

1) AAV-Perturb-seq leads to consistent gene editing. One reason for the lack of transcriptional phenotype for any given perturbation could be low efficiency of Cas9-mediated gene editing. While current computational tools to generate gRNAs for Cas9 are robust, we cannot completely ignore that some gRNAs may lead to low percentage of gene editing (*i.e.*, low indel frequency). To test for this, we perturbed separately the four genes with a strong transcriptional change (Dgcr8, Dgcr14, Gnb1l, and Ufd1l) and four randomly chosen genes with no apparent transcriptional phenotype (Comt, Med15, Ranbp1, and Pi4ka). Indel analysis revealed that the percentage of mutated cells was similar across all tested genes, with the majority of edited cells harboring frame-shifting loss of function mutations in the targeted gene (**Figure 2e and Extended Data Figure 6a**). Here, we rule out that the lack of transcriptional phenotypes was due to the efficiency of gRNAs.

2) Genetic perturbations can lead to subtle changes not detected by scRNA-seq. It is well known that single-cell and single-nucleus RNA-seq are limited when it comes to detecting subtle changes in the transcriptome. Some examples of why include technical noise, limited

sensitivity for low-abundance transcripts, and high gene dropout rates¹. As scRNA-seq is not an absolute technique, i.e., it does not capture all available RNA molecules inside a cell, low-abundance transcripts tend to be less represented in the dataset which can confound data analysis and signal identification. Furthermore, single-cell data suffers from high gene dropout rates with a significant fraction of genes not detected in individual cells. These limit the identification of true subtle gene expression dynamics of individual cells. We tried to minimize the impact of these variables by performing pseudobulk DE identification (which is less sensitive to gene dropouts as counts are aggregated²) and focusing on truly differential expressed genes (FDR < 0.05). To provide more information and clarity, in our manuscript we elaborate the point as well as included references directing readers to published reviews that fully delve into single-cell limitations and mitigation strategies.

3) Genetic perturbations can lead to phenotypes not detectable by transcriptional readouts. While transcriptional analysis is standardly used to investigate cellular function and states, from genes encoded in the DNA to the function of the proteins encoded by them, there are multiple points of action to which a specific gene can contribute. Some genes may be involved in processes not detectable by RNA sequencing and thus appear as non-significant hits in such datasets. For example, loss-of-function mutations in genes that encode for proteins that transduce signals, such as certain types of ion channels or cell signaling proteins, may not lead to altered gene expression unless in the presence of an activating signal, such as an action potential or ligand, respectively. For clarity, we acknowledge this limitation of transcriptional readouts in our manuscript. In the future, with the emergence of new omics methods and further mechanistic experiments considering the activating signals, we will be able to get a full picture of genetic functions across the multiple levels of cellular events.

4) The 22q11.2 deletion syndrome is not unique to the brain. While in our manuscript we focus on the brain function of 22q11.2 locus genes, patients carrying the deletion can also suffer from Congenital heart disease, Immune deficiency, Gastrointestinal abnormalities, among others³. Thus, not all genes in the deletion are equally important across tissues and organs, with some presenting tissue-specific expression and function³.

5) Deletions in the 22q11.2 locus have different lengths. The most common deletion carried by 22q11.2 DS patients is 3 Mb long and affects 46 protein coding genes. About 5% of cases present a smaller 1.5 Mb deletion that affects a subset of the genes in the proximal deletion region and is typically classified as “minimal critical deleted region”. Evidence suggests that both deletion groups have similar phenotypic features, indicating that genes outside the 1.5 Mb region may be less relevant. Interestingly, in the screen we targeted genes across the 3 Mb deletion, but all the gene hits identified by us are inside the 1.5 Mb region.

6) Many 22q11.2 locus genes don't lead to apparent phenotypes. Genes of the 22q11.2 locus have been studied in isolation with orthogonal methods other than transcriptomic analysis. For instance, mouse models carrying heterozygous deletion of *Comt* (encodes an enzyme that degrades catecholamines) have shown no impairment in physiological and behavioral tests (*e.g.*, including prepulse inhibition, social behaviors, and anxiety-related behaviors) and its functional relevance to brain function and neuropsychiatric disorders remains unclear⁴. Similar observations have been made for a subset of 22q11.2 genes related with mitochondrial function (*Mrpl40*, *Slc25a1*, *Tango2*, *Txnrd2*, and *Zddhc8*) with prepulse inhibition remaining at normal levels in mice carrying perturbations of those genes. While mitochondria dysfunction seems to be strongly related with Parkinson's disease (where the principal phenotype is neuronal death), its contribution to neuropsychiatric diseases is unclear. Another 22q11.2 gene, *Tbx1*, a transcription factor involved in organ formation during embryonic development, has been implicated as a strong modulator of cortical abnormalities related to 22q11.2 DS. Interestingly, this gene is not expressed in the adult brain, which indicates that it acts during development and perturbing it in adults would not lead to the same (or any) transcriptional phenotype.

In sum, we believe that AAV-Perturb-seq is robust to false negatives despite the technical limitations of Cas9-mediated gene editing and scRNA-seq. On the biological side, 22q11.2 locus genes may have time-, tissue-, organ-, and/or activity-specific roles. Our study was designed to study the activity of all expressed 22q11.2 genes in mature brain cells, but similar studies during development, in other organs, or considering activating stimuli would be extremely informative to fully dissect the broader functionalities of all implicated genes.

b. How did the authors assess general toxicity of Cas9 in neurons?

While we did not assess the general toxicity of Cas9 in neurons in this study, we carefully assessed this previously when establishing the Cas9 mouse model⁵. In our 2014 paper, we performed both an electrophysiological characterization (**Figure 1c-h**) and a histology-based toxicity analysis (**Figure S1**) and found no difference between wild-type and Cas9 expressing cells. Furthermore, in another unrelated study not performed by us, the authors performed several body-wide measurements of various tissues and found that Cas9 expression had no impact on gross anatomical measurements, tissue function, or metabolism⁶.

c. A weakness of Cas9 screens can be an inability to detect critical roles of lowly expressed genes. Can the authors test if there is a relationship between gene expression levels and the hits reported?

We took this possibility into consideration while designing our screen and also assessed the possibility while analyzing our data.

1) We target genes expressed in the adult mouse brain. The human 22q11.2 locus is composed of 45 protein coding genes, of which 37 are conserved in mouse. To make sure we focus on genes with detectable RNA-seq expression in brain cells, we used a publicly available scRNA-seq mouse brain atlas (Dropviz⁷) to identify which genes are expressed in adult mouse brain. We identified 29 genes and design our screen focusing on these (**Extended Data Figure 3a**).

2) There is no apparent correlation between gene expression levels and screen hits.

Looking at the expression of our hit genes (Dgcr8, Dgcr14, Gnb1l, and Ufd1l) across cell types, none of them is among the highest expressed genes (**Rebuttal Figure 1**). Our hits Dgcr14 and Gnb1l show less average expression than most other target genes while the other hits have average expression comparable to non-hit genes. Overall, this indicates that the expression level of the target gene was not a major confounder in hit detection.

Rebuttal Figure 1: Number of DEGs per perturbation. Genes are ranked from highest (left) to lowest (right) expression.

d. Can the authors speculate about the potential non-cell-autonomous impact of knockout cells on each other?

This is an interesting question that we have thought a lot about and while there could be interesting effects, they would be exceedingly rare and only contribute a negligible amount of noise to our dataset. Here is our logic.

We have shown that animals injected with AAV-Perturb-seq libraries will have mosaic cells of diverse genotypes sparsely scattered throughout the brain (**Figure 1b**). Potential non-cell autonomous impacts of perturbed cells could emerge either through short- or long-range interactions with wild-type or other perturbed cells. By ensuring sparse infection, our experimental methodology almost exclusively captures short and long-range interactions between perturbed and wild-type cells while drastically reducing the possibility of other interaction types between perturbed cells. Support for the relevance of interactions between wild-type and perturbed cells comes from the fact that our arrayed experiment aligns with our pooled findings and that our data recapitulates many findings from previous studies.

In terms of short-range interactions, for example in cells neighboring blood vessels where AAV infection rates are higher, perturbed cells may be in close proximity and interact with each other. However, because the overall infection rate is low, the combinatorial possibilities between all gene-gene interactions is large ($2^{(29 \text{ gene-targeted gRNAs} + 5 \text{ control gRNAs})} = 1156$), most perturbations have no measurable impact, and the existence of many control

gRNAs in the library, any specific interaction would be exceedingly rare in the tissue we sample and thus contribute a negligible amount of noise.

In terms of long-range interactions, for example where a perturbed neuron synapses with another perturbed neuron in a distal location, these could also emerge in principle.

However, using similar logic as above with short-range interactions, these events are also likely to be exceedingly rare and only contribute a negligible amount of noise.

e. The authors include arrayed validation using eight individual AAV viruses of four 22q11.2 hit genes (Dgcr8, Dgcr14, Gnb1l, and Ufd1l) and four 22q11.2 genes with no apparent transcriptional phenotype (Comt, Med15, Ranbp1, and Pi4ka), which were then individually injected into distinct mice (SI Fig. 6,7). How similar/correlated were the pooled and arrayed gene program scores (SI Fig 11)?

We explored the similarity/correlation between pooled and arrayed profiles from multiple angles.

1) Figure 3b shows the correlation of expression profiles between pooled and arrayed experiments across cell types. This plot highlights the strong correlation between pooled and arrayed data.

2) Figure 3c shows changes in the expression of genes across gene programs and different perturbations and cell types. This plot highlights perturbation-specific phenotypes that are consistent across gene programs and shared across neuron types.

3) Extended Data Figure 10a uses the same data as **Figure 3c** to show that hierarchical analysis clusters conditions (cell types from both arrayed and pooled experiments) by perturbation, ultimately highlighting the similarity between pooled and arrayed expression profiles.

f. Dosage: The authors attempted to computationally resolve the extent that CRISPR pool result in heterozygous or knockout gene deletions. Unexpectedly, they inferred that heterozygous and homozygous groups have undistinguishable expression of the genes previously used to define transcriptional phenotypes (SI. Fig. 8).

i. For at least one of four 22q11.2 hit genes (Dgcr8, Dgcr14, Gnb1l, and Ufd1l), it would be extremely informative to empirically test this in confirmed heterozygous and homozygous neurons.

We understand and acknowledge the concern raised by this reviewer and previous reviewers regarding the distinction between heterozygous and homozygous mutations in CRISPR experiments. First, to clarify, heterozygous and homozygous groups are indistinguishable in terms of the dysregulated genes that define the transcriptional phenotypes but are distinguishable in terms of the expression levels of those genes (**Extended Data Figure 9j-o**). Second, the mosaicism associated with CRISPR is a widely known and recognized challenge with gene editing in general. The only widely used approaches for controlling zygosity are through deriving single-cell clones or transgenic animals, working in haploid cells, or through haploidizing targeted loci. None of these strategies are compatible with high throughput in vivo gene editing experiments. Another possibility is to perform single-cell genotyping along with single-cell transcriptomics. We answered similar questions to earlier reviewers and have therefore summarized our responses that thoroughly address the point again here.

1) Combining single-cell genotyping along with single-cell transcriptomics. We agree with the reviewer that it would be extremely informative to have ground truth data on the genotype of perturbed cells, we therefore set out to develop a method that extracts the genotype of gene edited cells from single-cell library preparations. We have attempted single-cell genotyping by capture indel information directly from cDNA with PacBio long read sequencing. Of note, direct capture of indels from cDNA is not possible using conventional single-cell library construction protocols as these rely on sequencing of the 5' or 3' mRNA regions. We found limitations that did not give us confidence in the method: a) not all target genes are equally expressed and thus the long read library was mainly populated by higher expressed genes; b) as we use nuclei isolation, most cDNA molecules represent long unspliced mRNAs and tend to not cover the entire gene body, ultimately missing indel information; and c) long read sequence has limited throughput that does not scale well to the number of cells and sequencing reads necessary for a single-cell CRISPR screen dataset (more than 100 000 cells).

2) Computational stratification of zygosity cell states in perturbed nuclei. In our manuscript's section "Perturbation of 22q11.2-associated genes results in heterozygous and homozygous cells with similar transcriptional phenotypes" we proposed performing a diffusion map-based analysis to stratify cells along a continuum of genotypes, from wild type to homozygous. We found that, across perturbations, heterozygous and homozygous groups are undistinguishable in terms of the dysregulated genes that define the transcriptional phenotypes but are distinguishable in terms of the expression levels of those genes (**Extended Data Figure 8j-o**). While these results strongly suggest that heterozygous and homozygous mutations in *Dgcr8*, *Dgcr14*, and *Gnb1l* result in a continuous phenotype where the assessment of both or either zygosity captures the impact of the perturbation (and thus are relevant to haploinsufficiency), we agree with the reviewer's opinion that an empirical heterozygous state would solidify our conclusions. Given the challenge to perform such experiments directly in adult mice in vivo, we looked for suitable alternatives to model *Dgcr8* haploinsufficiency.

3) Modeling haploinsufficiency with CRISPRi. The LgDel model carries a heterozygous deletion in the mouse equivalent of the human 22q11.2 locus (mouse chromosome 16). In **Figure 4c** we show that genes with a deleted copy present a reduction of approximately 50% (LFC = -1) in their transcript levels when compared with wild-type litter mate control animals. Similar observations were reported for other mouse models and human patients^{8,9}. We hypothesized that CRISPRi-mediated knockdown may reduce target gene expression to levels observed in a heterozygous situation and thus simulate *Dgcr8* mRNA levels and ultimately the phenotype generated by haploinsufficiency (**Extended Data Figure 9**).

To explore this hypothesis, we prepared gRNAs targeting the promoter region of *Dgcr8* and control sequences. We injected AAV.PHP.B carrying gRNA and hSyn mediated GFP-KASH into a dCas9-KRAB mouse line (JAX stock #030000) and followed our AAV-Perturb-seq workflow. Across cell types, CRISPRi-mediated *Dgcr8* mRNA reduction was comparable to the values observed for LgDel, indicating our ability to model heterozygous expression. We also confirmed that CRISPRi (KD) and CRISPR (KO) mediated *Dgcr8* perturbations led to analogous transcriptional phenotypes. Support for this came from: a) high similarity between LFC values (**Extended Data Figure 9d**); b) genes considered DE by both experiments (**Extended**

Data Figure 9e); and c) the discovery of DE genes relevant to Dgcr8's known biological function, i.e., genes associated with the process "gene silencing by miRNA" (**Extended Data Figure 9f**).

All in all, our additional computational (zygosity stratification with diffusion maps) and experimental (modeling haploinsufficiency with CRISPRi) work provide strong evidence supporting our original conclusions. We believe that our efforts to provide a new approach to assessing zygosity-dependent cell states and extend the applicability of AAV-Perturb-seq to CRISPRi technology add further value to our manuscript and to the genetic screening field. In the discussion section we also acknowledge that we, along with seemingly all other non-clonal gene editing approaches, cannot perfectly model haploinsufficiency and therefore such results require further validation. Nonetheless, we believe that AAV-Perturb-seq provides a powerful way to prioritize such efforts.

ii. For Dgcr8, CRISPRi-mediated Dgcr8 mRNA reduction was comparable to the values observed for 22q11.2 DS (SI Fig. 9). What was the average and distribution of single cell knockdown of Dgcr8 observed by CRISPRi? What was the correlation of transcriptomic profiles of Dgcr8 CRISPRi to CRISPRko?

1) The average of Dgcr8 knockdown observed in CRISPRi is plotted in **Extended Data Figure 9c** as expression LFC calculated against control nuclei. Across cell types, Dgcr8 mRNA reduction was equal to LFC - 0.8 (approximately 40% mRNA reduction) comparable to the values observed in our LgDel dataset (**Figure 4c**), indicating our ability to model heterozygosity.

2) **Rebuttal Figure 2** shows the side-by-side expression of Dgcr8 in control and CRISPRi perturbed nuclei and highlights the strong reduction in Dgcr8 expression across single cells of all cell types (**Extended Data Figure 9b**) in the Dgcr8-perturbed group.

3) We report the correlation between Dgcr8 CRISPRi and CRISPRko with cosine similarity in **Extended Data Figure 9d**. The similarity between transcriptional profiles was approximately 80%, 91%, and 89% for Supp. Layer Neurons, Deep Layer Neurons, and Interneurons, respectively.

Rebuttal Figure 2: UMAP embedding showing single-nucleus expression of Dgcr8 mRNA in control and Dgcr8 CRISPRi perturbations.

2. Physiological relevance (Fig. 4-5): The authors highlight cell-type-specific gene set enrichment of snRNAseq DEGs in 8 week old 22q11.2 (LgDel^{+/-} and LgDel^{+/+}) mice (Fig. 4d) and by pooled CRISPRko in 6-8 week old AAV mice (Fig. 5). This is an interesting analysis that could be much more comprehensively examined and validated.

a. Developmental impact: Perhaps beyond the scope of this report, but it would be extremely informative to contrast the impact of in vivo fetal and adult AAV-CRISPR knockout of 22q11.2 genes. Even if no new CRISPR experiments are performed, perhaps new or existing bulk or snRNAseq datasets from LgDel^{+/-} and LgDel^{+/+} mice could inform the extent of overlap of CRISPR ko DEGs across development.

As the reviewer mentions, we think that performing AAV-Perturb-seq across many developmental timepoints is extremely interesting but perhaps beyond the scope of the current manuscript. Nonetheless, we now add a new LgDel^{+/-} dataset from newborn pups that can give us a glimpse of the developmental picture at a single time point.

We generated a dataset containing scRNA-seq from 2 WT and 2 LgDel^{+/-} P1.5 mouse pups (**Rebuttal Figure 3a**). P1.5 is the developmental stage at which circuits and cell identities are being specified and study of transcriptional consequences at this stage could elucidate the role of 22q11.2 locus genes in circuit and neuronal maturation¹⁰. We identified major cell types as in the adult dataset (**Figure 4b**), namely superficial and deep layers neurons,

interneurons, and astrocytes (**Rebuttal Figure 3b**). Applying our pseudobulk approach, we identified a higher number of DEGs in adult neurons, with superficial layer neurons having the highest number of DEGs in both ages (**Rebuttal Figure 3c**). We then compared the biological processes altered in the pup and adult datasets (**Rebuttal Figure 3d**). We observed a down-regulation of neuronal transmission, signaling, and development processes across the two ages. While in the adult dataset these processes are significantly ($FDR < 0.05$) down-regulated for all neuron types examined, at P1.5 the cell types generally follow a similar trend, but only superficial layer neurons reach statistical significance. These results indicate that neuronal activity processes seem to be already altered in newborns and then become further dysregulated throughout development. On the opposite side, we found up-regulated processes unique to the P1.5 dataset. These processes are related with gene expression regulation, both at the DNA level (regulation of histone modification) and at the RNA level (RNA metabolic processes). Interestingly, these biological functions are similar to the ones identified as up-regulated in Dgcr14-perturbed neurons (**Figure 3c**), perhaps indicating an important role of Dgcr14 also at the P1.5 developmental stage.

As the reviewer suggests in **Comment 2f**, it would be interesting to compare our mouse in vivo results with human data. To do so, we focused on a recent publication⁹ that used induced pluripotent stem cells (iPSCs) from a cohort of 15 patients with 22q11.2 DS to generate human cortical spheroids (hCS) and measured transcriptional changes between control and 22q11.2 hCS. In **Figure 1f of their article**, the authors show a subset of biological terms altered across multiple timepoints (25, 50, 75, and 100 DIV). We explored **their Supplementary Table 6**, which contains the unfiltered list and significance of biological terms. For hCSs 75 DIV, we found up-regulation of gene expression processes and down-regulation of neuronal development and activity. These observations in a human neuronal model strongly recapitulate our findings in P1.5 mouse pups, especially superficial layer neurons (**Rebuttal Figure 3e**). At 100 DIV, we observe that 22q11 hCSs have reduced expression of genes related with neuronal activity and communication, but no longer show significant up-regulation of genes related with DNA and RNA processes. These results strongly compare with our mouse in vivo observations and indicate that while more mature neurons may require less activity of transcription-related genes, these cells suffer from reduced expression of signaling related genes throughout development.

In the revised manuscript we comment on the comparison between adult mouse and human data, include **Rebuttal Figure 3e** as **Extended Data Figure 14b**, and guide readers to literature references.

Rebuttal Figure 3: Transcriptional analysis of P1.5 LgDel^{+/-} pups. **a.** Experimental design to generate scRNA-seq of LgDel^{+/+} and LgDel^{+/-} brain cells isolated from P1.5 pups. **b.** UMAP embedding of brain cell types in the P1.5 dataset. **c.** Number of DEGs in the P1.5 and adult datasets for each neuron type. **d.** Biological processes dysregulated in P1.5 and adult LgDel^{+/-} neuronal cells. **e.** Biological processes dysregulated in hCSs at 75 (top) and 100 (bottom) DIV. Data from Khan et al, Nature Medicine, 2020.

b. Convergence: The authors report very little shared DEGs (SI Fig. 11), biological processes / gene ontology (SI Fig. 10-11) between the four 22q11.2 hit genes (Dgcr8, Dgcr14, Gnb1l, and Ufd1l). Is there any evidence instead that DEGs of these four genes are co-expressed and/or form interconnected downstream networks? If not, might this be explained by cell type specific effects in glutamatergic or GABAergic neurons?

Our enrichment analysis in **Figure 3c** and **Extended Data Figure 10** shows that perturbation-specific DEGs are associated with unique biological processes, with little overlap between perturbations. To follow the reviewer's suggestion and further explore possible co-expression and/or interconnected downstream networks and processes, we use STRING analysis to summarize the network of predicted associations between proteins dysregulated by Dgcr8, Dgcr14, and Gnb1l. After network plotting and clustering we identified 25 groups of genes with more than 7 elements. By labeling DEGs by their associated perturbation we observed that clusters are mainly dominated by genes associated with one perturbation but also contain a minor contribution of genes associated with other perturbations. To confirm that DEGs are grouped by biological function, we performed an enrichment analysis using the entire list of DEGs. This analysis revealed that DEGs not only cluster by their association with a perturbation, but also by their biological function (**Rebuttal Figure 4**), with RNA splicing and binding clusters largely populated by Dgcr14-perturbed genes and neuronal transmission clusters mainly dominated by Gnb1l-perturbed genes. Overall, these results are consistent with our original interpretation that transcriptional phenotypes are mostly perturbation-specific but there is indeed an intriguing possibility that there are interactions between downstream networks.

Rebuttal Figure 4: STRING network analysis of DEGs identified with AAV-Perturb-seq.

c. Combinatorial perturbation: It is important to note that individual 22q11.2 neurons are never disrupted in isolation. It would be extremely informative to compare medium and very high titer AAV-CRISPR (to achieve knockout of all four 22q11.2 genes in the same cell), in order to explore additive effects between risk genes specifically in glutamatergic neurons and interneurons.

We agree with the reviewer that combinatorial perturbation of our hit genes would be extremely interesting. While the idea of injecting very high titer of AAVs to achieve knockout of all four 22q11.2 genes in the same cell seems promising, we believe it would require more than that to do well. Below we explain our logic as well as reanalyze our dataset to see if we can gain any first insights into genetic interactions between our 22q11 hits.

1) Performing a genetic interaction screen on genes within the same locus would be extremely challenging and require new technologies and an entirely new study. It is well known that inducing multiple DNA DSBs within the same locus often results in a deletion in the intervening segment and can lead to larger chromosomal aberrations¹¹. Given that all 4 targets are in the same locus, we anticipate that the proportion of unwanted genetic outcomes would be prohibitive without a new method coupling long-range genotyping with single-cell genotyping. We believe that this is out of the scope of the current study.

Another alternative could be CRISPRi, however, it is also well known that the impact of CRISPRi can extend beyond the target gene to neighboring genes¹². CRISPRi could possibly work for the interaction screen but ensuring robustness would require extensive testing and benchmarking of different CRISPRi technologies and gRNAs for each target gene. We also believe that this is out of the scope of the current study.

Yet another alternative would be to use a precision gene editing technology that does not depend on DSBs (ie base editing) and design an experiment to have all combinations of 2, 3, and 4 perturbations. While we believe that this would be the best strategy for robustly assessing genetic interactions in the context of 22q11, this would require establishing an entirely new in vivo methodology, and also likely creating a base editing transgenic mouse, which is out of the scope of the current study.

2) Exploring combinatorial perturbations in double infected nuclei. In our manuscript, we focused and reported on cells expressing a unique gRNA (MOI = 1) as those represent single

perturbations. However, our dataset also contains cells infected with two distinct AAVs (MOI = 2) that carry two different perturbations (**Extended Data Figure 4a**). We explored the transcriptional phenotype of these double infected cells, with the caveat of two key limitations, including the low number of cells per double per cell type as well as the possibility of chromosomal deletions or aberrations. Given the similarity of perturbation-induced changes across neuron types, we grouped all neurons carrying the same double perturbation to increase the number of cells per condition. Then, focusing our attention on Dgcr8, Dgcr14, Gnb1l, and all doublet combinations of these genes, we ran our analysis pipeline in both single and double perturbed groups. We fitted the regression model shown in **Figure 4f** to decompose the signature induced by combinatorial perturbations in terms of each perturbation alone. As a result, the model coefficients highlight how much the signal of each single perturbation is concordant with the combination. The correlation between true and predicted double profiles indicates how much of the double perturbation profile is explained by the combination of single perturbations. We observed an overall additive effect across hit combinations (Dgcr8-Dcgr14, Dgcr8-Gnb1l, and Dgcr14-Gnb1l) with a high correlation between true and predicted profiles (**Rebuttal Figure 5**). These results indicate that single perturbations mostly have unique transcriptional phenotypes and that double infected cells present a transcriptional phenotype altered by both perturbations.

While we consider these results interesting, given the low number of cells per condition and the possible aberrant genomic alterations induced by DSBs in double infected cells, we lack the confidence to include these analyses in the revised manuscript.

Rebuttal Figure 5: Heatmap showing the LFC values for the top predicted genes in individual and double perturbations, the model predictions based on individual perturbations profiles, and the correlation (dcor) between true and predicted double perturbation profiles.

d. Rescue: Perhaps beyond the scope of this report, but it would be extremely valuable to attempt a CRISPRa rescue of the four genes in heterozygous 22q11.2 mice. This would allow testing of point-of-no-return and direct assessment of whether these effects are developmental or indeed specific to post-mortem neurons.

Thank you for your comment and suggestion for this potential follow-up experiment. We agree that performing rescue experiments in the adult LgDel mouse model would be an extremely exciting direction to pursue and could potentially provide valuable therapeutic insights. While CRISPRa-based multiplexed activation of the 4 identified genes in the disease model mouse would be possible from a gene editing perspective, we believe that such a rescue experiment deserves further efforts and controls to assure activation of all 4 genes in all cells and fine-tune the dosage of each gene to ultimately assess phenotypic rescue. Furthermore, CRISPRa can also act on neighboring genes so one would have to be very careful in attributing the rescue to the specific target gene and not their neighbors (albeit rescuing the expression of all genes within the locus is an interesting rescue strategy in and

of itself). As the reviewer mentioned, we also believe that this experiment is beyond the scope of the current manuscript.

e. Although the authors show an enrichment of SCZ, BP, ASD, ADHD genes by gene ontology, it's unclear how these disease gene lists were generated. Is there an enrichment of 22q11.2 DEGs for the genes predicted to be genetically regulated by GWAS loci or rare genes with mutations associated with ASD, SZ or other psychiatric disorders? Can the authors also test for enrichment of postmortem disease signatures as reported by the CommonMind Consortium, NIH HBCC and/or PsychEncode UCLA (Geschwind) analyses?

In **Figure 5** and **Extended Data Figure 15e** we performed enrichment of SCZ, BP, ASD, and ADHD genes using curated datasets from DisGenNet¹³. Disease-specific lists of genes in DisGenNet integrate data from expert curated repositories, GWAS catalogues, animal models, and the scientific literature. Given the refinement of this dataset, we feel confident with our enrichment analysis. We have revised the manuscript to clearly mention the used datasets.

f. Are gene expression differences validated in existing 22q11.2 postmortem or hiPSC datasets (ie. hiPSC PMID: 27846841 & 32989314?) Is there any evidence to support a recent report that mitochondrial defects exist in 22q11.2 neurons (PMID: 30833507)?

In our answer to **Comment 2a** and in **Rebuttal Figure 3e** we show that our in vivo mouse data (P1.5 and adult) recapitulates observations made in human cortical spheroids generated with iPSCs from 22q11.2 DS patients (Khan et al 2020, PMID:32989314). We have changed our manuscript to include this information and direct readers to the reference human datasets.

In **Extended Data Figure 14a** we show all non-redundant biological processes altered in LgDel+/-, with no apparent evidence of mitochondria related defects. To further explore a possible dysregulation of mitochondrial functions, we plotted the expression of mitochondria-related genes from Khan et al 2020⁹. In our mouse dataset (**Rebuttal Figure 6**), while we found a slight down-regulation of Mrrf, a component of the mitochondrial translational machinery, none of the tested genes reached the significance threshold (LFC > 0.5 & FDR < 0.01).

In sum, our current dataset does not support mitochondrial dysfunction in the LgDel+/- mouse model.

Rebuttal Figure 6: Log-fold change values for the mitochondria-related gene set used in Khan et al 2020.

References

1. Luecken, M. D. & Theis, F. J. Current best practices in single-cell RNA-seq analysis: a tutorial. *Molecular Systems Biology* **15**, e8746 (2019).
2. Squair, J. W. *et al.* Confronting false discoveries in single-cell differential expression. *Nature Communications* 2021 12:1 **12**, 1–15 (2021).
3. Du, Q., de la Morena, M. T. & van Oers, N. S. C. The Genetics and Epigenetics of 22q11.2 Deletion Syndrome. *Frontiers in Genetics* **10**, 1365 (2020).
4. Zinkstok, J. R. *et al.* Neurobiological perspective of 22q11.2 deletion syndrome. *The Lancet Psychiatry* **6**, 951–960 (2019).

5. Platt, R. J. *et al.* CRISPR-Cas9 knockin mice for genome editing and cancer modeling. *Cell* **159**, 440–455 (2014).
6. Bond, S. T. *et al.* Tissue-specific expression of Cas9 has no impact on whole-body metabolism in four transgenic mouse lines. *Molecular Metabolism* **53**, 101292 (2021).
7. Saunders, A. *et al.* Molecular Diversity and Specializations among the Cells of the Adult Mouse Brain. *Cell* **174**, 1015-1030.e16 (2018).
8. Meechan, D. W. *et al.* Modeling a model: Mouse genetics, 22q11.2 Deletion Syndrome, and disorders of cortical circuit development. *Progress in Neurobiology* **130**, 1–28 (2015).
9. Khan, T. A. *et al.* Neuronal defects in a human cellular model of 22q11.2 deletion syndrome. *Nat Med* **26**, 1888–1898 (2020).
10. Jabaudon, D. Fate and freedom in developing neocortical circuits. *Nat Commun* **8**, 16042 (2017).
11. Kosicki, M., Tomberg, K. & Bradley, A. Repair of double-strand breaks induced by CRISPR–Cas9 leads to large deletions and complex rearrangements. *Nat Biotechnol* **36**, 765–771 (2018).
12. Tycko, J. *et al.* Mitigation of off-target toxicity in CRISPR-Cas9 screens for essential non-coding elements. *Nat Commun* **10**, 4063 (2019).
13. Piñero, J. *et al.* The DisGeNET knowledge platform for disease genomics: 2019 update. *Nucleic Acids Research* **48**, D845–D855 (2020).

Reviewer Reports on the Second Revision:

Referees' comments:

Referee #2 (Remarks to the Author):

The authors have addressed my concerns and I believe the paper is suitable for publication in Nature.

Referee #4 (Remarks to the Author):

The authors provided a thoughtful and detailed response to my questions. Although limited new data or analyses are presented in the revised manuscript, their textual responses have clarified the analyses and the limitations of their studies, addressing my major concerns. I am happy to recommend this manuscript for publication at Nature at this time.